# Learning Canonical Representations for Unified 3D Molecular Modeling

## Abstract

3D molecular foundation models must handle diverse tasks, from predicting scalar properties to generating 3D coordinates, yet existing approaches force a choice between invariant and equivariant architectures, each with inherent limitations. We show that this tradeoff is unnecessary. By canonicalizing molecules into a learned standard pose before encoding, a single non-equivariant model can support both invariant and equivariant tasks, and we theoretically demonstrate that this approach overcomes the fundamental constraints of prior paradigms. Beyond standard benchmarks, we further propose evaluation criteria that directly probe representation quality independent of downstream tasks. Pretrained on large-scale molecular data, our model consistently rivals methods purpose-built for each task across property prediction, generation, and structure-based optimization. Our code is available at https://github.com/themolsubmission/TheMol

## 1. Introduction

Three-dimensional molecular structure encodes richer information than any other representation, making it the natural substrate for molecular modeling. To leverage this geometric richness, recent work has developed SE(3)-invariant and -equivariant architectures that respect the symmetries of 3D space. However, task-specific datasets remain small, limiting generalization. This has motivated a shift toward foundation models: pretraining on large-scale unlabeled molecular data, then finetuning across diverse downstream tasks.

Yet a fundamental question remains unaddressed. Existing molecular foundation models extract embeddings that are either invariant or equivariant, committing to a single symmetry regime at the representation level. This raises a natural

---
[1]Anonymous Institution, Anonymous City, Anonymous Region, Anonymous Country. Correspondence to: Anonymous Author <anon.email@domain.com>.

Preliminary work. Under review by the International Conference on Machine Learning (ICML). Do not distribute.

concern: Can such models truly serve as general-purpose foundations when downstream tasks require different output types (invariant vs. equivariant)? In this work, we answer this question. We show that the invariant-versus-equivariant dichotomy is not an inherent constraint but an artifact of conventional design choices. Through canonicalization, a single representation can flexibly support both invariant and equivariant tasks without requiring separate architectures for each regime. We formalize this insight theoretically, validate it empirically across diverse benchmarks, and propose evaluation criteria that assess representation quality beyond task-specific performance. Our contributions are as follows:

- We theoretically analyze the limitations of invariant, equivariant, and augmentation-based representations when serving diverse downstream tasks, and show that canonicalization overcomes these constraints.
- We present THEMOL, the first canonicalization-based 3D molecular representation learning framework that unifies property prediction, generation, and optimization within a single architecture.
- We propose evaluation criteria that assess representation quality beyond benchmark performance, and demonstrate competitive results against methods specifically designed for each task.

## 2. Background and Representation Taxonomy

**SE(3)-Invariant Representations.** Let $\mathbf{x} \in \mathbb{R}^{N \times 3}$, $\mathbf{h}$ denote atomic coordinates and auxiliary features, respectively. A representation model $f$ is *SE(3)-invariant* if $f(g \cdot \mathbf{x}, \mathbf{h}) = f(\mathbf{x}, \mathbf{h}), \forall g \in SE(3)$. This is commonly implemented either by constructing invariant inputs (e.g., inter-atomic distances/angles) or by producing intermediate equivariant features to obtain a global embedding. Representative examples of models designed for specific task that adopt this approach include SchNet (Schütt et al., 2017) and DimeNet(Gasteiger et al.). For notational convenience, we refer to this representation scheme as Repr.$\mathcal{A}$ hereafter.

**SE(3)-Equivariant Representations.** An SE(3) transform $g = (R, t)$ acts as $g \cdot \mathbf{x} = R\mathbf{x} + t$. A representation model $f$ is *SE(3)-equivariant* if $f(g \cdot \mathbf{x}, \mathbf{h}) = \rho(g) f(\mathbf{x}, \mathbf{h}), \forall g \in SE(3)$, where $\rho$ is a chosen group representation on the output space. In practice, translation is handled by centering or by using relative displacements $(\mathbf{x}_i - \mathbf{x}_j)$ within the network.

Representative examples of models designed for specific task that adopt this approach include EGNN (Satorras et al., 2021) and EquiformerV2 (Liao et al., 2023). Likewise, we henceforth denote this representational approach Repr.$\mathcal{B}$.

**Rotation-Augmented Non-Equivariant Representations.** This type uses a non-equivariant architecture $f_\theta$ and enforces rotation robustness through data augmentation. Training typically minimizes an augmentation-averaged risk: $\min_\theta \mathbb{E}_{(\mathbf{x},\mathbf{h},y)} \mathbb{E}_{R\sim\mu}[\ell(f_\theta(R\cdot\mathbf{x}, \mathbf{h}), y)]$, where $\mu$ is a distribution over $SO(3)$ and $y$ is a task-specific target. This approach finds its embodiment in task-oriented models such as Tabasco (Vonessen et al., 2025a), and extends its reach to protein modeling through architectures like AlphaFold3 (Abramson et al., 2024). Likewise, we henceforth denote this representational approach Repr.$\mathcal{C}$.

# 3. 3D Molecular Representation with Canonicalization

## 3.1. Theoretical Revisting and Proposed Framework

Prior work on 3D molecular representation learning typically builds symmetry into the architecture by adopting either invariant or equivariant or non-equivariant layers according to the downstream task. This design is often reasonable when a single task has a clear symmetry requirement and the architectural bias is well aligned with it.

However, for foundation representations intended to support heterogeneous tasks with different symmetry requirements, fixing the layer type a priori makes it difficult to achieve uniformly high efficiency across tasks. For instance, one may use embeddings derived from invariant features for coordinate-level outputs, or conversely use embeddings derived from equivariant features for invariant property prediction. In such settings, at least one of expressivity, sample efficiency, or inference cost typically deteriorates. Although some architectures incorporate both invariant and equivariant layers, the embedding ultimately extracted for downstream finetuning remains confined to a single symmetry type (Ji et al., 2024; Zhou et al., 2023). We discuss potential limitations of Repr.$\mathcal{A}$, Repr.$\mathcal{B}$, and Repr.$\mathcal{C}$ for foundation model setting, and then present **Repr.$\mathcal{Z}$** as an alternative that mitigates these limitations.

### 3.1.1. PROBLEM OF REPR.$\mathcal{A}$

Invariant representations (e.g., distance/internal-coordinate based features) can be efficient for scalar property prediction. Yet they can be structurally limited for foundation settings. First, distance-only invariants are effectively SE(3)-invariant, which can erase reflection-odd signals (Schoenberg, 1935; Adams et al., 2021) and thus fail to distinguish certain chiral information, as shown Figure. 5. Second, purely invariant latents fundamentally lack the

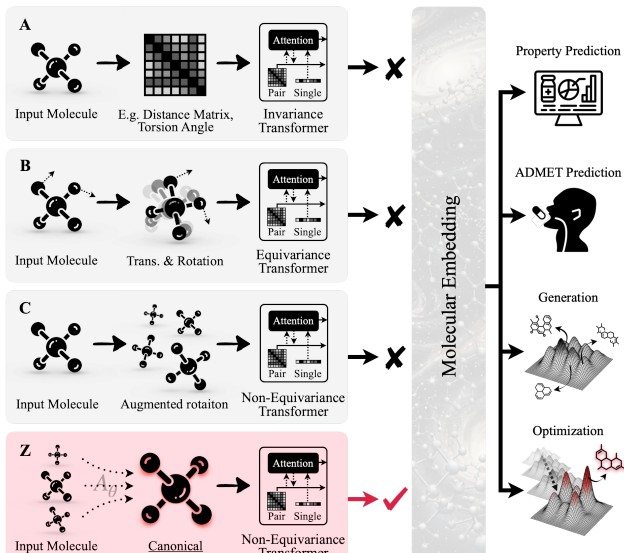

*Figure 1.* Overview of the THEMOL framework. Repr.$\mathcal{A}$ encode molecules using geometric invariants. Repr.$\mathcal{B}$ maintain coordinate structure throughout the network. Repr.$\mathcal{C}$ apply random rotations during training with a non-equivariant backbone. (Z) Our **Repr.$\mathcal{Z}$** first aligns molecules to a learned canonical pose, then processes them with a non-equivariant transformer, enabling both invariant and equivariant downstream tasks within a unified framework.

pose/gauge information required for coordinate reconstruction or generation—they capture *what* the molecule looks like, but not *how* it is oriented (Bronstein et al., 2021). Representative examples of foundation models adopting this representation include Uni-Mol[1] (Zhou et al.; Ji et al., 2024) and JMP (Shoghi et al.). For further details on this, please refer to the Appendix D.1.

### 3.1.2. PROBLEM OF REPR.$\mathcal{B}$

Equivariant models are natural for coordinate-level outputs, but they introduce their own challenges for foundation representations. In an autoencoder setting, if the latent includes coordinates ($\mathbf{z}_\mathbf{x} \in \mathbb{R}^{N\times 3}$), the model can trivially copy input coordinates to the latent and reconstruct them directly, without learning meaningful abstractions (Xu et al., 2023b; Chen et al., 2025; Luo et al., 2025; Hong et al., 2024a; Zhang et al., 2025; Irwin et al., 2025). While regularizers can mitigate this, they require careful tuning and may limit representational capacity. For inherently invariant tasks, equivariant backbones must still learn to suppress rotational nuisance (typically via rotation augmentation or group averaging) which can increase training and/or inference cost. Moreover, in models such as You et al. (2024) that split latents into ($\mathbf{z}_\mathbf{x}, \mathbf{z}_\mathbf{h}$), relying only on the invariant part ($\mathbf{z}_\mathbf{h}$) for prediction can discard geometric information, while including ($\mathbf{z}_\mathbf{x}$) reintroduces the above inefficiencies. Repre-

---

[1]Though using equivariant layers in the output head, the molecular embeddings themselves remain inherently invariant.

sentative examples of models adopting this representation include GeoLDM[2] (Xu et al., 2023b), MACE (Batatia et al., 2023). For a more detailed analysis, please refer to the Appendix D.2.

### 3.1.3. PROBLEM OF REPR.$\mathcal{C}$

Intuitively, a non-equivariant backbone trained with rotation augmentation faces a fundamentally difficult task: the same molecule appears with different coordinates at each iteration, yet the model must recognize the invariant structure behind these arbitrarily oriented inputs without explicit geometric priors.

Rotation augmentation minimizes an average-case objective over random orientations, but this does not control worst-case behavior: a predictor with low augmented risk may still vary substantially across rotations at test time. The only remedy is multi-view inference (explicitly averaging predictions over rotations) which is computationally expensive and scales poorly with the desired stability.

**Assumption 3.1** (Augmentation training with single-view inference). *A predictor $f$ is trained by minimizing the augmented risk $\mathcal{R}_{\mathrm{aug}}(f) := \mathbb{E}_{(\mathbf{x},\mathbf{h})}\mathbb{E}_{R\sim\mu}[\ell(f(R\mathbf{x},\mathbf{h}),y(\mathbf{x},\mathbf{h}))]$, where $\mu$ is the Haar measure on $SO(3)$ and $y$ is rotation-invariant. Inference uses a single view unless explicit group averaging is performed.*

**Proposition 3.2** (Augmentation lacks worst-case guarantees; invariance requires multi-view cost). *Under Assumption 3.1: (i) There exist predictors with arbitrarily small $\mathcal{R}_{\mathrm{aug}}(f)$ but arbitrarily large $\sup_R |f(R\mathbf{x},\mathbf{h}) - f(\mathbf{x},\mathbf{h})|$; (ii) Certifying $\varepsilon$-uniform invariance via test-time group averaging requires $K = \Omega(\varepsilon^{-3})$ views.*

Part (i) confirms that augmentation cannot guarantee worst-case stability; part (ii) shows that closing this gap via averaging incurs cubic cost in $\varepsilon^{-1}$ (since $SO(3)$ is 3-dimensional). Full analysis is in Appendix D.3.

### 3.1.4. PROPOSED REPRESENTATION, **REPR.$\mathcal{Z}$**

To address the limitations above, we propose a canonicalization-based representation learning framework. The core idea is simple: before feeding a molecule to the backbone, we first *rotate it into a canonical pose* determined by a learned alignment $A_\theta$. Since all rotated versions of the same molecule are mapped to the identical canonical representative, the backbone always sees the same input regardless of the original orientation. Representations are then learned in this canonical space using a standard (non-equivariant) backbone, and when coordinate-level outputs are needed, we *lift* the result back to the original frame via

---

[2]While not a foundation model, this is a pioneering work that explores 3D molecular representation via autoencoders.

the inverse of the selected rotation.

This canonicalization strategy addresses the specific limitations of Repr.$\mathcal{A}$ and Repr.$\mathcal{B}$. For Repr.$\mathcal{A}$: canonicalization aligns rotations only (not reflections), preserving chirality unlike distance-only $O(3)$-invariants; and the retained gauge $\widehat{R}_\theta$ provides the pose information needed for coordinate reconstruction, which purely invariant latents cannot support. For Repr.$\mathcal{B}$: equivariant layers constrain the latent to maintain coordinate structure (e.g., $\mathbb{R}^{N\times3}$) throughout the network, enabling pass-through degeneracy. In contrast, learning in canonical space with a non-equivariant backbone allows projection into higher-dimensional latent spaces (e.g., $\mathbb{R}^{N\times H}$ with $H \gg 3$), which structurally prevents trivial coordinate copying while enabling richer geometric representations. Consequently, a single shared backbone supports both invariant and equivariant objectives. In what follows, we formalize these intuitions.

**Setup: canonicalization map.** Let the atomic coordinates be $\mathbf{x} \in \mathbb{R}^{N\times3}$ and the per-atom features be $\mathbf{h}$. We remove the translational component via centering:

$$\mathbf{x}^{\mathrm{c}} := \mathbf{x} - \mathbf{1}\bar{\mathbf{x}}^\top, \qquad \bar{\mathbf{x}} := \frac{1}{N}\sum_{i=1}^N \mathbf{x}_i \in \mathbb{R}^3. \quad (1)$$

For a rotation $R \in SO(3)$, we denote the group action on coordinates by $\mathbf{x} \mapsto R\mathbf{x}$ (applied row-wise). We introduce a gauge/rotation selector $\widehat{\mathbf{R}}_\theta(\mathbf{x},\mathbf{h}) \in SO(3)$ that chooses a rotation for each input $(\mathbf{x},\mathbf{h})$, and define the canonicalization map as

$$A_\theta(\mathbf{x},\mathbf{h}) := \widehat{\mathbf{R}}_\theta(\mathbf{x},\mathbf{h})\,\mathbf{x}^{\mathrm{c}} \in \mathbb{R}^{N\times3}. \quad (2)$$

Intuitively, $A_\theta$ corresponds to "aligning the input molecule into a standard pose." A key condition is gauge consistency:

$$\widehat{\mathbf{R}}_\theta(R\mathbf{x}) = \widehat{\mathbf{R}}_\theta(\mathbf{x})R^\top \leftrightarrow A_\theta(R\mathbf{x}) = A_\theta(\mathbf{x}), \quad (3)$$

since $\mathbf{h}$ does not affect this operation, we omit it from the notation. That is, if canonicalization produces the same canonical representative even when the input is rotated, then rotations are eliminated in the canonical space.

Figure 2 (left) empirically validates this gauge consistency. When the same molecule is subjected to 1,000 random rotations, embeddings without canonicalization scatter across the representation space, whereas canonicalized embeddings collapse to a single point, confirming that the learned $\widehat{R}_\theta$ satisfies $A_\theta(Rx,h) = A_\theta(x,h)$ in practice.

**Proposition 3.3** (One backbone, two symmetry regimes). *Assume the canonicalization map satisfies*

$$A_\theta(R\mathbf{x},\mathbf{h}) = A_\theta(\mathbf{x},\mathbf{h}) \qquad \forall R \in SO(3). \quad (4)$$

*[Invariant prediction] The predictor $\widehat{y}_{\mathrm{inv}}(\mathbf{x},\mathbf{h}) := \widetilde{f}_\eta(A_\theta(\mathbf{x},\mathbf{h}),\mathbf{h})$ is rotation-invariant: $\widehat{y}_{\mathrm{inv}}(R\mathbf{x},\mathbf{h}) = \widehat{y}_{\mathrm{inv}}(\mathbf{x},\mathbf{h})$.*

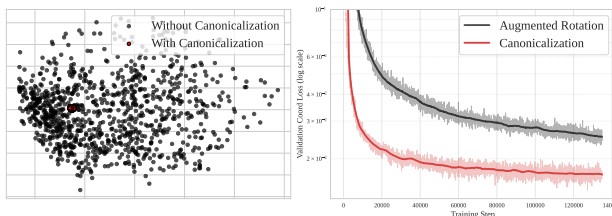

*Figure 2.* (**Left**) Embeddings of a single molecule (Aspirin) under 1,000 random rotations: without canonicalization (Repr.$\mathcal{C}$) vs. with canonicalization (Repr.$\mathcal{Z}$). (**Right**) Validation coordinate loss during pretraining for canonicalization (Repr.$\mathcal{Z}$) vs. rotation augmentation (Repr.$\mathcal{C}$). Experimental details are provided in Appendix F.

*[Equivariant outputs] For an output space $\mathcal{Y}$ equipped with a representation $\rho : SO(3) \to GL(\mathcal{Y})$ and a gauge selector $\widehat{R}_\theta$ satisfying $\widehat{R}_\theta(R\mathbf{x}, \mathbf{h}) = \widehat{R}_\theta(\mathbf{x}, \mathbf{h})R^\top$, the slice-and-lift predictor $\widehat{y}_{\mathrm{eq}}(\mathbf{x}, \mathbf{h}) := \rho(\widehat{R}_\theta(\mathbf{x}, \mathbf{h}))^{-1}\widetilde{f}_\eta(A_\theta(\mathbf{x}, \mathbf{h}), \mathbf{h})$ is $\rho$-equivariant: $\widehat{y}_{\mathrm{eq}}(R\mathbf{x}, \mathbf{h}) = \rho(R)\widehat{y}_{\mathrm{eq}}(\mathbf{x}, \mathbf{h})$.*

Proposition 3.3 shows that a single canonical backbone can serve both invariant prediction and equivariant outputs, with only a lightweight lifting operator for the latter.

This design also resolves the two limitations of rotation augmentation (Repr.$\mathcal{C}$). Recall that augmentation suffers from Prop. 3.2.(i) a structural gap between average-case training and worst-case behavior, and Prop. 3.2.(ii) prohibitive multiview cost to close this gap. Canonicalization eliminates the first issue by mapping all rotated versions of an input to the same canonical representative: since the backbone receives identical input regardless of the original orientation, worst-case variation is removed by construction, not merely controlled in expectation. Figure 2 (right) compares the validation coordinate loss between canonicalization and rotation augmentation during pretraining. Canonicalization achieves faster convergence and approximately 40% lower final reconstruction error, demonstrating the practical advantage of eliminating rotational variance at the input level. The second issue is also resolved because this guarantee requires only a *single* forward pass, completely avoiding the $K = \Omega(\varepsilon^{-3})$ multi-view cost. Canonicalization eliminates this burden: the backbone always receives consistent canonical input, reducing the task to standard representation learning (Figure 2(left)). Formally, canonicalization achieves $\sup_R |\widehat{y}(Rx, h) - \widehat{y}(x, h)| \le L_f \delta$ (Proposition A.5), where $\delta$ is the learned canonicalization error. For further details and proofs, see Appendix A, B.

**Proposition 3.4** (Canonicalize-and-lift for equivariant generation). *Assume $p_t$ is $SO(3)$-invariant and let $s^*(x, t) = \nabla_x \log p_t(x)$. Under Assumption A.1, for any score field $s$, there exists a canonical-space field $\widetilde{s}$ such that $s_{\mathrm{canonical}}(x, t) := \widehat{R}_\theta(x, h)^\top \widetilde{s}(A_\theta(x, h), t)$ satisfies*

$$\mathcal{J}(s_{\mathrm{canonical}}) \le \mathcal{J}(s), \tag{5}$$

*where $\mathcal{J}(s) := \mathbb{E}_t \mathbb{E}_{x \sim p_t}[\|s(x, t) - s^*(x, t)\|_F^2]$.*

Proposition 3.4 shows that restricting to canonical-space score models does not sacrifice optimality. Consequently, SE(3)-equivariant layers are not required throughout the backbone; equivariance is enforced at the interface via canonicalization and lifting, avoiding the per-layer overhead in iterative denoising. When evaluation is defined on rotation orbits, two-stage generation (canonical sampling followed by uniform lifting) achieves orbit-level fidelity bounded by the canonical modeling error plus an $O(\delta_A)$ alignment term (Appendix C).

### 3.2. THEMOL Framework

#### 3.2.1. PRE-TRAINING

In this section, we introduce the THEMOL framework, which leverages the proposed representation method to pretrain on large-scale datasets (Refer to Appendix I).

**Architecture.** The pre-training architecture consists of two components: (i) a canonicalization operator that identifies a canonical rotation from an arbitrarily rotated molecular input, and (ii) an backbone that takes the canonically rotated molecule as its input. After pre-training, we reuse the latent embeddings produced by the learned autoencoder for downstream fine-tuning. Below, we describe the designs of the canonicalization operator and the autoencoder.

Canonicalization Operator. To identify a canonical orientation for each molecular geometry, we introduce a learned *canonicalization operator* $\widehat{R}_\theta : \mathbb{R}^{N \times 3} \times \mathcal{H} \to SO(3)$. Following established practice on stable parameterizations of 3D rotations, we implement $\widehat{R}_\theta$ via an SVD-projection map. Given the centered coordinates $x^c$ and atom-level features $h$, the operator predicts an unconstrained matrix $\mathbf{M}_\theta(x, h) \in \mathbb{R}^{3 \times 3}$ and converts it to a proper rotation via this SVD-based projection:

$$\widehat{\mathbf{R}}_\theta(\mathbf{x}, \mathbf{h}) = \mathrm{SVD}^+(\mathbf{M}_\theta(\mathbf{x}, \mathbf{h})), \text{where} \tag{6}$$
$$\mathbf{M} = U\Sigma V^\top, \mathrm{SVD}^+(\mathbf{M}) = U\mathrm{diag}(1, 1, \det(UV^\top))V^\top$$

This construction projects $\mathbf{M}_\theta(\mathbf{x}, \mathbf{h})$ onto the nearest element of $SO(3)$ in the Frobenius norm, i.e., it solves an orthogonal Procrustes projection. We then define the canonicalization map $A_\theta(\mathbf{x}, \mathbf{h}) = \widehat{\mathbf{R}}_\theta(\mathbf{x}, \mathbf{h})\,\mathbf{x}^c$. Rather than requiring $\widehat{R}_\theta$ itself to be rotation-invariant, we enforce a consistency constraint so that the *canonicalized* coordinates are invariant:

$$\widehat{\mathbf{R}}_\theta(R\mathbf{x}, \mathbf{h}) = \widehat{\mathbf{R}}_\theta(\mathbf{x}, \mathbf{h})\,R^\top \to A_\theta(R\mathbf{x}, \mathbf{h}) = A_\theta(\mathbf{x}, \mathbf{h}).$$

In practice, we impose this constraint with a regularizer (e.g., in canonical space) $\mathcal{L}_{\mathrm{gc}} = \mathbb{E}_{R \sim \mu}[\|A_\theta(R\mathbf{x}, \mathbf{h}) - A_\theta(\mathbf{x}, \mathbf{h})\|_F^2]$. The resulting canonicalized geometry $A_\theta(\mathbf{x}, \mathbf{h})$ is used as the sole geometric input to the autoencoder backbone during pre-training.

Because a ground-truth canonical rotation is not uniquely definable, we do not supervise $\widehat{\mathbf{R}}_\theta$ to regress a predetermined correct orientation. Instead, $\widehat{\mathbf{R}}_\theta$ is learned in an unsupervised manner as a data-driven gauge that, under the autoencoder objective, jointly minimizes reconstruction error and enforces gauge consistency. Consequently, the resulting canonicalization is optimized to yield a latent space that best preserves information for faithful reconstruction (Appendix G.5).

Setup. Let $\mathbf{x} = (\mathbf{x}_1, ..., \mathbf{x}_N) \in \mathbb{R}^{N \times 3}$ be atomic coordinates, $\mathbf{h} = (\mathbf{h}_1, ..., \mathbf{h}_N) \in \mathbb{R}^{N \times d}$ be atom tokens. We define coordinates are centered as $\mathbf{x}^c = \mathbf{x} - \frac{1}{N} \sum_{i=1}^{N} \mathbf{x}_i$. We form token-wise initial embeddings:

$$\mathbf{X}^{\text{Init}} = \text{Linear}(\mathbf{x}^c) + \text{Linear}(\mathbf{h}), \ \mathbf{X}^{\text{Init}} \in \mathbb{R}^{N \times H} \quad (7)$$

, and we compute an initial pairwise attention bias $\mathbf{B}^{(0)} \in \mathbb{R}^{N \times N \times H}$ from pairwise distances $\mathbf{D}_{ij} = \|\mathbf{x}_i^c - \mathbf{x}_j^c\|_2$ and edge types $\mathbf{e}_{ij}$ via Gaussian basis features:

$$\mathbf{B}^{(0)} = \text{Linear}\big(\text{GBF}(\mathbf{D}_{ij}, \mathbf{e}_{ij})\big) \in \mathbb{R}^{N \times N \times H}, \quad (8)$$

where GBF expands distances into $K$ Gaussian bases. Given token embeddings $\mathbf{X}^{(\ell)} = [\mathbf{X}_1^{(\ell)}, \ldots, \mathbf{X}_N^{(\ell)}] \in \mathbb{R}^{N \times H}$ and current pair bias $\mathbf{B}^{(\ell)} \in \mathbb{R}^{N \times N \times H}$, a Transformer block performs standard multi-head self-attention with additive pair bias:

$$\mathbf{K}_i^{\ell,h} = W_K^{\ell,h} \mathbf{X}_i^\ell, \mathbf{V}_j^{\ell,h} = W_V^{\ell,h} \mathbf{X}_j^\ell, \mathbf{Q}_i^{\ell,h} = W_Q^{\ell,h} \mathbf{X}_i^\ell$$

$$\mathbf{A}_{ij}^{\ell,h} = \text{Softmax}_j \left( \frac{\langle \mathbf{Q}_i^{\ell,h}, \mathbf{K}_j^{\ell,h} \rangle}{\sqrt{d_h}} + \mathbf{B}_{ij}^{\ell,h} \right)$$

$$\widetilde{\mathbf{X}}_i^\ell = \sum_{h=1}^{H} W_A^{\ell,h} \sum_{j=1}^{N} \mathbf{A}_{ij}^{\ell,h} \mathbf{V}_j^{\ell,h}, \ \ \mathbf{X}_i^{\ell+1} = \mathbf{X}_i^\ell + \widetilde{\mathbf{X}}_i^\ell. \quad (9)$$

Crucially, we couple token and pair updates by feeding the attention weights back as the next-layer pair state:

$$\mathbf{X}^{(\ell+1)} := \boldsymbol{\zeta}^\ell(\mathbf{X}^{(\ell)}, \mathbf{B}^{(\ell)}) \in \mathbb{R}^{N \times H}. \quad (10)$$

Backbone. We instantiate the above stack with a shallow encoder $\boldsymbol{\zeta}_{\text{rot}}$ to predict a canonical rotation. Let $\mathbf{X}^{\text{rot}} = \boldsymbol{\zeta}_{\text{rot}}(\mathbf{X}^{\text{Init}}, \mathbf{B}^{\text{Init}})$ and we map it to an unconstrained matrix and project it to $SO(3)$ (Here, $\mathbf{X}$ is obtained from Eq.7):

$$\mathbf{M}_\theta(\mathbf{X}) = \text{Reshape}\left( \text{FFN}_{\text{rot}}\left( \frac{1}{N} \sum_{i=1}^{N} \mathbf{X}_i^{\text{rot}} \right) \right) \in \mathbb{R}^{3 \times 3},$$

$$\widehat{\mathbf{R}}_\theta(\mathbf{X}) = \text{Proj}_{SO(3)}(\mathbf{M}_\theta(\mathbf{X})). \quad (11)$$

The canonically rotated coordinates are then $\mathbf{X}^\star = \mathbf{X}^c \widehat{\mathbf{R}}_\theta(\mathbf{X})$. Using $\mathbf{X}^\star$, we repeat the same embedding and bias construction and run the main encoder stack:

$\mathbf{X}_{\text{Enc}} = \boldsymbol{\zeta}_{\text{Enc}}(\mathbf{X}^\star)$. The encoder outputs per-token Gaussian parameters, from which we sample the latent representation: $\mathbf{X}_{\mathbf{z}} \sim \mathcal{N}(\boldsymbol{\mu}, \boldsymbol{\sigma}^2)$, where $\boldsymbol{\mu} = \text{Linear}_\mu(\mathbf{X}_{\text{Enc}})$, $\log \boldsymbol{\sigma}^2 = \text{Linear}_\sigma(\mathbf{X}_{\text{Enc}})$.

In the decoding process, starting with the latent variable sampled from the encoding, we employ $\boldsymbol{\zeta}_{\text{Dec}}$, which shares an identical architecture, to extract $\mathbf{X}^{\text{Final}}$, the representation immediately preceding the prediction of the input molecule's various properties.

$$\mathbf{X}_{\text{Final}} = \boldsymbol{\zeta}_{\text{Dec}}(\mathbf{X}_{\mathbf{z}}, \mathbf{B}_{\mathbf{z}}), \text{where } \mathbf{B}_{\mathbf{z}} = \mathbf{0} \in \mathbb{R}^{N \times N \times H}$$

**Objective.** The backbone is pretrained to reconstruct 3D coordinates, atom types, and bond types, along with auxiliary molecular properties. The loss comprises reconstruction terms, a KL divergence for latent regularization, and a canonical consistency term; details of $\mathcal{L}_{\text{Coord}}$ are in Appendix G.5.

$$\mathcal{L}_{\text{Pre}} = \mathcal{L}_{\text{Coord}} + \mathcal{L}_{\text{Type}} + \mathcal{L}_{\text{Bond}} + \mathcal{L}_{\text{KLD}} + \mathcal{L}_{\text{gc}} \quad (12)$$

### 3.2.2. FINE-TUNING

For fine-tuning, the representation $\mathbf{X}^{\text{Final}}$ is extracted using the pretraining backbone and subsequently employed to train the model across five benchmarks within three categories of downstream tasks.

**Property Prediction.** Diverging from the Uni-Mol (Zhou et al.), we do not utilize the [CLS] token during the fine-tuning phase for property prediction. This is because, in contrast to NLP tasks, the 3D coordinates corresponding to a [CLS] token are ill-defined within 3D molecular representations. Instead, we apply a linear head to the entire $\mathbf{X}_{\text{Final}}$ embedding to predict the target properties.

**Generation** We learn a generative model directly on the pretrained representation space $\mathbf{X}_{\mathbf{z}} \in \mathbb{R}^{N \times H}$, treating the empirical distribution of encoded training molecules as our target $q_1$. We adopt *optimal-transport conditional flow matching* (OT-CFM) (Tong et al., 2024) with a clean-target prediction parameterization (Yim et al., 2023), which yields low-variance training objectives and straighter flows that require fewer integration steps during inference.

Let $q_0 = \mathcal{N}(\mathbf{0}, \mathbf{I})$ be an isotropic Gaussian prior over $\mathbb{R}^{N \times H}$. Given minibatches $\{\mathbf{X}_1^{(i)}\}_{i=1}^{B} \sim q_1$ and $\{\mathbf{X}_0^{(i)}\}_{i=1}^{B} \sim q_0$, we compute the optimal transport plan:

$$\Pi^\star \in \arg\min_{\Pi \in \mathcal{U}} \sum_{i,j} \Pi_{ij} \left\| \mathbf{X}_0^{(i)} - \mathbf{X}_1^{(j)} \right\|_{F,\Omega}^2, \quad (13)$$

where $\mathcal{U}$ denotes the set of doubly-stochastic matrices and $\Omega$ indexes non-padding atoms. Pairs $(\mathbf{X}_0, \mathbf{X}_1)$ are then sampled according to $\Pi^\star$, biasing the coupling toward short

displacements and reducing the variance of the regression target. For each paired sample, we construct the linear interpolation: $\mathbf{X}_t = (1-t)\mathbf{X}_0 + t\mathbf{X}_1, \qquad t \sim \mathcal{U}[0, 1-\varepsilon]$.

Rather than directly regressing the velocity field, we train the network $f_\phi$ to predict the clean target $\mathbf{X}_1$ from the noised input and the velocity field $\mathbf{v}_\phi$ is then reconstructed as:

$$\widehat{\mathbf{X}}_1 = f_\phi(t, \mathbf{X}_t), \quad \mathbf{v}_\phi(t, \mathbf{X}_t) = \frac{\widehat{\mathbf{X}}_1 - \mathbf{X}_t}{1-t}. \qquad (14)$$

This reparameterization is equivalent to velocity regression but avoids numerical instabilities near $t \approx 1$ where the conditional velocity can diverge. And, the flow matching loss is:

$$\mathcal{L}_{\mathrm{FM}} = \mathbb{E}_{(\mathbf{X}_0,\mathbf{X}_1)\sim\Pi^\star,t}\left[ w(t)\,\|f_\phi(t, \mathbf{X}_t) - \mathbf{X}_1\|_{F,\Omega}^2 \right],$$

where $w(t) = \min\{(1-t)^{-2}, w_{\max}\}$ with $w_{\max} = \varepsilon^{-2}$ prevents numerical instability near $t \approx 1$.

To generate a novel molecule, we sample $\mathbf{X}_0 \sim q_0$ and integrate the ODE:

$$\frac{d\mathbf{X}_t}{dt} = v_\phi(t, \mathbf{X}_t) = \frac{f_\phi(t, \mathbf{X}_t) - \mathbf{X}_t}{1-t}, \; t \in [0, 1]. \quad (15)$$

The OT-induced straighter trajectories allow accurate integration with substantially fewer steps than diffusion-based alternatives.

The terminal representation $\widetilde{\mathbf{X}}_{\mathbf{z}} := \mathbf{X}_1$ is decoded into molecular structure via the frozen pretrained heads: $\widehat{\mathbf{x}}^\star = \mathrm{Linear}_{\mathrm{coord}}(\boldsymbol{\zeta}_{\mathrm{Dec}}(\widetilde{\mathbf{X}}_{\mathbf{z}}))$, $\widehat{\mathbf{h}} = \mathrm{Linear}_{\mathrm{type}}(\boldsymbol{\zeta}_{\mathrm{Dec}}(\widetilde{\mathbf{X}}_{\mathbf{z}}))$. The decoded coordinates $\widehat{\mathbf{x}}^\star \in \mathbb{R}^{N\times3}$ lie in the canonical frame learned during pretraining. For applications requiring arbitrary orientations, we apply *uniform lifting*: sample $R \sim \mu$ from the Haar measure on $SO(3)$ and return $\widehat{\mathbf{x}} = \widehat{\mathbf{x}}^\star R$. This preserves the orbit-level distribution while restoring rotational diversity.

For further details on generation, see Appendix J.

**Optimization.** We optimize the pretrained flow-based generator to produce molecules with high binding affinity to a target protein pocket $\mathcal{T}$, without retraining the flow model. Let $\Phi_\phi$ be the frozen flow map and $\boldsymbol{\zeta}_{\mathrm{Dec}}$ be the frozen decoder. We seek an initial-state distribution $p_\psi$ that maximizes the expected score:

$$\max_\psi \; J(\psi) := \mathbb{E}_{\mathbf{X}_0\sim p_\psi}\Big[\mathcal{S}\big(\boldsymbol{\zeta}_{\mathrm{Dec}}(\Phi_\phi(\mathbf{X}_0)); \mathcal{T}\big)\Big], \quad (16)$$

where the scoring function $\mathcal{S}(\mathcal{M}; \mathcal{T}) = -E_{\mathrm{bind}}(\mathcal{M}; \mathcal{T}) + \lambda_{\mathrm{SA}} \cdot S_{\mathrm{SA}}(\mathcal{M})$ combines binding energy from UniDock and synthetic accessibility.

Since $\mathcal{S}$ involves non-differentiable operations (docking, discrete decoding), we employ derivative-free optimization inspired by EvoSBDD (Reidenbach, 2024). We

parameterize $p_\psi$ as a Gaussian $\mathcal{N}(\mathbf{m}, \sigma^2\mathbf{C})$ and optimize $\psi = (\mathbf{m}, \sigma, \mathbf{C})$ using LRA-CMA-ES (Nomura et al., 2025), an evolution strategy that adapts its covariance matrix to the local landscape and adjusts learning rates based on the signal-to-noise ratio of updates.

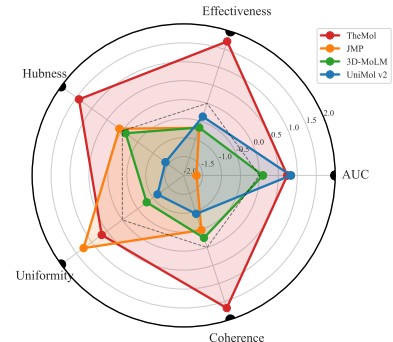

*Figure 3.* Radar chart summarizing ROC-AUC, Coherence, Effective Dimension, Hubness, and Uniformity for each model.

At each iteration, we sample $\lambda$ initial states from the current distribution, generate molecules via the frozen flow and decoder, evaluate their fitness, and update the distribution parameters toward high-scoring regions. After $T$ iterations, we sample from the optimized distribution $\mathcal{N}(\mathbf{m}^{(T)}, (\sigma^{(T)})^2\mathbf{C}^{(T)})$ and return top-scoring molecules. For further details on optimization, refer to the Appendix K.

### 3.3. How Should We Evaluate Molecular Representations?

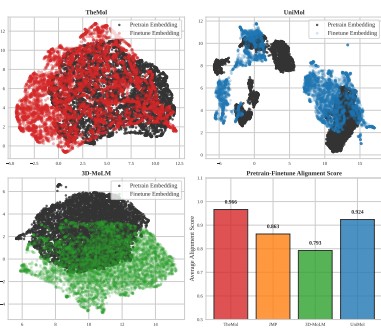

*Figure 4.* Alignment scores (bar chart) and scatter plots visualizing the distribution shift between pre-training sample embeddings and pre-trained embeddings.

Existing representation learning models, including pretrained foundation models, have predominantly demonstrated their effectiveness through benchmark performance alone. However, benchmark performance can be influenced by factors beyond the intrinsic quality of the learned representations, such as task-specific fine-tuning strategies, hyperparameter optimization, and dataset characteristics. We argue that a comprehensive evaluation of 3D molecular representation models should assess not only downstream task performance but also the quality of the learned embedding space itself.

To this end, we propose six evaluation metrics that directly probe different aspects of representation quality, independent of specific downstream tasks. All evaluations are conducted on molecules derived from the PDBbind 2020 database (Liu et al., 2017). The following provides a brief overview of the six proposed evaluation metrics. Detailed

*Table 1.* The overall results on 9 molecule classification datasets from the MoleculeNet (Wu et al., 2018). We report ROC-AUC score (higher is better) under scaffold splitting. The best result is highlighted in gray and the second-best is underlined. * denotes the mean calculated on reported tasks only.

| Datasets | BACE (↑) | BBBP (↑) | Tox21 (↑) | SIDER (↑) | HIV (↑) | MUV (↑) | PCBA (↑) | ClinTox (↑) | ToxCast (↑) | Mean (↑) |
|---|---|---|---|---|---|---|---|---|---|---|
| PretrainGNN | 84.5 | 72.6 | 78.1 | 62.7 | 79.9 | 81.3 | 86.0 | 72.6 | 65.7 | 75.93 |
| GROVER | 82.6 | 70.0 | 74.3 | 64.8 | 62.5 | 62.5 | 76.5 | 81.2 | 65.4 | 71.09 |
| MolCLR | 82.4 | 72.2 | 75.0 | 58.9 | 78.1 | 79.6 | - | 91.2 | 69.2 | 75.83* |
| MoleBLEND | 83.7 | 73.0 | 77.8 | 64.9 | 79.0 | 77.2 | - | 87.6 | 66.1 | 76.16* |
| Uni-Mol | 83.2 | 71.5 | 78.9 | 57.7 | 78.6 | 72.6 | 88.1 | 84.1 | 69.1 | 75.98 |
| Mol-AE | 84.1 | 72.0 | 80.0 | 67.0 | 80.6 | 81.6 | 88.9 | 87.8 | 69.6 | 79.04 |
| UniCorn | 85.8 | 74.2 | 79.3 | 64.0 | - | 82.6 | - | 92.1 | 69.4 | 78.0* |
| **THEMOL** | **89.9**$_{\pm0.7}$ | **91.2**$_{\pm0.5}$ | **81.3**$_{\pm0.4}$ | 64.3$_{\pm0.8}$ | **85.0**$_{\pm0.6}$ | 78.1$_{\pm1.2}$ | 83.7$_{\pm0.3}$ | **98.9**$_{\pm0.4}$ | **69.7**$_{\pm0.5}$ | **81.65** |

*Table 2.* 3D molecule generation results on GEOM-DRUG under two evaluation protocols. **(Left)** v1 setting evaluates Atom Stability and molecular Validity. **(Right)** v2 setting employs stricter metrics: PB-Valid, OOD Rings, and geometry relaxation measures.

| | **GEOM-DRUG** v1 (↑) | | | **GEOM-DRUG** v2 (↓) | | | |
|---|---|---|---|---|---|---|---|
| # Metrics | Atom Sta (%) | Valid (%) | # Metrics | PB-Valid (%) | OOD Rings (%) | Med. $\Delta E_{relax}$ | Med. $\Delta R_{relax}$ |
| Data | 86.5 | 99.9 | Data | 93.2$_{\pm0.01}$ | 0.05 | 0.00 | 0.00 |
| GeoLDM | 82.4 | 92.8 | EQGAT-Diff | 77.6 | 0.28 | 6.51 | 0.60 |
| GeoBFN | 78.9 | 93.1 | Megalodon | 86.6 | 0.17 | 3.17 | 0.41 |
| EquiFM | 84.1 | 98.9 | SemlaFlow | 88.5 | 0.0 | 31.92 | 0.24 |
| GOAT | 84.8 | 96.2 | FlowMol3 | 91.9 | 0.10 | 3.83 | 0.39 |
| **THEMOL** | **86.8**$_{\pm0.11}$ | **99.9** | **THEMOL** | **93.7**$_{\pm0.93}$ | 0.18$_{\pm0.01}$ | 4.97$_{\pm0.02}$ | **0.17**$_{\pm0.03}$ |

experimental protocols are provided in Appendix E.

- ROC-AUC measures molecular identity preservation by evaluating whether different conformers of the same molecule cluster together while remaining separated from different molecules.
- Coherence quantifies chemical semantic consistency via the ratio of inter-group to intra-group distance, where groups are defined by functional groups or molecular scaffolds.
- Effective Dimension measures intrinsic dimensionality of the embedding space via the participation ratio of covariance eigenvalues, where higher values indicate greater utilization of available dimensions.
- Hubness quantifies skewness of the $k$-nearest neighbor occurrence distribution, where lower values indicate more uniform neighbor relationships.
- Uniformity (Wang & Isola, 2020) measures how evenly embeddings are distributed on the hypersphere, where lower values indicate avoidance of representation collapse.
- Alignment Score evaluates consistency between pretraining and finetuning distributions via principal subspace similarity, where higher values indicate better downstream transferability.

We compare our approach against 3D molecular foundation models: **Uni-Mol2** (Ji et al., 2024), **JMP** (Shoghi et al.), and **3D-MoLM** (Li et al., 2024a). Details on baseline models are provided in Appendix E.

## 4. Experiments

### 4.1. Representation Quality Experiment

As shown in the Figure 5, Our evaluation reveals systematic trade-offs that validate the theoretical analysis in Section 3.1. A fundamental trade-off emerges between geometric regularity and chemical semantics: UniMol achieves the best geometric properties (lowest Hubness and Uniformity) but the worst Coherence, while THEMOL shows the opposite pattern with the highest Coherence. This confirms that distance-only invariant features (REPR.$\mathcal{A}$) sacrifice chemical semantic information for geometric regularity, as predicted by our theoretical framework. THEMOL's highest Effective Dimension validates that canonicalization preserves richer information content by processing coordinates in canonical space rather than reducing them to invariant features. Furthermore, THEMOL achieves the highest Alignment Score, indicating that 96.6% of downstream task variations are already captured during pretraining—enabling confident attribution of downstream performance to representation quality rather than finetuning adaptation. Furthermore, the gap between pre-training and fine-tuning embeddings for each model, as shown in the scatter plots of Figure 4, further supports our interpretation of the alignment scores. The distributional gap between pretrain and finetune embeddings suggests that models with larger gaps require more substantial adaptation during finetuning to bridge this representational mismatch.

Notably, the divergence between Coherence and Alignment Score reveals that these metrics capture complementary aspects: semantic structure versus statistical structure. THEMOL uniquely achieves top performance on both, demonstrating its capacity to learn representations that are simultaneously chemically meaningful and statistically aligned with downstream tasks. These results collectively validate canonicalization as a principled approach to 3D molecular foundation modeling that addresses the theoretical limitations of existing representation paradigms. For further interpretation of this experiment, refer to Appendix E.6.

*Table 3.* Performance of THEMOL on the TDC Benchmark and its comparison with the TDC Leaderboard and MiniMol. The best results are **bolded**, underlined, and highlighted in gray. The mean rank is computed using only the tasks in the TDC leaderboard.

| Methods | TDC Dataset | | | TDC Leaderboard (January 2026) | | MiniMol | | THEMOL | |
|---|---|---|---|---|---|---|---|---|---|
| | Dataset | Metric | Size | SOTA Model | SOTA Result | Result | Rank | Result | Rank |
| Absorption | Caco2 | MAE(↓) | 906 | CaliciBoost | $0.256_{\pm0.006}$ | $0.350_{\pm0.018}$ | 16 | $0.291_{\pm0.009}$ | 8 |
| | HIA | AUROC(↑) | 578 | MiniMol | $0.993_{\pm0.005}$ | $0.993_{\pm0.005}$ | 2 | $\mathbf{0.996}_{\pm0.003}$ | **1** |
| | Pgp | AUROC(↑) | 1212 | MapLight + GNN | $0.938_{\pm0.002}$ | - | - | $\mathbf{0.939}_{\pm0.004}$ | **1** |
| | Bioavailability | AUROC(↑) | 640 | MiniMol | $0.942_{\pm0.002}$ | $\underline{\mathbf{0.942}}_{\pm0.002}$ | **1** | $0.706_{\pm0.015}$ | - |
| | Lipophilicity | MAE(↓) | 4200 | MiniMol | $0.456_{\pm0.008}$ | $\underline{\mathbf{0.456}}_{\pm0.008}$ | **1** | $\mathbf{0.456}_{\pm0.007}$ | **1** |
| | Solubility | MAE(↓) | 9982 | MiniMol | $0.741_{\pm0.013}$ | $0.741_{\pm0.013}$ | 2 | $\mathbf{0.722}_{\pm0.010}$ | **1** |
| Distribution | BBB | AUROC(↑) | 1975 | MiniMol | $0.924_{\pm0.003}$ | $0.924_{\pm0.003}$ | 2 | $\mathbf{0.927}_{\pm0.004}$ | **1** |
| | PPBR | MAE(↓) | 1797 | Gradient Boost | $7.440_{\pm0.024}$ | $7.696_{\pm0.125}$ | 5 | $7.461_{\pm0.078}$ | 2 |
| | VDss | Spearman(↑) | 1130 | MapLight + GNN | $0.713_{\pm0.007}$ | $0.535_{\pm0.027}$ | 10 | $0.704_{\pm0.016}$ | 3 |
| Metabolism | CYP2D6 inhibition | AUPRC(↑) | 13130 | MapLight + GNN | $0.790_{\pm0.001}$ | $0.719_{\pm0.004}$ | 5 | $0.703_{\pm0.005}$ | 7 |
| | CYP3A4 inhibition | AUPRC(↑) | 12328 | MapLight + GNN | $0.916_{\pm0.000}$ | $0.877_{\pm0.001}$ | 7 | $0.880_{\pm0.004}$ | 6 |
| | CYP2C9 inhibition | AUPRC(↑) | 12092 | MapLight + GNN | $0.859_{\pm0.001}$ | $0.823_{\pm0.006}$ | 4 | $0.798_{\pm0.006}$ | 5 |
| | CYP2D6 substrate | AUPRC(↑) | 664 | ContextPred | $0.736_{\pm0.024}$ | $0.695_{\pm0.032}$ | 8 | $\mathbf{0.775}_{\pm0.021}$ | **1** |
| | CYP3A4 substrate | AUROC(↑) | 667 | CFA | $0.667_{\pm0.019}$ | $0.663_{\pm0.008}$ | 3 | $\mathbf{0.787}_{\pm0.014}$ | **1** |
| | CYP2C9 substrate | AUPRC(↑) | 666 | MiniMol | $0.474_{\pm0.025}$ | $0.474_{\pm0.025}$ | 2 | $\mathbf{0.542}_{\pm0.019}$ | **1** |
| Excretion | Half life | Spearman(↑) | 667 | CFA | $0.576_{\pm0.025}$ | $0.495_{\pm0.042}$ | 7 | $0.559_{\pm0.026}$ | 3 |
| | Clearance mircosome | Spearman(↑) | 1102 | MapLight + GNN | $0.630_{\pm0.010}$ | $0.628_{\pm0.005}$ | 3 | $\mathbf{0.692}_{\pm0.011}$ | **1** |
| | Clearance hepatocyte | Spearman(↑) | 1020 | CFA | $0.536_{\pm0.020}$ | $0.446_{\pm0.029}$ | 6 | $0.479_{\pm0.023}$ | 3 |
| Toxicity | hERG | AUROC(↑) | 648 | MapLight + GNN | $0.880_{\pm0.002}$ | $0.846_{\pm0.016}$ | 7 | $0.863_{\pm0.013}$ | 5 |
| | Ames | AUROC(↑) | 7255 | ZairaChem | $0.871_{\pm0.002}$ | $0.849_{\pm0.004}$ | 7 | $0.835_{\pm0.005}$ | 13 |
| | DILI | AUROC(↑) | 475 | MiniMol | $0.956_{\pm0.006}$ | $\underline{\mathbf{0.956}}_{\pm0.006}$ | **1** | $0.933_{\pm0.010}$ | 2 |
| | LD50 | MAE(↓) | 7385 | BaseBoosting | $0.552_{\pm0.009}$ | $0.585_{\pm0.008}$ | 3 | $0.608_{\pm0.009}$ | 6 |
| | | | | | TDC Leaderboard Mean Rank: | | 4.9 | | **3.4** |

*Table 4.* Summary of binding affinity and molecular properties of reference molecules and molecules generated by THEMOL and baselines. Top 2 results are **bolded** and underlined, respectively.

| Methods | | SMINA (↓) | | Vina Dock (↓) | | SA (↑) | | Diversity (↑) | |
|---|---|---|---|---|---|---|---|---|---|
| | | Avg. | Med. | Avg. | Med. | Avg. | Med. | Avg. | Med. |
| | Reference | -6.37 | -7.92 | -6.36 | -7.45 | 0.73 | 0.74 | - | - |
| Gen. | TargetDiff | -8.51 | -8.56 | -7.80 | -7.91 | 0.58 | 0.58 | 0.72 | 0.71 |
| | DecompDiff | -8.33 | -8.27 | -8.39 | -8.43 | 0.61 | 0.60 | 0.68 | 0.68 |
| | MolCRAFT | -8.56 | **-8.41** | -7.59 | -7.31 | 0.68 | 0.67 | 0.70 | 0.73 |
| Gen. + Opt. | DecompOpt | -8.10 | -8.35 | -8.98 | **-9.01** | 0.65 | 0.65 | 0.60 | 0.61 |
| | TacoGFN | -7.46 | -7.15 | -7.74 | -7.35 | **0.79** | **0.80** | 0.56 | 0.56 |
| | ALIDiff | -8.47 | -8.21 | -8.90 | -8.81 | 0.57 | 0.56 | 0.73 | 0.71 |
| | **THEMOL** | **-8.63** | -8.36 | **-9.12** | -8.80 | 0.75 | 0.77 | **0.85** | **0.84** |

## 4.2. Benchmark Performance Experiment

We evaluate THEMOL across five benchmarks closely related to drug discovery, spanning property prediction, generation, and optimization. Details on the dataset, evaluation metrics, and baseline models are provided in Appendices I, I.7, and I.8, respectively.

**Molecular Property Prediction.** On MoleculeNet (Table 1), THEMOL achieves the highest mean ROC-AUC of 81.65, ranking first on six of nine datasets including BACE, BBBP, and ClinTox. On the TDC ADMET benchmark (Table 3), THEMOL attains a mean rank of 3.4 across 22 tasks, outperforming MiniMol with first-place results on 15 tasks spanning all ADMET categories.

**3D Molecule Generation.** On GEOM-DRUG (Table 2), THEMOL achieves 86.8% Atom Stability and 99.9% Validity under the v1 setting, matching training data quality. Under the stricter v2 evaluation, THEMOL attains the highest PB-Valid and lowest $\Delta R_{\text{relax}}$, indicating physically plausible geometries. While specialized equivariant models

show advantages on certain metrics, THEMOL maintains competitive generation quality within a unified framework.

**3D Molecule Optimization.** On CrossDocked2020 (Table 4), THEMOL achieves the strongest binding affinity with balanced molecular properties: competitive synthetic accessibility and the highest diversity. This balance contrasts with methods like TacoGFN that achieve higher SA at the cost of diversity.

Across all benchmarks, THEMOL exhibits consistently competitive performance, validating canonicalization as a versatile foundation for diverse molecular tasks. Additional ablation experiments on each representation paradigm are provided in the Appendix H.

## 5. Conclusion

We have presented THEMOL, a canonicalization-based framework that unifies invariant and equivariant tasks within a single 3D molecular representation model. By mapping molecules to a learned canonical pose before encoding, our approach overcomes the fundamental limitations. Experiments across property prediction, generation, and optimization demonstrate consistently competitive performance against task-specific methods, while our proposed evaluation criteria reveal that THEMOL achieves superior representation quality in terms of both chemical semantics and downstream transferability. Looking forward, extending this framework to larger molecular systems (e.g. Protein) and integrating multimodal information present promising directions for more general-purpose molecular foundation models.

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

# A. Canonicalization for Invariant Prediction: Quotient Factorization and Stability

## A.1. Setup

Let $\mathbf{x} \in \mathbb{R}^{N \times 3}$ be 3D coordinates and $\mathbf{h}$ be per-node features (e.g., atom types). We remove translations by centering

$$\mathbf{x}^{\mathrm{c}} := \mathbf{x} - \mathbf{1}\bar{\mathbf{x}}^{\top}, \qquad \bar{\mathbf{x}} := \frac{1}{N} \sum_{i=1}^{N} \mathbf{x}_i \in \mathbb{R}^3.$$

For $R \in \mathrm{SO}(3)$, the action on coordinates is $x \mapsto Rx$ (row-wise). We use a learned *rotation selector* $\widehat{R}_\theta(\mathbf{x}, \mathbf{h}) \in \mathrm{SO}(3)$ and define the *canonicalization map*

$$A_\theta(\mathbf{x}, \mathbf{h}) := \widehat{R}_\theta(\mathbf{x}, \mathbf{h})\, \mathbf{x}^{\mathrm{c}} \in \mathbb{R}^{N \times 3}. \tag{17}$$

A generic non-equivariant predictor in canonical space is

$$\widehat{y}(\mathbf{x}, \mathbf{h}) := \widetilde{f}_\eta\big(A_\theta(\mathbf{x}, \mathbf{h}), \mathbf{h}\big). \tag{18}$$

*Permutation symmetry is not assumed:* we construct the dataset using RDKit's canonical atom ordering, which fixes a deterministic node indexing.

A target $y(\mathbf{x}, \mathbf{h}) \in \mathbb{R}$ is *rotation-invariant* if $y(R\mathbf{x}, \mathbf{h}) = y(\mathbf{x}, \mathbf{h})$ for all $R \in \mathrm{SO}(3)$.

## A.2. Exact factorization through a canonical representative

**Assumption A.1** (Well-defined canonicalization on the data support). There exists a subset $\mathcal{S} \subseteq \mathbb{R}^{N \times 3} \times \mathcal{H}$ containing the data support such that:

(gauge consistency) $\quad \widehat{\mathbf{R}}_\theta(Rx, h) = \widehat{\mathbf{R}}_\theta(\mathbf{x}, \mathbf{h})R^{\top}, \quad \forall R \in \mathrm{SO}(3), \; (\mathbf{x}, \mathbf{h}) \in \mathcal{S},$

(orbit-separation) $\quad A_\theta(\mathbf{x}, \mathbf{h}) = A_\theta(\mathbf{x}', \mathbf{h}') \Rightarrow \mathbf{h} = \mathbf{h}'$ and $\exists R \in \mathrm{SO}(3)$ s.t. $\mathbf{x}' = R\mathbf{x}, \; \forall (\mathbf{x}, \mathbf{h}), (\mathbf{x}', \mathbf{h}') \in \mathcal{S}.$

Equivalently, $A_\theta(R\mathbf{x}, \mathbf{h}) = A_\theta(\mathbf{x}, \mathbf{h})$ and $A_\theta$ is injective on rotation orbits over $\mathcal{S}$.

Global orbit-separating sections may fail at symmetric/degenerate configurations (non-trivial stabilizers), leading to ambiguity or discontinuity. In practice this is mitigated by focusing on generic inputs (symmetries are measure-zero under perturbations) and by using deterministic node indexing (RDKit canonical ordering) so that $\widehat{R}_\theta(\mathbf{x}, \mathbf{h})$ can implement consistent tie-breaking on the data support.

**Proposition A.2** (Invariant targets factor through canonicalization). *Under Assumption A.1, for any rotation-invariant target $y$ there exists a function $\widetilde{y}$ such that*

$$y(\mathbf{x}, \mathbf{h}) = \widetilde{y}\big(A_\theta(\mathbf{x}, \mathbf{h}), \mathbf{h}\big) \qquad \text{for all } (\mathbf{x}, \mathbf{h}) \in \mathcal{S}. \tag{19}$$

*Proof.* Fix $(\mathbf{x}, \mathbf{h}) \in \mathcal{S}$ and set $u = A_\theta(\mathbf{x}, \mathbf{h})$. Gauge consistency implies $u$ is constant on the orbit $\{(R\mathbf{x}, \mathbf{h}) : R \in \mathrm{SO}(3)\}$, and invariance implies $y$ is constant on the same orbit. Orbit-separation ensures $(\mathbf{u}, \mathbf{h})$ identifies the orbit within $\mathcal{S}$, so defining $\widetilde{y}(\mathbf{u}, \mathbf{h}) := y(\mathbf{x}, \mathbf{h})$ is well-defined. $\qquad\square$

**Proposition A.3** (Universal approximation in canonical space). *Assume Assumption A.1. If $y$ is continuous and rotation-invariant on $\mathcal{S}$ and $\widetilde{f}_\eta(\cdot, \mathbf{h})$ is a universal approximator for continuous functions on the domain of $\big(A_\theta(\mathbf{x}, \mathbf{h}), \mathbf{h}\big)$ (restricted to $(\mathbf{x}, \mathbf{h}) \in \mathcal{S}$), then for any $\varepsilon > 0$ there exist parameters $\eta$ such that*

$$\sup_{(\mathbf{x}, \mathbf{h}) \in \mathcal{S}} \big| \widetilde{f}_\eta(A_\theta(\mathbf{x}, \mathbf{h}), \mathbf{h}) - y(\mathbf{x}, \mathbf{h}) \big| \leq \varepsilon. \tag{20}$$

By Proposition A.2, $y(\mathbf{x}, \mathbf{h}) = \widetilde{y}(A_\theta(\mathbf{x}, \mathbf{h}), \mathbf{h})$ for some continuous $\widetilde{y}$ on the induced domain. Universality yields $\eta$ approximating $\widetilde{y}$ uniformly to error $\varepsilon$.

## A.3. Approximate invariance and rotation-augmented risk stability

**Assumption A.4** (Alignment consistency and Lipschitz stability). Assume there exist $\delta \geq 0$, $L_f \geq 0$, and $L_\ell \geq 0$ such that for all $(x, h) \in \mathcal{S}$:

$$\sup_{R \in \mathrm{SO}(3)} \left\| A_\theta(R\mathbf{x}, \mathbf{h}) - A_\theta(\mathbf{x}, \mathbf{h}) \right\|_F \leq \delta,$$

$\widetilde{f}_\eta(\cdot, \mathbf{h})$ is $L_f$-Lipschitz in its first argument, and $\ell(\hat{y}, y)$ is $L_\ell$-Lipschitz in $\hat{y}$:

$$\left| \widetilde{f}_\eta(u_1, \mathbf{h}) - \widetilde{f}_\eta(u_2, \mathbf{h}) \right| \leq L_f \|u_1 - u_2\|_F, \qquad |\ell(\hat{y}_1, y) - \ell(\hat{y}_2, y)| \leq L_\ell |\hat{y}_1 - \hat{y}_2|.$$

**Proposition A.5** (Approximate invariance of canonical predictors). *Under Assumption A.4, the predictor equation 18 satisfies*

$$\sup_{R \in \mathrm{SO}(3)} |\widehat{y}(R\mathbf{x}, \mathbf{h}) - \widehat{y}(\mathbf{x}, \mathbf{h})| \leq L_f \delta \qquad \textit{for all } (\mathbf{x}, \mathbf{h}) \in \mathcal{S}.$$

*If moreover $y$ is rotation-invariant, then for each fixed $(\mathbf{x}, \mathbf{h}) \in \mathcal{S}$,*

$$\left| \mathbb{E}_{R \sim \mu}\left[ \ell(\widehat{y}(R\mathbf{x}, \mathbf{h}), y(\mathbf{x}, \mathbf{h})) \right] - \ell(\widehat{y}(\mathbf{x}, \mathbf{h}), y(\mathbf{x}, \mathbf{h})) \right| \leq L_\ell L_f \delta.$$

*Proof.* Lipschitzness gives $|\widehat{y}(R\mathbf{x}, \mathbf{h}) - \widehat{y}(\mathbf{x}, \mathbf{h})| \leq L_f \|A_\theta(R\mathbf{x}, \mathbf{h}) - A_\theta(\mathbf{x}, \mathbf{h})\|_F \leq L_f \delta$. For invariant $y$, apply $L_\ell$-Lipschitzness to the loss difference and average over $R$. $\square$

# B. Canonicalization for Equivariant Prediction

## B.1. $\rho$-equivariant targets

Let $\mathcal{Y}$ be an output vector space (e.g., $\mathbb{R}^3$ or $\mathbb{R}^{N \times 3}$) and $\rho : \mathrm{SO}(3) \to GL(\mathcal{Y})$ be a linear representation. A target $y(\mathbf{x}, \mathbf{h}) \in \mathcal{Y}$ is *$\rho$-equivariant* if

$$y(R\mathbf{x}, \mathbf{h}) = \rho(R)\, y(\mathbf{x}, \mathbf{h}) \qquad \forall R \in \mathrm{SO}(3). \tag{21}$$

## B.2. Exact Canonicalize-and-Lift factorization

**Proposition B.1** (Canonical factorization of $\rho$-equivariant maps). *Assume Assumption A.1. A function $y : \mathbb{R}^{N \times 3} \times \mathcal{H} \to \mathcal{Y}$ is $\rho$-equivariant on $\mathcal{S}$ if and only if there exists $\widetilde{y}$ such that*

$$y(\mathbf{x}, \mathbf{h}) = \rho(\widehat{\mathbf{R}}_\theta(\mathbf{x}, \mathbf{h}))^{-1}\, \widetilde{y}(A_\theta(\mathbf{x}, \mathbf{h}), \mathbf{h}), \qquad \forall (\mathbf{x}, \mathbf{h}) \in \mathcal{S}. \tag{22}$$

*Proof.* Assume $y$ is $\rho$-equivariant on $\mathcal{S}$. For any $(\mathbf{x}, \mathbf{h}) \in \mathcal{S}$, set $\mathbf{u} := A_\theta(\mathbf{x}, \mathbf{h})$ and define

$$\widetilde{y}(\mathbf{u}, \mathbf{h}) := \rho(\widehat{\mathbf{R}}_\theta(\mathbf{x}, \mathbf{h}))\, y(\mathbf{x}, \mathbf{h}). \tag{23}$$

To see that $\widetilde{y}$ is well-defined, let $(\mathbf{x}_1, \mathbf{h}), (\mathbf{x}_2, \mathbf{h}) \in \mathcal{S}$ satisfy $A_\theta(\mathbf{x}_1, \mathbf{h}) = A_\theta(\mathbf{x}_2, \mathbf{h}) = \mathbf{u}$. By orbit-separation there exists $R \in SO(3)$ such that $\mathbf{x}_2 = R\mathbf{x}_1$. Using $\rho$-equivariance, gauge consistency, and that $\rho$ is a representation, we have

$$y(\mathbf{x}_2, \mathbf{h}) = y(R\mathbf{x}_1, \mathbf{h}) = \rho(R)\, y(\mathbf{x}_1, \mathbf{h}),$$

$$\widehat{\mathbf{R}}_\theta(\mathbf{x}_2, \mathbf{h}) = \widehat{\mathbf{R}}_\theta(R\mathbf{x}_1, \mathbf{h}) = \widehat{\mathbf{R}}_\theta(x_1, h)R^\top,$$

$$\rho(\widehat{\mathbf{R}}_\theta(\mathbf{x}_2, \mathbf{h}))\, y(\mathbf{x}_2, \mathbf{h}) = \rho(\widehat{\mathbf{R}}_\theta(\mathbf{x}_1, \mathbf{h})R^\top)\, \rho(R)\, y(\mathbf{x}_1, \mathbf{h})$$
$$= \rho(\widehat{\mathbf{R}}_\theta(\mathbf{x}_1, \mathbf{h}))\, \rho(R^\top)\rho(R)\, y(\mathbf{x}_1, \mathbf{h}) = \rho(\widehat{\mathbf{R}}_\theta(\mathbf{x}_1, \mathbf{h}))\, y(\mathbf{x}_1, \mathbf{h}), \tag{24}$$

where $\rho(R^\top) = \rho(R)^{-1}$ for $R \in SO(3)$. Hence $\widetilde{y}(\mathbf{u}, \mathbf{h})$ does not depend on the choice of $(\mathbf{x}, \mathbf{h})$ with $\mathbf{u} = A_\theta(\mathbf{x}, \mathbf{h})$, and rearranging the definition yields $y(\mathbf{x}, \mathbf{h}) = \rho(\widehat{\mathbf{R}}_\theta(\mathbf{x}, \mathbf{h}))^{-1}\widetilde{y}(A_\theta(\mathbf{x}, \mathbf{h}), \mathbf{h})$ for all $(\mathbf{x}, \mathbf{h}) \in \mathcal{S}$.

Conversely, suppose there exists $\widetilde{y}$ such that $y(\mathbf{x}, \mathbf{h}) = \rho(\widehat{\mathbf{R}}_\theta(\mathbf{x}, \mathbf{h}))^{-1}\widetilde{y}(A_\theta(\mathbf{x}, \mathbf{h}), \mathbf{h})$ on $\mathcal{S}$. Then for any $R \in SO(3)$, gauge consistency and the homomorphism property of $\rho$ give

$$
\begin{aligned}
y(Rx, h) &= \rho(\widehat{\mathbf{R}}_\theta(R\mathbf{x}, \mathbf{h}))^{-1}\, \widetilde{y}(A_\theta(R\mathbf{x}, \mathbf{h}), \mathbf{h}) \\
&= \rho(\widehat{\mathbf{R}}_\theta(\mathbf{x}, \mathbf{h})R^\top)^{-1}\, \widetilde{y}(A_\theta(\mathbf{x}, \mathbf{h}), \mathbf{h}) \\
&= (\rho(\widehat{\mathbf{R}}_\theta(\mathbf{x}, \mathbf{h}))\rho(R^\top))^{-1}\, \widetilde{y}(A_\theta(\mathbf{x}, \mathbf{h}), \mathbf{h}) \\
&= \rho(R)\, \rho(\widehat{\mathbf{R}}_\theta(\mathbf{x}, \mathbf{h}))^{-1}\, \widetilde{y}(A_\theta(\mathbf{x}, \mathbf{h}), \mathbf{h}) = \rho(R)\, y(\mathbf{x}, \mathbf{h}),
\end{aligned}
$$

which shows that $y$ is $\rho$-equivariant on $\mathcal{S}$. $\qquad\square$

### B.3. Approximate equivariance bounds

Define the Canonicalize-and-Lift predictor

$$
\widehat{y}(\mathbf{x}, \mathbf{h}) := \rho(\widehat{R}_\theta(\mathbf{x}, \mathbf{h}))^{-1}\, \widetilde{f}_\eta(A_\theta(\mathbf{x}, \mathbf{h}), \mathbf{h}).
$$

**Assumption B.2** (Approximate canonicalization and Lipschitz stability). Assume there exist $\delta_A, \delta_R \geq 0$ such that for all $(x, h) \in \mathcal{S}$ and all $R \in \mathrm{SO}(3)$,

$$
\|A_\theta(R\mathbf{x}, \mathbf{h}) - A_\theta(x, h)\|_F \leq \delta_A, \qquad \|\widehat{R}_\theta(R\mathbf{x}, \mathbf{h}) - \widehat{R}_\theta(\mathbf{x}, \mathbf{h})R^\top\|_F \leq \delta_R.
$$

Assume $\widetilde{f}_\eta(\cdot, h)$ is $L_{\widetilde{f}}$-Lipschitz and bounded by $B_{\widetilde{f}}$:

$$
\|\widetilde{f}_\eta(u_1, \mathbf{h}) - \widetilde{f}_\eta(u_2, \mathbf{h})\| \leq L_{\widetilde{f}}\|u_1 - u_2\|_F, \qquad \|\widetilde{f}_\eta(u, \mathbf{h})\| \leq B_{\widetilde{f}}.
$$

Assume $\rho(Q)^{-1}$ is $L_\rho$-Lipschitz in operator norm w.r.t. $Q$:

$$
\|\rho(Q_1)^{-1} - \rho(Q_2)^{-1}\|_{\mathrm{op}} \leq L_\rho\|Q_1 - Q_2\|_F.
$$

**Proposition B.3** (Approximate equivariance of Canonicalize-and-Lift predictors). *Under Assumption B.2, for all $(x, h) \in \mathcal{S}$ and $R \in \mathrm{SO}(3)$,*

$$
\left\|\widehat{y}(R\mathbf{x}, \mathbf{h}) - \rho(R)\widehat{y}(\mathbf{x}, \mathbf{h})\right\| \ \leq\ L_{\widetilde{f}}\delta_A \ + \ L_\rho\, B_{\widetilde{f}}\, \delta_R.
$$

*Proof.* Add/subtract $\rho(\widehat{R}_\theta(R\mathbf{x}, \mathbf{h}))^{-1}\widetilde{f}_\eta(A_\theta(\mathbf{x}, \mathbf{h}), \mathbf{h})$. Bound (i) the $\widetilde{f}_\eta$ change by $L_{\widetilde{f}}\delta_A$ and (ii) the $\rho(\widehat{R}_\theta)^{-1}$ change by $L_\rho\delta_R$ times $\|\widetilde{f}_\eta\| \leq B_{\widetilde{f}}$, then use $\rho(\widehat{R}_\theta(\mathbf{x}, \mathbf{h})R^\top)^{-1} = \rho(R)\rho(\widehat{R}_\theta(\mathbf{x}, \mathbf{h}))^{-1}$. $\qquad\square$

## C. Generative Modeling in Canonical Coordinates

### C.1. Score matching and equivariant projection

Let $p_t$ be a time-marginal of an isotropic noising process on $\mathbb{R}^{N \times 3}$ and

$$
\mathbf{s}^*(\mathbf{x}, \mathbf{t}) := \nabla_\mathbf{x} \log p_t(\mathbf{x}).
$$

For a score model $s(x, t)$, define the population MSE

$$
\mathcal{J}(s) := \mathbb{E}_t \, \mathbb{E}_{\mathbf{x} \sim p_t}\big[\|s(x, t) - s^*(\mathbf{x}, t)\|_F^2\big].
$$

**Assumption C.1** (Rotation-invariant marginals). Assume $p_t$ is $\mathrm{SO}(3)$-invariant: $p_t(R\mathbf{x}) = p_t(\mathbf{x})$ for all $R \in \mathrm{SO}(3)$. Then $\mathbf{s}^*$ is $\mathrm{SO}(3)$-equivariant: $\mathbf{s}^*(R\mathbf{x}, t) = R\, s^*(\mathbf{x}, t)$ (Lu et al.)

**Definition C.2** (Equivariant projection). For a vector field $\mathbf{s}(\cdot, t)$ define

$$
(\mathcal{P}_{\mathrm{eq}} \cdot \mathbf{s})(\mathbf{x}, t) := \mathbb{E}_{R \sim \mu}\big[R\,\mathbf{s}(R^\top\mathbf{x}, t)\big], \tag{25}
$$

where $\mu$ is the Haar measure on $\mathrm{SO}(3)$ and $\mathcal{P}_{\mathrm{eq}}$ denotes the equivariant projection operator that maps an arbitrary vector field to an $SO(3)$-equivariant one by Haar-averaging over rotations.

**Proposition C.3** (WLOG canonical coordinates for score matching). *Assume (i) $p_t$ is SO(3)-invariant and hence $s^*$ is SO(3)-equivariant, and (ii) Assumption A.1 holds. Then for any score field $s$, there exists a canonical-space field $\widetilde{s}(\cdot, t)$ such that*

$$s_{\text{canonical}}(\mathbf{x}, t) := \widehat{R}_\theta(\mathbf{x}, \mathbf{h})^\top \widetilde{s}(A_\theta(\mathbf{x}, \mathbf{h}), t) \tag{26}$$

*satisfies*

$$\mathcal{J}(s_{\text{canonical}}) \leq \mathcal{J}(s). \tag{27}$$

*Proof.* We construct $s_{\text{canonical}}$ via the equivariant projection. Let $\Delta := s - s^*$. Since $s^*$ is SO(3)-equivariant, $\mathcal{P}_{\text{eq}} s^* = s^*$, and thus $\mathcal{P}_{\text{eq}} s - s^* = \mathcal{P}_{\text{eq}} \Delta$. By Jensen's inequality and orthogonality invariance of $\|\cdot\|_F$,

$$\|\mathcal{P}_{\text{eq}} \Delta(\mathbf{x}, t)\|_F^2 = \left\|\mathbb{E}_R[R\,\Delta(R^\top \mathbf{x}, t)]\right\|_F^2 \leq \mathbb{E}_R \|\Delta(R^\top \mathbf{x}, t)\|_F^2.$$

Taking $\mathbb{E}_t \mathbb{E}_{\mathbf{x} \sim p_t}$ and using SO(3)-invariance of $p_t$ yields $\mathcal{J}(\mathcal{P}_{\text{eq}} s) \leq \mathcal{J}(s)$.

Now, $\mathcal{P}_{\text{eq}} s$ is SO(3)-equivariant by construction. Under Assumption A.1, any SO(3)-equivariant score admits the canonicalize-and-lift form (cf. Proposition B.1):

$$(\mathcal{P}_{\text{eq}} s)(\mathbf{x}, t) = \widehat{R}_\theta(\mathbf{x}, \mathbf{h})^\top \widetilde{s}(A_\theta(\mathbf{x}, \mathbf{h}), t)$$

for some $\widetilde{s}$. Setting $s_{\text{canonical}} := \mathcal{P}_{\text{eq}} s$ completes the proof. □

## C.2. Quotient-space evaluation beyond the ideal section

Let $G = \text{SO}(3)$, $\varphi(\mathbf{x}) := [\mathbf{x}]_G$, and $d_{\text{orb}}(\varphi(\mathbf{x}), \varphi(y)) := \min_{R \in \text{SO}(3)} \|R\mathbf{x} - y\|_F$. Let $W_1^{\text{orb}}$ be the 1-Wasserstein distance induced by $d_{\text{orb}}$.

**Assumption C.4** (Approximate orbit consistency). On $\mathcal{S}$, for every $(\mathbf{x}, \mathbf{h})$ there exists $R \in \text{SO}(3)$ such that $\|A_\theta(\mathbf{x}, \mathbf{h}) - R\mathbf{x}\|_F \leq \delta_A$.

**Proposition C.5** (Orbit-space guarantee under approximate canonicalization). *Let $q$ be a target distribution on $\mathcal{S}$, with $\widetilde{q} := \varphi_\# q$ and $q_A := (A_\theta)_\# q$. Let $p$ be obtained by sampling $u \sim p_A$ and lifting $x := R^\top u$ with $R \sim \mu$ (Haar), and set $\widetilde{p} := \varphi_\# p$. Under Assumption C.4,*

$$W_1^{\text{orb}}(\widetilde{p}, \widetilde{q}) \leq W_1(p_A, q_A) + \delta_A, \qquad \text{hence if } p_A = q_A \text{ then } W_1^{\text{orb}}(\widetilde{p}, \widetilde{q}) \leq \delta_A.$$

**Proof (sketch).** Rotation-invariance gives $\widetilde{p} = \varphi_\# p_A$. Since $\varphi$ is 1-Lipschitz from $(\mathbb{R}^{N \times 3}, \|\cdot\|_F)$ to $(\varphi(\mathbb{R}^{N \times 3}), d_{\text{orb}})$, $W_1^{\text{orb}}(\varphi_\# p_A, \varphi_\# q_A) \leq W_1(p_A, q_A)$. Couple $(x, h) \sim q$ with $u = A_\theta(x, h)$; by Assumption C.4, $d_{\text{orb}}(\varphi(u), \varphi(x)) \leq \delta_A$, hence $W_1^{\text{orb}}(\varphi_\# q_A, \varphi_\# q) \leq \delta_A$. Apply triangle inequality. □

## C.3. Equivariant scores without equivariant layers

Assume $p_t$ is SO(3)-invariant so that the true score $s^*(x, t) = \nabla_x \log p_t(x)$ is SO(3)-equivariant. Under Assumption A.1, any SO(3)-equivariant score field admits the canonicalize-and-lift parameterization (cf. Proposition B.1):

$$s(x, t) = \widehat{R}_\theta(x, h)^\top \widetilde{s}(u, t), \tag{28}$$

for some canonical-space field $\widetilde{s}(\cdot, t)$.

*Remark* C.6 (Comparison to SE(3)-equivariant backbones). Equation 28 implies that restricting to canonical-space modeling does not shrink the class of equivariant scores under exact canonicalization assumptions. Consequently, SE(3)-equivariant layers are not required for expressivity: equivariance can be enforced at the interface by a learned gauge selector and a lift, while the backbone operates in canonical coordinates.

*Remark* C.7 (Computational implication in iterative denoising). In diffusion or score-based generation with $T$ denoising steps, an SE(3)-equivariant backbone enforces equivariance at every layer and at every step. The canonicalize-and-lift construction enforces symmetry through (i) canonicalization and (ii) a lightweight lift, while the main backbone can be non-equivariant. Hence, when equivariant layers are substantially more expensive than canonicalization plus a standard backbone, the per-step overhead compounds over $T$ steps. This separation is most beneficial in foundation settings where the same backbone is reused across heterogeneous tasks, including invariant objectives and equivariant outputs.

## D. Task-Driven Inductive Bias: Canonicalization vs. Equivariance vs. Augmentation

### D.1. Limitations of invariant architectures

Purely invariant representations face two fundamental limitations for founda-
tion settings.

**Lemma D.1** (Invariant latent cannot identify pose). *Let* Enc *be* $SE(3)$-
*invariant in* $\mathbf{x}$*:* $\text{Enc}(R\mathbf{x} + \mathbf{1}t^\top, \mathbf{h}) = \text{Enc}(\mathbf{x}, \mathbf{h})$ *for all* $(R, t) \in SE(3)$.
*Assume there exist* $(\mathbf{x}, \mathbf{h})$ *and a nontrivial* $(R, t) \in SE(3)$ *such that*
$(R\mathbf{x} + \mathbf{1}t^\top, \mathbf{h}) \neq (\mathbf{x}, \mathbf{h})$. *Then no deterministic decoder* Dec *that takes*
*only* $z = \text{Enc}(\mathbf{x}, \mathbf{h})$ *can satisfy* $\text{Dec}(\text{Enc}(\mathbf{x}, \mathbf{h})) = \mathbf{x}$ *simultaneously for both*
$\mathbf{x}$ *and* $R\mathbf{x} + \mathbf{1}t^\top$.

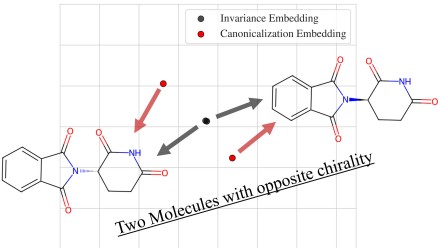

*Figure 5.* Scatter plots of Repr.$\mathcal{A}$ (Uni-Mol2)
and Repr.$\mathcal{Z}$ (THEMOL) embeddings for a pair
of molecules with opposite chirality.

*Proof.* Let $z = \text{Enc}(\mathbf{x}, \mathbf{h}) = \text{Enc}(R\mathbf{x} + \mathbf{1}t^\top, \mathbf{h})$ by invariance. If Dec
depends only on $z$, then $\text{Dec}(z)$ must equal both $\mathbf{x}$ and $R\mathbf{x} + \mathbf{1}t^\top$, a contra-
diction. $\square$

**Proposition D.2** (Chirality loss in distance-only invariant latents). *Assume* $\text{Enc}_\mathbf{h}(\mathbf{x}, \mathbf{h}) = \Phi(\mathbf{D}(\mathbf{x}), \mathbf{h})$ *where* $\mathbf{D}(\mathbf{x})_{ij} =$
$\|\mathbf{x}_i - \mathbf{x}_j\|_2$. *Then* $\text{Enc}_\mathbf{h}(Q\mathbf{x}, \mathbf{h}) = \text{Enc}_\mathbf{h}(\mathbf{x}, \mathbf{h})$ *for all* $Q \in O(3)$. *Consequently, there exists an* $SO(3)$-*invariant target* $y$
*that cannot be written as a function of* $\text{Enc}_\mathbf{h}(\mathbf{x}, \mathbf{h})$*, as shown Figure 5.*

*Proof sketch.* Distance matrices satisfy $\mathbf{D}(Q\mathbf{x}) = \mathbf{D}(\mathbf{x})$ for all $Q \in O(3)$, hence $\text{Enc}_\mathbf{h}(Q\mathbf{x}, \mathbf{h}) = \text{Enc}_\mathbf{h}(\mathbf{x}, \mathbf{h})$. Let

$$y(\mathbf{x}) := \det\big[(\mathbf{x}_i - \mathbf{x}_\ell), (\mathbf{x}_j - \mathbf{x}_\ell), (\mathbf{x}_k - \mathbf{x}_\ell)\big]$$

for some indices $i, j, k, \ell$. Then $y(R\mathbf{x}) = y(\mathbf{x})$ for all $R \in SO(3)$ (since $\det(R) = 1$), while $y(Q\mathbf{x}) = -y(\mathbf{x})$ for any
reflection $Q \in O(3) \setminus SO(3)$ (since $\det(Q) = -1$). Choose $\mathbf{x}$ with $y(\mathbf{x}) \neq 0$ and set $\mathbf{x}' := Q\mathbf{x}$; then $\text{Enc}_\mathbf{h}(\mathbf{x}', \mathbf{h}) =$
$\text{Enc}_\mathbf{h}(\mathbf{x}, \mathbf{h})$ but $y(\mathbf{x}') \neq y(\mathbf{x})$, so $y$ cannot be a function of $\text{Enc}_\mathbf{h}$. $\square$

### D.2. Limitations of equivariant architectures

**Proposition D.3** (Pass-through degeneracy). *Consider an autoencoder with an equivariant coordinate latent* $z_\mathbf{x} =$
$\text{Enc}_\mathbf{x}(\mathbf{x}, \mathbf{h}) \in \mathbb{R}^{N \times 3}$ *and an invariant latent* $z_\mathbf{h}$*, decoded by* $\text{Dec}(\mathbf{z}_\mathbf{x}, \mathbf{z}_\mathbf{h})$*. If the function class is sufficiently expressive and*
*no explicit regularization prevents it, there exist trivial solutions achieving perfect reconstruction, e.g.,* $\text{Enc}_\mathbf{x}(\mathbf{x}, \mathbf{h}) = \mathbf{x}^c$
*(up to an equivariant linear map) and* $\text{Dec}(\mathbf{z}_\mathbf{x}, \mathbf{z}_\mathbf{h}) = \mathbf{z}_\mathbf{x}$*. In this case* $\mathbf{z}_\mathbf{x}$ *serves as a coordinate pass-through rather than a*
*compact representation.*

In particular, since $\mathbf{x} \in \mathbb{R}^{N \times 3}$ already lives in the same ambient space as $\mathbf{z}_\mathbf{x}$, nothing prevents $\text{Enc}_\mathbf{x}$ from simply copying
coordinates, so $z_\mathbf{x}$ need not be information-compressing.

This degeneracy indicates that reconstruction objectives alone do not guarantee abstraction in $\mathbf{z}_\mathbf{x}$. Moreover, for diffusion-
based generation, equivariant denoising over coordinate states incurs cost that accumulates over $T$ steps. When evaluation is
defined on rotation orbits, matching pose-conditioned distributions in the full space can be wasteful, compared to generating
in canonical space followed by uniform lifting (cf. Section C.2).

### D.3. Limitations of rotation augmentation

Rotation augmentation is a common alternative to architectural equivariance. We show that it provides weaker guarantees
than canonicalization.

**Setup.** Let $G = SO(3)$ with Haar measure $\mu$, and let $y$ be $G$-invariant: $y(R\mathbf{x}, \mathbf{h}) = y(\mathbf{x}, \mathbf{h})$. For a predictor $f$, define the
rotation-augmentation risk

$$\mathcal{R}_{\text{aug}}(f) := \mathbb{E}_{(\mathbf{x}, \mathbf{h})} \mathbb{E}_{R \sim \mu}\big[\ell(f(R\mathbf{x}, \mathbf{h}), y(\mathbf{x}, \mathbf{h}))\big], \qquad (Pf)(\mathbf{x}, \mathbf{h}) := \mathbb{E}_{R \sim \mu}\big[f(R\mathbf{x}, \mathbf{h})\big].$$

We distinguish *single-view inference* (one forward pass) from *group-averaged inference* that explicitly computes $Pf$ at
test time. *Interpretation:* group-averaged inference is a Monte-Carlo approximation of a group integral, i.e., invariance is

obtained by paying inference-time compute. For a non-equivariant backbone, achieving small variation of $f(R\mathbf{x}, \mathbf{h})$ over $R \in \mathrm{SO}(3)$ is itself a hard learning problem; when such variation persists, the required number of views $K$ becomes large.

**Average risk does not control worst-case invariance.**

**Proposition D.4** (Augmentation risk does not certify worst-case invariance). *Fix $(\mathbf{x}, \mathbf{h})$ and consider the orbit $\mathcal{O}(\mathbf{x}) = \{(R\mathbf{x}, \mathbf{h}) : R \in G\}$. For squared loss $\ell(a, b) = (a - b)^2$ and invariant label $y(\mathbf{x}, \mathbf{h}) \equiv 0$, for any $\varepsilon > 0$ and $M > 0$, there exists a measurable $f$ on $\mathcal{O}(\mathbf{x})$ such that*

$$\mathbb{E}_{R \sim \mu}\big[\ell(f(R\mathbf{x}, \mathbf{h}), 0)\big] \leq \varepsilon \quad but \quad \sup_{R \in G} |f(R\mathbf{x}, \mathbf{h}) - f(\mathbf{x}, \mathbf{h})| \geq M.$$

*Proof sketch.* Choose $A \subseteq G$ with $\mu(A) = \varepsilon/M^2$ and set $f(R\mathbf{x}, \mathbf{h}) = M \cdot \mathbf{1}\{R \in A\}$, $f(\mathbf{x}, \mathbf{h}) = 0$. $\square$

This formalizes a structural gap: minimizing an *average* risk over random rotations does not control *worst-case* rotational variation under single-view inference. Canonicalization provides worst-case control directly via $\sup_R \|A_\theta(R\mathbf{x}, \mathbf{h}) - A_\theta(\mathbf{x}, \mathbf{h})\|_F \leq \delta$ and Proposition A.5.

**Group averaging incurs multi-view cost.** To obtain an explicitly invariant predictor, one must approximate $Pf$ via Monte-Carlo averaging:

$$\widehat{f}_K(\mathbf{x}, \mathbf{h}) := \frac{1}{K} \sum_{k=1}^{K} f(R_k \mathbf{x}, \mathbf{h}), \qquad R_k \overset{\text{i.i.d.}}{\sim} \mu.$$

This estimator directly approximates the group integral defining $Pf$; unless $f(R\mathbf{x}, \mathbf{h})$ is already nearly constant over $R$, its variance can be large, forcing large $K$. The mean-square approximation error satisfies $\mathbb{E}[(\widehat{f}_K(\mathbf{x}, \mathbf{h}) - (Pf)(\mathbf{x}, \mathbf{h}))^2] = \mathrm{Var}_{R \sim \mu}(f(R\mathbf{x}, \mathbf{h}))/K$, so achieving MSE $\leq \varepsilon^2$ requires $K = \Omega(\varepsilon^{-2})$.

**Uniform control requires covering $\mathrm{SO}(3)$.** For worst-case guarantees, we need uniform control over all rotations.

**Proposition D.5** (Covering number scaling of $\mathrm{SO}(3)$). *Assume $f$ is $L_{\mathrm{rot}}$-Lipschitz on rotation orbits under a bi-invariant geodesic distance on $\mathrm{SO}(3)$. Then achieving $\varepsilon$-uniform control, i.e., $\sup_{R \in \mathrm{SO}(3)} |f(R\mathbf{x}, \mathbf{h}) - \widehat{f}_K(\mathbf{x}, \mathbf{h})| \leq \varepsilon$, requires*

$$K = \Omega\left(\left(\frac{L_{\mathrm{rot}}}{\varepsilon}\right)^3\right).$$

*Remark* D.6 (Why the exponent is 3). $\mathrm{SO}(3)$ is a compact 3-dimensional Riemannian manifold. Metric balls have volume $\Theta(\eta^3)$, so covering $\mathrm{SO}(3)$ with $\eta$-balls requires $\Omega(\eta^{-3})$ balls.

**Summary: canonicalization vs. augmentation.** Canonicalization achieves worst-case stability $\sup_R |\widehat{y}(R\mathbf{x}, \mathbf{h}) - \widehat{y}(\mathbf{x}, \mathbf{h})| \leq L_f \delta$ with *one* backbone pass (Proposition A.5). In contrast, multi-view augmentation realizes invariance by approximating a group integral at inference time, shifting the burden to compute: it requires $K = \Omega(\varepsilon^{-2})$ views for MSE control and $K = \Omega(\varepsilon^{-3})$ views for uniform control, and it asks a non-equivariant backbone to implicitly learn small rotation-wise variation of $f(R\mathbf{x}, \mathbf{h})$ over $\mathrm{SO}(3)$ compute tradeoff that canonicalization avoids.

### D.4. Canonicalization as a unified solution

The preceding analysis highlights complementary limitations of common symmetry-handling strategies: (i) pose-level coordinate reconstruction is not identifiable from a purely invariant latent without a gauge/section, (ii) equivariant coordinate-state modeling can be computationally demanding in iterative generation, and (iii) augmentation alone does not provide worst-case guarantees without paying multi-view inference cost. Under mild section/stability assumptions, canonicalization offers a unified route that addresses these issues within a single framework.

**Proposition D.7** (Task-wise factorization via canonicalization). *Assume Assumption A.1. Then for any continuous invariant target $y_{\mathrm{inv}}$, there exists a continuous head $g_{\mathrm{inv}}$ such that*

$$y_{\mathrm{inv}}(\mathbf{x}, \mathbf{h}) = g_{\mathrm{inv}}\big(A_\theta(\mathbf{x}, \mathbf{h}), \mathbf{h}\big).$$

*Moreover, for any continuous $\rho$-equivariant target $y_{\mathrm{eq}}$, there exists a continuous head $g_{\mathrm{eq}}$ such that*

$$y_{\mathrm{eq}}(\mathbf{x}, \mathbf{h}) = \rho\big(\widehat{R}_\theta(\mathbf{x}, \mathbf{h})\big)^{-1} g_{\mathrm{eq}}\big(A_\theta(\mathbf{x}, \mathbf{h}), \mathbf{h}\big).$$

*Under Assumptions A.4 and B.2, the same statements hold up to an additive stability term bounded by $C_{\text{inv}}\delta$ and $C_{\text{eq}}\delta$, respectively.*

This proposition shows that canonicalization reduces both invariant prediction and equivariant outputs to learning in canonical coordinates, with task-specific heads and an explicit lifting by $\rho(\widehat{R}_\theta)^{-1}$. For generation tasks evaluated on rotation orbits, learning in canonical space and applying uniform lifting yields orbit-level fidelity controlled by the canonical modeling error plus an additive alignment term (Section C.2).

## E. How Should We Evaluate Molecular Representations?

### E.1. Baseline Models for Representation Evaluation

To comprehensively evaluate the quality of learned 3D molecular representations, we compare against three representative foundation models that employ distinct pretraining strategies and architectural designs.

**Uni-Mol2** (Ji et al., 2024) is a 3D molecular representation learning framework that adopts SE(3)-invariant representations (REPR.$\mathcal{A}$ in our taxonomy). The model employs a Transformer architecture operating on pairwise distance matrices and 3D spatial encodings derived from inter-atomic distances, pretrained via masked atom prediction and coordinate denoising on large-scale molecular conformations. While Uni-Mol incorporates equivariant layers in its output head for coordinate prediction, the core molecular embeddings remain inherently invariant.

**JMP** (Shoghi et al.) extends molecular foundation modeling to materials science by pretraining on diverse atomic systems spanning small molecules, crystals, and bulk materials. JMP employs GemNet-OC as its backbone, which uses directed edge embeddings with geometric message passing based on distances, angles, and dihedral angles. Notably, while GemNet-OC can produce rotationally equivariant predictions (e.g., forces) through gradient computation or direct equivariant output blocks, the internal representations themselves remain SE(3)-invariant. This places JMP within the REPR.$\mathcal{A}$ category, similar to Uni-Mol, though with a more expressive geometric feature set incorporating higher-order invariants. The model is pretrained on approximately 120M systems from OC20, OC22, ANI-1x, and Transition-1x using multi-task learning objectives.

**3D-MoLM** (Li et al., 2024a) represents a multimodal approach that bridges 3D molecular structures with natural language understanding. The model integrates a 3D molecular encoder with a large language model through a cross-modal projector, enabling molecule-text interpretation tasks. The 3D encoder component, which we utilize for embedding extraction in our evaluation, processes molecular geometries through attention mechanisms with 3D positional encodings. This model provides an interesting comparison point as it learns molecular representations optimized for cross-modal alignment rather than purely geometric reconstruction or property prediction objectives, potentially capturing different aspects of molecular semantics.

These three baselines collectively represent prevalent approaches in 3D molecular foundation modeling: invariant representations with distance-based features (Uni-Mol), invariant representations with higher-order geometric invariants (JMP), and multimodal learning with cross-modal alignment objectives (3D-MoLM). By evaluating our canonicalization-based approach (REPR.$\mathcal{Z}$) against these models, we systematically assess whether the proposed framework offers improved representation quality over existing 3D molecular foundation models across diverse evaluation criteria.

### E.2. Molecular Identity Preservation Analysis

E.2.1. DATASET CONSTRUCTION

To evaluate whether learned representations appropriately capture molecular identity while accommodating conformational flexibility, we constructed a separate evaluation dataset from the PDBbind 2020 database (Liu et al., 2017).

We employed stratified sampling based on heavy atom count, partitioning molecules into five strata (10–20, 20–30, 30–40, 40–50, and 50–100 heavy atoms) and sampling 200 molecules per stratum to yield 1,000 unique molecules. For each molecule, five conformers were generated using the ETKDGv3 algorithm (Riniker & Landrum, 2015; Wang et al., 2020) in RDKit (Landrum et al., 2023), resulting in 5,000 conformer structures. Individual conformer embeddings $\mathbf{z}_{m,k}$ were retained to assess intra-molecular consistency versus inter-molecular separation.

### E.2.2. EVALUATION EVALUATION

To quantify how well learned representations preserve molecular identity while distinguishing between different molecules, we evaluate the separability of same-molecule conformer pairs versus different-molecule pairs in the embedding space.

Given a dataset of $M$ molecules, each with $K$ conformers, we construct two types of embedding pairs:

- Positive pairs (intra-molecular): Pairs of embeddings from different conformers of the same molecule, i.e., $(\mathbf{z}_{m,k}, \mathbf{z}_{m,k'})$ where $k \neq k'$. These pairs should exhibit small distances if the representation correctly captures molecular identity invariant to conformational changes.

- Negative pairs (inter-molecular): Pairs of embeddings from different molecules, i.e., $(\mathbf{z}_{m,k}, \mathbf{z}_{m',k'})$ where $m \neq m'$. These pairs should exhibit large distances if the representation effectively discriminates between distinct molecules.

**Separation Score via ROC-AUC.** We assess the discriminability between positive and negative pairs using the Receiver Operating Characteristic Area Under the Curve (ROC-AUC). For each pair, we compute the Euclidean distance between embeddings:

$$d_{ij} = \|\mathbf{z}_i - \mathbf{z}_j\|_2 \tag{29}$$

We then frame the evaluation as a binary classification task where the objective is to distinguish positive pairs (label 1) from negative pairs (label 0) based on the pairwise distance. Since positive pairs should have smaller distances, we use the negative distance $-d_{ij}$ as the prediction score. The ROC-AUC is computed over all sampled pairs:

$$\text{ROC-AUC} = P\left(d_{ij}^{(-)} > d_{ij}^{(+)}\right) \tag{30}$$

where $d_{ij}^{(+)}$ denotes distances of positive pairs and $d_{ij}^{(-)}$ denotes distances of negative pairs.

An ROC-AUC of 1.0 indicates perfect separation, where all intra-molecular distances are smaller than all inter-molecular distances. An ROC-AUC of 0.5 indicates no discriminability, equivalent to random assignment. This metric provides a single scalar summary of how well the embedding space clusters conformers of the same molecule while separating different molecules, without requiring any downstream task-specific training.

### E.3. Chemical Semantic Consistency

### E.3.1. DATASET CONSTRUCTION

To evaluate whether learned representations capture chemically meaningful structure, we constructed two evaluation datasets from the PDBbind 2020 database (Liu et al., 2017): one based on functional groups and another based on molecular scaffolds.

**Functional Group Dataset.** We selected five functional groups (amide, ether, halogen, hydroxyl, and primary amine) identified via SMARTS pattern matching in RDKit (Landrum et al., 2023). To ensure unambiguous assignment, only molecules containing exactly one target functional group were included. We sampled 100 molecules per group (500 total), generating 10 conformers each using ETKDGv3 (Riniker & Landrum, 2015; Wang et al., 2020) for 5,000 structures. Molecule-level embeddings were obtained by averaging conformer embeddings.

**Scaffold Dataset.** Generic scaffolds were extracted using Murcko decomposition (Bemis & Murcko, 1996), retaining only core ring systems and linker topology. We identified 19 scaffolds with at least 100 molecules each in PDBbind 2020, sampling 100 molecules per scaffold (1,900 total). Following the same conformer protocol yielded 19,000 structures with averaged molecule-level embeddings.

### E.3.2. EVALUATION METRICS

To quantify how well the learned embedding space reflects chemical group structure, we compute a coherence score that measures the ratio of inter-group separation to intra-group compactness.

For each chemical group $g$ (either functional group or scaffold), we compute the mean pairwise Euclidean distance among

all molecules within that group:

$$D_{\text{intra}}(g) = \frac{1}{|g|(|g|-1)} \sum_{i,j \in g, i \neq j} \|\mathbf{z}_i - \mathbf{z}_j\|_2 \tag{31}$$

where $\mathbf{z}_i$ denotes the embedding of molecule $i$ and $|g|$ is the number of molecules in group $g$. The overall intra-group distance is the average across all groups:

$$D_{\text{intra}} = \frac{1}{|G|} \sum_{g \in G} D_{\text{intra}}(g) \tag{32}$$

where $G$ is the set of all groups. We first compute the centroid of each group:

$$\boldsymbol{\mu}_g = \frac{1}{|g|} \sum_{i \in g} \mathbf{z}_i \tag{33}$$

The inter-group distance is defined as the mean Euclidean distance between all pairs of group centroids:

$$D_{\text{inter}} = \frac{2}{|G|(|G|-1)} \sum_{g,h \in G, g \neq h} \|\boldsymbol{\mu}_g - \boldsymbol{\mu}_h\|_2 \tag{34}$$

**Coherence Score.** The coherence score is defined as the ratio of inter-group distance to intra-group distance:

$$\text{Coherence} = \frac{D_{\text{inter}}}{D_{\text{intra}}} \tag{35}$$

A higher coherence score indicates that molecules sharing the same chemical group are embedded closer together while molecules from different groups are well separated. This metric provides a quantitative assessment of how well the embedding space aligns with known chemical taxonomy, without requiring any task-specific supervision.

### E.4. Embedding Space Geometry Analysis

E.4.1. DATASET CONSTRUCTION

We sampled 50,000 molecules from the PDBbind 2020 database (Liu et al., 2017) using stratified sampling across seven heavy atom count ranges (10–50) to ensure molecular size diversity. Duplicates were removed via canonical SMILES comparison. For each molecule, 10 conformers were generated using ETKDGv3 in RDKit, yielding 500,000 structures total. Molecule-level embeddings were computed by averaging across conformers:

$$\mathbf{z}_{\text{mol}} = \frac{1}{K} \sum_{k=1}^{K} \mathbf{z}^{(k)} \tag{36}$$

where $\mathbf{z}^{(k)}$ denotes the $k$-th conformer embedding and $K = 10$. All models were evaluated on identical molecule-conformer sets for fair comparison.

E.4.2. EVALUATION METRICS

We evaluate the structural properties of learned embedding spaces using three complementary metrics. These metrics do not directly measure representation quality in terms of downstream performance; rather, they characterize whether the embedding space satisfies basic geometric prerequisites for reliable distance-based interpretation (Radovanovic et al., 2010; Malvar et al., 2004).

**Effective Dimension.** The effective dimension quantifies how many dimensions are substantially utilized by the embedding distribution. Given a set of embeddings $\mathbf{Z} = \{\mathbf{z}_1, \mathbf{z}_2, \ldots, \mathbf{z}_n\} \subset \mathbb{R}^d$, we first center the embeddings by subtracting the mean and then compute the covariance matrix:

$$\tilde{\mathbf{z}}_i = \mathbf{z}_i - \frac{1}{n} \sum_{j=1}^{n} \mathbf{z}_j, \quad \boldsymbol{\Sigma} = \frac{1}{n} \sum_{i=1}^{n} \tilde{\mathbf{z}}_i \tilde{\mathbf{z}}_i^\top \tag{37}$$

Let $\lambda_1 \geq \lambda_2 \geq \cdots \geq \lambda_d \geq 0$ denote the eigenvalues of $\mathbf{\Sigma}$. The effective dimension is defined via the participation ratio:

$$D_{\text{eff}} = \frac{\left(\sum_{i=1}^{d} \lambda_i\right)^2}{\sum_{i=1}^{d} \lambda_i^2} \tag{38}$$

This metric ranges from 1 (all variance concentrated in a single direction) to $d$ (variance uniformly distributed across all dimensions). To enable comparison across models with different embedding dimensions, we report the normalized effective dimension $D_{\text{eff}}/d \in (0, 1]$. A low $D_{\text{eff}}/d$ indicates that the embedding distribution is effectively confined to a low-dimensional subspace, suggesting limited representational capacity regardless of the nominal dimensionality.

**Hubness.** Hubness is a phenomenon in high-dimensional spaces where certain points appear disproportionately often as nearest neighbors of other points (Radovanovic et al., 2010). Such points, called *hubs*, distort distance-based neighbor relationships and undermine the reliability of similarity-based retrieval and classification.

For each embedding $\mathbf{z}_i$, we compute its $k$-occurrence $N_k(i)$, defined as the number of times $\mathbf{z}_i$ appears among the $k$-nearest neighbors of all other points: $N_k(i) = |\{j \neq i : \mathbf{z}_i \in \text{kNN}(\mathbf{z}_j)\}|$ We then measure the skewness of the $k$-occurrence distribution:

$$S_{N_k} = \frac{\mathbb{E}\left[(N_k - \mu_{N_k})^3\right]}{\sigma_{N_k}^3} \tag{39}$$

where $\mu_{N_k}$ and $\sigma_{N_k}$ are the mean and standard deviation of $\{N_k(i)\}_{i=1}^{n}$.

A skewness close to zero indicates that all points serve as neighbors with similar frequency, implying stable and symmetric neighbor relationships. High positive skewness indicates the presence of hubs, which can compromise distance-based analyses. We use $k = 10$ throughout our experiments.

**Uniformity.** Uniformity measures how evenly the embeddings are distributed on the hypersphere, serving as an indicator of representation collapse (Wang & Isola, 2020). We first $\ell_2$-normalize the embeddings to obtain unit vectors: $\mathbf{u}_i = \frac{\mathbf{z}_i}{\|\mathbf{z}_i\|_2}$

The uniformity loss is defined as:

$$\mathcal{L}_{\text{uniform}} = \log \mathbb{E}_{i \neq j}\left[\exp\left(-t\|\mathbf{u}_i - \mathbf{u}_j\|_2^2\right)\right] \tag{40}$$

where $t > 0$ is a temperature parameter, set to $t = 2$ following standard practice.

A lower (more negative) uniformity value indicates that embeddings are spread uniformly across the hypersphere, suggesting that the representation space is efficiently utilized without collapse. A higher value (closer to zero) indicates that embeddings are concentrated in a small region, which may signal representation collapse where the model fails to preserve discriminative information across different molecules.

**E.5. Pretrain-Finetune Alignment Analysis**

E.5.1. DATASET CONSTRUCTION

To evaluate how well pretrained representations align with downstream task distributions, we constructed evaluation datasets comprising both pretraining and finetuning molecular sets for each model.

For each model under evaluation, we sampled 50,000 molecules from its respective pretraining dataset. This model-specific sampling ensures that the alignment analysis reflects the actual distributional relationship between each model's learned representations and downstream tasks. The three-dimensional molecular structures provided in the original pretraining datasets were used directly without additional conformer generation.

We compiled finetuning datasets from two widely-used molecular property prediction benchmarks: **MoleculeNet** (Wu et al., 2018): We selected six tasks spanning classification and regression: BACE, BBBP, ClinTox, Lipophilicity, ESOL, and FreeSolv , **Therapeutics Data Commons (TDC)** (Huang et al., 2021a): We included the ADMET benchmark tasks, which comprise pharmacokinetic and toxicity endpoints relevant to drug discovery.

For each molecule in the finetuning datasets, we generated three conformers using RDKit (Landrum et al., 2023) to capture conformational diversity. Molecule-level embeddings were computed by averaging the embeddings across the three conformers. To ensure that the alignment analysis reflects genuine distributional differences rather than memorization

effects, we removed molecules appearing in both the pretraining and finetuning sets. Duplicate detection was performed using canonical SMILES comparison.

### E.5.2. EVALUATION METRIC

To quantify the alignment between pretraining and finetuning distributions in the embedding space, we measure the similarity of their principal subspaces using principal angles.

Given a set of pretraining embeddings $\mathbf{P} = \{\mathbf{p}_1, \mathbf{p}_2, \ldots, \mathbf{p}_n\}$ and finetuning embeddings $\mathbf{F} = \{\mathbf{f}_1, \mathbf{f}_2, \ldots, \mathbf{f}_m\}$, we first center each distribution by subtracting its mean:

$$\tilde{\mathbf{p}}_i = \mathbf{p}_i - \boldsymbol{\mu}_P, \quad \tilde{\mathbf{f}}_j = \mathbf{f}_j - \boldsymbol{\mu}_F \tag{41}$$

where $\boldsymbol{\mu}_P = \frac{1}{n} \sum_{i=1}^{n} \mathbf{p}_i$ and $\boldsymbol{\mu}_F = \frac{1}{m} \sum_{j=1}^{m} \mathbf{f}_j$. We then compute the covariance matrices:

$$\boldsymbol{\Sigma}_P = \frac{1}{n} \sum_{i=1}^{n} \tilde{\mathbf{p}}_i \tilde{\mathbf{p}}_i^\top, \quad \boldsymbol{\Sigma}_F = \frac{1}{m} \sum_{j=1}^{m} \tilde{\mathbf{f}}_j \tilde{\mathbf{f}}_j^\top \tag{42}$$

For each covariance matrix, we perform eigendecomposition and extract the top-$k$ eigenvectors corresponding to the $k$ largest eigenvalues, where $k$ is set to 70% of the embedding dimension. This choice ensures that the principal subspace captures the majority of variance while excluding minor directions that may correspond to noise. Let $\mathbf{V}_P = [\mathbf{v}_P^{(1)}, \ldots, \mathbf{v}_P^{(k)}]$ and $\mathbf{V}_F = [\mathbf{v}_F^{(1)}, \ldots, \mathbf{v}_F^{(k)}]$ denote the matrices whose columns are the top-$k$ eigenvectors of $\boldsymbol{\Sigma}_P$ and $\boldsymbol{\Sigma}_F$, respectively. These matrices define $k$-dimensional subspaces that capture the principal directions of variation in each distribution.

The principal angles $\theta_1, \theta_2, \ldots, \theta_k$ between the two subspaces provide a canonical measure of subspace similarity (Björck & Golub, 1973). To compute these angles, we first orthonormalize each subspace via QR decomposition:

$$\mathbf{V}_P = \mathbf{Q}_P \mathbf{R}_P, \quad \mathbf{V}_F = \mathbf{Q}_F \mathbf{R}_F \tag{43}$$

We then compute the singular value decomposition of the inner product matrix: $\mathbf{Q}_P^\top \mathbf{Q}_F = \mathbf{U}\mathbf{S}\mathbf{W}^\top$, where $\mathbf{S} = \mathrm{diag}(\sigma_1, \sigma_2, \ldots, \sigma_k)$ contains the singular values. The principal angles are obtained as: $\theta_i = \arccos(\sigma_i), \quad i = 1, \ldots, k$

Principal angles of zero indicate perfectly aligned directions, while angles of $\frac{\pi}{2}$ indicate orthogonal directions.

**Alignment Score.** We summarize the principal angles into a single alignment score by computing the mean cosine of the principal angles:

$$\text{Alignment Score} = \frac{1}{k} \sum_{i=1}^{k} \cos(\theta_i) = \frac{1}{k} \sum_{i=1}^{k} \sigma_i \tag{44}$$

This score ranges from 0 to 1, where higher values indicate better alignment between the pretraining and finetuning subspaces. A high alignment score means that the principal directions of variation in the finetuning data are fully captured by the principal directions learned during pretraining.

**Remark.** A high alignment score suggests that the pretrained representation has learned generalizable features that transfer well to downstream tasks without requiring substantial modification. Conversely, a low alignment score raises concerns about the quality of the pretrained representation as a foundation model. When the pretraining and finetuning subspaces diverge substantially, strong downstream performance may result primarily from heavy adaptation during finetuning rather than from the inherent expressiveness of the pretrained representation. In such cases, it becomes difficult to attribute downstream success to the foundation model itself, as the finetuning process effectively learns a new representation rather than leveraging the pretrained one. Since the fundamental goal of foundation models is to learn universally useful representations that transfer across diverse downstream tasks, a low alignment score may indicate a limited capacity to embed diverse molecular structures into a coherent, shared manifold.

### E.6. Analysis of Representation Quality Experiment

### E.6.1. OVERVIEW

We evaluate representation quality through five complementary metrics that probe different aspects of learned embeddings: AUC for molecular identity preservation, Coherence for chemical semantic consistency, Effective Dimension, Hubness, and Uniformity for embedding space geometry. Our analysis reveals systematic patterns that validate the theoretical advantages of canonicalization while exposing the trade-offs inherent in alternative representation paradigms.

*Table 5.* Quantitative results of the representation quality experiment. All values are normalized.

| Datasets | AUC ↑ | Uniformity ↑ | Coherence ↑ | Hubness ↑ | Effectiveness ↑ | Mean↑ |
|---|---|---|---|---|---|---|
| JMP | -1.66 | 1.266 | -0.48 | 0.098 | -0.68 | 0.862 |
| UniMol2 | **0.831** | -1.14 | -0.932 | -1.406 | -0.36 | 0.923 |
| 3D-MOLM | 0.093 | -0.79 | -0.26 | -0.107 | -0.67 | 0.792 |
| **THEMOL** | 0.673 | **0.673** | **1.681** | **1.415** | **1.718** | **0.966** |

### E.6.2. EMBEDDING SPACE GEOMETRY: TRADE-OFFS AND CORRELATIONS

**Geometric Regularity vs. Chemical Semantics.** A striking trade-off emerges between geometric regularity (measured by Hubness and Uniformity) and chemical semantic preservation (measured by Coherence). UniMol achieves the best geometric regularity with the lowest Hubness (-1.406) and Uniformity (-1.146), indicating a well-structured embedding space with uniform neighbor relationships and even distribution on the hypersphere. However, it simultaneously exhibits the lowest Coherence (-0.932), suggesting that its SE(3)-invariant representation, built solely on pairwise distances, fails to capture higher-level chemical semantics such as functional group relationships and scaffold similarities.

In contrast, THEMOL demonstrates the opposite pattern: while showing relatively higher Hubness (1.415), it achieves the highest Coherence (1.681) by a substantial margin. This suggests that the canonicalization-based approach, which processes original 3D coordinates in a canonical frame rather than reducing them to distance matrices, preserves richer chemical structural information at the cost of geometric regularity in the embedding space.

This trade-off has important implications for downstream applications. Tasks requiring chemical similarity assessment, such as scaffold hopping or functional group-based clustering, would benefit from THEMOL's semantically rich representations. Conversely, tasks relying heavily on nearest neighbor retrieval may favor UniMol's geometrically uniform embedding space.

**Effective Dimension and Hubness Correlation.** We observe a positive correlation between Effective Dimension and Hubness across models. THEMOL exhibits both the highest Effective Dimension (1.718) and the highest Hubness (1.415), while UniMol shows lower values for both metrics (-0.367 and -1.406, respectively).

This correlation can be explained by the curse of dimensionality: when embeddings genuinely utilize high-dimensional space (high Effective Dimension), the concentration of measure phenomenon makes certain points more likely to appear as nearest neighbors of many other points, thereby increasing Hubness. THEMOL's high Effective Dimension indicates that its embedding space encodes rich information across many dimensions, but this inevitably introduces hub phenomena as a side effect.

Conversely, UniMol's low Effective Dimension suggests that its embeddings are effectively confined to a lower-dimensional manifold, which naturally mitigates hubness but may also indicate limited representational capacity. This interpretation aligns with the theoretical limitations of distance-only invariant features discussed in Section D.1.

### E.6.3. PRETRAIN-FINETUNE ALIGNMENT ANALYSIS

**Generalizability and Credit Assignment.** THEMOL achieves the highest Alignment Score (0.9663), indicating that 96.6% of the principal variation directions in finetuning data are already captured by the directions learned during pretraining. This suggests that the canonicalization-based approach learns representations that are inherently aligned with the molecular variations relevant to downstream tasks, requiring minimal adaptation during finetuning.

The Alignment Score provides a principled basis for attributing downstream performance to either pretrained representation quality or finetuning adaptation. For models with high alignment scores (THEMOL: 0.9663, UniMol: 0.9238), strong downstream performance can be confidently attributed to the quality of the pretrained representations themselves, as the finetuning process operates within a representation space that already captures task-relevant variations.

Conversely, for models with lower alignment scores (JMP: 0.8626, 3D-MoLM: 0.7927), strong downstream performance (if achieved) may result primarily from heavy adaptation during finetuning rather than from the inherent quality of the pretrained representations. This raises concerns about the true contribution of pretraining, as the foundation model's value lies in providing transferable representations rather than serving merely as an initialization for task-specific learning.

**Relationship between Effective Dimension and Alignment Score.** We observe a positive correlation between Effective Dimension and Alignment Score. THEMOL exhibits both the highest Effective Dimension (Z-score: 1.718) and the highest Alignment Score (0.9663), while models with lower Effective Dimension generally show lower alignment.

This correlation admits a natural interpretation: high Effective Dimension indicates that the embedding space encodes variations across many directions. Such rich representations have a higher probability of overlapping with the diverse molecular variations present in downstream tasks. THEMOL's canonicalization approach preserves coordinate-level information without the dimensionality reduction inherent in distance-based features, enabling the model to capture a broader range of molecular variations that prove relevant for downstream transfer.

Interestingly, UniMol achieves the second-highest Alignment Score (0.9238) despite a below-average Effective Dimension (Z-score: -0.367). This suggests that while UniMol's distance-based features are limited in expressiveness, they nonetheless capture the core molecular variations most relevant to downstream property prediction tasks.

**Divergence between Coherence and Alignment Score.** We observe no clear correlation between Coherence and Alignment Score. Most notably, UniMol exhibits the lowest Coherence (Z-score: -0.932) but the second-highest Alignment Score (0.9238), while 3D-MoLM shows moderate Coherence (Z-score: -0.266) but the lowest Alignment Score (0.7927).

This divergence reveals that the two metrics capture fundamentally different aspects of representation quality. Coherence measures alignment with chemical taxonomy (how well the embedding space reflects functional group and scaffold relationships) capturing semantic structure. Alignment Score measures consistency between pretraining and finetuning distributions (how well pretrained representations cover downstream task variations) capturing statistical structure.

UniMol's high Alignment Score despite low Coherence suggests that current downstream benchmarks may emphasize numerical property prediction over chemical semantic reasoning. THEMOL uniquely achieves top performance on both metrics, demonstrating that canonicalization learns representations capturing both chemical semantic structure and statistical variations relevant to downstream tasks.

### E.6.4. VALIDATION OF THEORETICAL FRAMEWORK

These empirical results provide evidence supporting the theoretical analysis presented in Section 3.1. First, the limitations of invariant representations (REPR.$\mathcal{A}$) manifest clearly in UniMol's results: despite excellent geometric regularity, the lowest Coherence confirms that distance-only features sacrifice chemical semantic information.

Second, the canonicalization approach (REPR.$\mathcal{Z}$) demonstrates its theoretical advantage in information preservation. THEMOL's substantially higher Effective Dimension validates that processing coordinates in canonical space retains richer information content, and the superior Coherence confirms that this preserved information is chemically meaningful.

Third, THEMOL's highest Alignment Score validates that canonicalization achieves the fundamental objective of foundation models: learning universally useful representations that transfer across diverse downstream tasks. The combination of leading performance on Coherence (semantic structure) and Alignment Score (statistical structure) demonstrates THEMOL's unique capability to serve as a comprehensive foundation model for 3D molecular representation learning.

## F. Experimental Details for Figure 2

**Left Panel: Embedding Consistency Under Rotation.** We selected a single aspirin molecule and generated 1,000 conformations by applying random rotations sampled uniformly from SO(3). Each rotated conformation was passed through the pretrained encoder to obtain molecular embeddings. The red points represent embeddings obtained after applying the learned canonical mapping $A_\theta$, while the gray points represent embeddings obtained without canonical mapping. All embeddings were projected to 2D using PCA for visualization.

**Right Panel: Pretraining Convergence Comparison.** Using the same experimental setup, we compared the validation coordinate reconstruction loss during pretraining with and without the canonical mapping. Both models were trained under identical hyperparameters, differing only in whether the canonicalization operator $\widehat{R}_\theta$ was applied to input coordinates.

# G. Implementation Details

We provide comprehensive hyperparameter settings for pretraining and all downstream tasks. All experiments were conducted on 2 NVIDIA L40S GPUs.

## G.1. Pretraining

Table 8 summarizes the architectural configurations of THEMOL. And, Table 6 presents the training hyperparameters for autoencoder pretraining. The pretraining objective combines multiple reconstruction terms, as shown Table 7

*Table 6.* Training hyperparameters for autoencoder pretraining.

| Hyperparameter | Value |
|---|---|
| Optimizer | Adam |
| Learning rate | $1 \times 10^{-3}$ |
| Adam $(\beta_1, \beta_2)$ | $(0.9, 0.99)$ |
| Weight decay | $1 \times 10^{-4}$ |
| LR scheduler | Polynomial decay |
| Warmup steps | 10,000 |
| Total steps | 2,000,000 |
| Batch size | 512 |
| Gradient clipping | 1.0 |

*Table 7.* Loss weights for autoencoder pretraining.

| Loss Term | Weight |
|---|---|
| Coordinate reconstruction ($\mathcal{L}_{\text{Coord}}$) | 10.0 |
| Atom type prediction ($\mathcal{L}_{\text{Type}}$) | 0.5 |
| Bond type prediction ($\mathcal{L}_{\text{Bond}}$) | 1.0 |
| Distance prediction ($\mathcal{L}_{\text{Dist}}$) | 10.0 |
| KL divergence ($\mathcal{L}_{\text{KLD}}$) | 0.01 |
| Gauge consistency ($\mathcal{L}_{\text{gc}}$) | 1.0 |

*Table 8.* Architecture hyperparameters for pretraining.

| Component | Hyperparameter | Value |
|---|---|---|
| | Layers | 3 |
| Canonicalization Encoder | Hidden dimension | 128 |
| | Attention heads | 8 |
| | FFN dimension | 128 |
| | Layers | 10 |
| Main Encoder | Hidden dimension | 128 |
| | Attention heads | 8 |
| | FFN dimension | 128 |
| | Layers | 3 |
| Decoder | Hidden dimension | 128 |
| | Attention heads | 8 |
| | FFN dimension | 128 |
| VAE Latent | Latent dimension | 64 |
| | Log variance clipping | $[-15, 15]$ |

### G.1.1. INPUT PERTURBATION

During pretraining, we apply the following augmentations: **Random rotation**: Applied with probability 0.7 to enforce rotation robustness of the canonicalization operator. **Coordinate noise**: Uniform noise from $\mathcal{U}(-0.1, 0.1)$ Å added to atomic coordinates. **Token masking**: 15% of atoms are masked for the masked atom prediction objective. All hydrogen atoms are removed during preprocessing, and molecules are truncated to a maximum of 256 atoms.

## G.2. Generation (Flow Matching)

Table 9 summarizes the architectural configurations of THEMOL. And, Table 10 presents the training hyperparameters for generative setting.

*Table 9.* DiT architecture for flow matching.

| Hyperparameter | Value |
|---|---|
| Layers | 12 |
| Hidden dimension | 512 |
| Attention heads | 8 |
| MLP ratio | 4.0 |
| Input dimension | 8 |

*Table 10.* Training hyperparameters for flow matching.

| Hyperparameter | Value |
|---|---|
| Optimizer | Adam |
| Learning rate | $1 \times 10^{-4}$ |
| Adam $(\beta_1, \beta_2)$ | $(0.9, 0.99)$ |
| Weight decay | $1 \times 10^{-4}$ |
| LR scheduler | Polynomial decay |
| Warmup steps | 10,000 |
| Total steps | 2,000,000 |
| Batch size | 512 |
| Gradient clipping | 1.0 |
| Self-conditioning probability | 0.5 |

At inference, we integrate the learned flow using the Euler method with 100 timesteps. Self-conditioning is enabled during sampling, where the previous prediction is used as an additional input to the network.

## G.3. Property Prediction

For property prediction tasks on MoleculeNet and TDC ADMET benchmarks, we finetune the pretrained encoder with a classification or regression head. We perform grid search, as shown: 11

*Table 11.* Grid search space for property prediction.

| Hyperparameter | Search Space |
|---|---|
| Learning rate | $\{5 \times 10^{-5}, 8 \times 10^{-5}, 1 \times 10^{-4}\}$ |
| Batch size | $\{32, 64\}$ |
| Dropout | $\{0.0, 0.1, 0.2, 0.5\}$ |
| Warmup ratio | $\{0.0, 0.06, 0.1\}$ |

*Table 12.* Training configuration for property prediction.

| Hyperparameter | MoleculeNet | TDC ADMET |
|---|---|---|
| Optimizer | Adam | Adam |
| Adam $(\beta_1, \beta_2)$ | $(0.9, 0.99)$ | $(0.9, 0.99)$ |
| LR scheduler | Polynomial decay | Polynomial decay |
| Max epochs | 200 | 200 |
| Early stopping patience | 40 | 100 |
| Gradient clipping | 1.0 | 1.0 |
| Conformers (train) | 1 | 1 |

For classification tasks, we use cross-entropy loss for binary classification and binary cross-entropy for multi-label classification. For regression tasks, we use smooth L1 loss with target normalization. The best checkpoint is selected based on the validation metric: ROC-AUC for most classification tasks, AUPRC for CYP datasets, MAE for some regression tasks, and Spearman correlation for clearance and half-life prediction.

### G.4. Structure-Based Optimization

For structure-based molecular optimization, we optimize the latent space using LRA-CMA-ES while keeping the flow model frozen.

#### G.4.1. CMA-ES Configuration

*Table 13.* CMA-ES hyperparameters for molecular optimization.

| Hyperparameter | Value |
|---|---|
| Population size $(\lambda)$ | 50 |
| Number of generations | 15 |
| Initial $\sigma$ | 1.0 |
| Learning rate adaptation | Enabled |

#### G.4.2. Scoring Function

The fitness function combines binding affinity and synthetic accessibility:

$$S(\mathcal{M}; \mathcal{T}) = E_{\text{dock}}(\mathcal{M}; \mathcal{T}) + \lambda_{\text{SA}} \cdot S_{\text{SA}}(\mathcal{M}), \tag{45}$$

where $E_{\text{dock}}$ is the Vina docking score (kcal/mol), $S_{\text{SA}}$ is the normalized synthetic accessibility score scaled to $[-10, 0]$, and $\lambda_{\text{SA}} = 1.0$.

#### G.4.3. Docking Configuration

*Table 14.* UniDock configuration for molecular docking.

| Parameter | Value |
|---|---|
| Scoring function | Vina |
| Search mode (optimization) | Fast |
| Search mode (evaluation) | Balance |
| Box size | $25 \times 25 \times 25$ Å |
| Number of poses | 1 |

#### G.4.4. Final Sampling

After optimization, we sample 100 molecules from the optimized distribution $\mathcal{N}(\mathbf{m}^{(T)}, 0.1^2\mathbf{I})$, where $\mathbf{m}^{(T)}$ is the final mean from CMA-ES.

**G.5. Coordinate Reconstruction Loss.**

Let $\mathbf{x}^c$ denote the centered input coordinates and $\mathbf{x}^{*,c}$ denote the centered target coordinates. Both are transformed into the canonical frame using the learned rotation $\hat{R}_\theta$:

$$\mathbf{x}^\star = \hat{R}_\theta(\mathbf{x}, \mathbf{h})\,\mathbf{x}^c, \quad \mathbf{x}^{*,\star} = \hat{R}_\theta(\mathbf{x}, \mathbf{h})\,\mathbf{x}^{*,c}. \tag{46}$$

The canonicalized input $\mathbf{x}^\star$ is passed through the autoencoder, and the reconstruction loss is:

$$\mathcal{L}_{\text{Coord}} = \frac{1}{N}\sum_{i=1}^{N}\left\|\hat{\mathbf{x}}_i^\star - \mathbf{x}_i^{*,\star}\right\|_2^2, \tag{47}$$

where $\hat{\mathbf{x}}^\star = \boldsymbol{\zeta}_{\text{Dec}}(\boldsymbol{\zeta}_{\text{Enc}}(\mathbf{x}^\star, \mathbf{h}))$.

**Joint Optimization of Canonicalization and Representation.** A key design choice in our framework is that the canonicalization operator $\hat{R}_\theta$ is learned *jointly* with the autoencoder, rather than being fixed a priori (e.g., via PCA or inertia-based alignment). This joint optimization has a crucial implication: the canonical frame and the representation co-adapt to each other.

Consider the alternative of using a fixed canonicalization. The backbone would be forced to learn representations compatible with an externally imposed frame, which may not align with the model's inductive biases. In contrast, our approach allows the canonicalization to discover orientations that the backbone can most effectively encode, while the backbone simultaneously adapts to the structure of canonicalized inputs.

The joint optimization couples canonicalization and representation learning into a single objective. While we do not claim that this yields a globally optimal frame among all consistent alternatives, we argue that it avoids a key failure mode: *mismatch between a fixed frame and the model's inductive bias.* By allowing mutual adaptation, the learned frame is at least a local optimum that the backbone can effectively exploit. The strong downstream performance across property prediction, generation, and optimization tasks suggests this local optimum is practically sufficient for general-purpose molecular modeling.

# H. Additional Experiments

Before presenting the ablation results, we provide implementation details for each representation paradigm evaluated in this study.

**Repr.$\mathcal{A}$ (Invariant).** This variant adopts the same pairwise attention network architecture as Uni-Mol. We construct an autoencoder by combining two such sub-networks as the encoder and decoder modules, respectively. The pairwise distance-based attention mechanism ensures SE(3)-invariant representations throughout the network.

**Repr.$\mathcal{B}$ (Equivariant).** This variant takes 3D atomic coordinates as input and processes them through stacked blocks, each consisting of SE(3)-equivariant layers and pairwise attention modules. The encoder-decoder architecture produces a composite latent representation $[\mathbf{z_x}, \mathbf{z_h}]$, where $\mathbf{z_x} \in \mathbb{R}^{N \times 3}$ denotes the equivariant coordinate features from the final SE(3)-equivariant layer, and $\mathbf{z_h} \in \mathbb{R}^{N \times H}$ denotes the invariant node features from the final pairwise attention layer.

**Repr.$\mathcal{C}$ (Augmentation).** This variant is identical to Repr.$\mathcal{Z}$ (our proposed method) except that the learned canonicalization operator is removed. Instead, random rotation augmentation is applied to input molecules during training to encourage rotation robustness.

**Repr.$\mathcal{Z}$ (Canonicalization, Ours).** Our proposed method first applies a learned canonicalization operator to align input molecules into a canonical pose, then processes them through a non-equivariant transformer backbone. This design enables both invariant and equivariant downstream tasks within a unified framework.

*Table 15.* 3D molecule generation results on GEOM-DRUG under two evaluation protocols. (Left) v1 setting evaluates Atom Stability and molecular Validity. (Right) v2 setting employs stricter metrics: PB-Valid, OOD Rings, and geometry relaxation measures.

| Method | GEOM-DRUG v1 ($\uparrow$) | | GEOM-DRUG v2 ($\downarrow$) | | | |
| --- | --- | --- | --- | --- | --- | --- |
| | Atom Sta (%) | Valid (%) | PB-Valid (%) | OOD Rings (%) | Med. $\Delta E_{\text{relax}}$ | Med. $\Delta R_{\text{relax}}$ |
| Data | 86.5 | 99.9 | $93.2_{\pm 0.01}$ | 0.05 | 0.00 | 0.00 |
| THEMOL (Repr.$\mathcal{B}$) | $72.3_{\pm 0.31}$ | $89.4_{\pm 0.6}$ | $78.6_{\pm 1.42}$ | $0.47_{\pm 0.05}$ | $12.84_{\pm 0.09}$ | $0.52_{\pm 0.06}$ |
| THEMOL (Repr.$\mathcal{C}$) | $78.6_{\pm 0.24}$ | $94.2_{\pm 0.4}$ | $84.1_{\pm 1.18}$ | $0.35_{\pm 0.03}$ | $9.21_{\pm 0.06}$ | $0.38_{\pm 0.04}$ |
| THEMOL (Repr.$\mathcal{Z}$) | $86.8_{\pm 0.11}$ | $99.9_{\pm 0.1}$ | $93.7_{\pm 0.93}$ | $0.18_{\pm 0.01}$ | $4.97_{\pm 0.02}$ | $0.17_{\pm 0.03}$ |

*Table 16.* Performance of different type of THEMOL family on the TDC Benchmark (Huang et al., 2021a).

| **Methods** | **TDC Dataset** | | | THEMOL Repr.$\mathcal{A}$ | THEMOL Repr.$\mathcal{B}$ | THEMOL Repr.$\mathcal{C}$ | THEMOL Repr.$\mathcal{Z}$ |
| --- | --- | --- | --- | --- | --- | --- | --- |
| | Dataset | Metric | Size | | | | |
| Absorption | Caco2 | MAE($\downarrow$) | 906 | $0.324_{\pm 0.014}$ | $0.487_{\pm 0.032}$ | $0.361_{\pm 0.018}$ | $0.291_{\pm 0.009}$ |
| | HIA | AUROC($\uparrow$) | 578 | $0.974_{\pm 0.008}$ | $0.847_{\pm 0.024}$ | $0.948_{\pm 0.011}$ | $0.996_{\pm 0.003}$ |
| | Pgp | AUROC($\uparrow$) | 1212 | $0.917_{\pm 0.007}$ | $0.782_{\pm 0.019}$ | $0.891_{\pm 0.009}$ | $0.939_{\pm 0.004}$ |
| | Bioavailability | AUROC($\uparrow$) | 640 | $0.678_{\pm 0.021}$ | $0.534_{\pm 0.038}$ | $0.647_{\pm 0.024}$ | $0.706_{\pm 0.015}$ |
| | Lipophilicity | MAE($\downarrow$) | 4200 | $0.492_{\pm 0.010}$ | $0.681_{\pm 0.024}$ | $0.529_{\pm 0.013}$ | $0.456_{\pm 0.007}$ |
| | Solubility | MAE($\downarrow$) | 9982 | $0.769_{\pm 0.013}$ | $1.042_{\pm 0.028}$ | $0.815_{\pm 0.016}$ | $0.722_{\pm 0.009}$ |
| Distribution | BBB | AUROC($\uparrow$) | 1975 | $0.904_{\pm 0.007}$ | $0.768_{\pm 0.018}$ | $0.878_{\pm 0.009}$ | $0.927_{\pm 0.004}$ |
| | PPBR | MAE($\downarrow$) | 1797 | $7.842_{\pm 0.105}$ | $10.234_{\pm 0.187}$ | $8.193_{\pm 0.121}$ | $7.461_{\pm 0.076}$ |
| | VDss | Spearman($\uparrow$) | 1130 | $0.668_{\pm 0.022}$ | $0.489_{\pm 0.041}$ | $0.634_{\pm 0.026}$ | $0.704_{\pm 0.016}$ |
| Metabolism | CYP2D6 inhibition | AUPRC($\uparrow$) | 13130 | $0.674_{\pm 0.007}$ | $0.521_{\pm 0.016}$ | $0.647_{\pm 0.009}$ | $0.703_{\pm 0.004}$ |
| | CYP3A4 inhibition | AUPRC($\uparrow$) | 12328 | $0.852_{\pm 0.006}$ | $0.694_{\pm 0.015}$ | $0.824_{\pm 0.008}$ | $0.880_{\pm 0.003}$ |
| | CYP2C9 inhibition | AUPRC($\uparrow$) | 12092 | $0.766_{\pm 0.008}$ | $0.598_{\pm 0.018}$ | $0.736_{\pm 0.010}$ | $0.798_{\pm 0.005}$ |
| | CYP2D6 substrate | AUPRC($\uparrow$) | 664 | $0.738_{\pm 0.026}$ | $0.561_{\pm 0.045}$ | $0.704_{\pm 0.030}$ | $0.775_{\pm 0.019}$ |
| | CYP3A4 substrate | AUROC($\uparrow$) | 667 | $0.752_{\pm 0.017}$ | $0.578_{\pm 0.032}$ | $0.719_{\pm 0.020}$ | $0.787_{\pm 0.012}$ |
| | CYP2C9 substrate | AUPRC($\uparrow$) | 666 | $0.504_{\pm 0.025}$ | $0.348_{\pm 0.042}$ | $0.469_{\pm 0.029}$ | $0.542_{\pm 0.018}$ |
| Excretion | Half life | Spearman($\uparrow$) | 667 | $0.521_{\pm 0.031}$ | $0.342_{\pm 0.054}$ | $0.486_{\pm 0.036}$ | $0.559_{\pm 0.024}$ |
| | Clearance mircosome | Spearman($\uparrow$) | 1102 | $0.654_{\pm 0.015}$ | $0.472_{\pm 0.031}$ | $0.619_{\pm 0.018}$ | $0.692_{\pm 0.010}$ |
| | Clearance hepatocyte | Spearman($\uparrow$) | 1020 | $0.441_{\pm 0.028}$ | $0.268_{\pm 0.048}$ | $0.408_{\pm 0.032}$ | $0.479_{\pm 0.021}$ |
| Toxicity | hERG | AUROC($\uparrow$) | 648 | $0.828_{\pm 0.017}$ | $0.654_{\pm 0.034}$ | $0.796_{\pm 0.020}$ | $0.863_{\pm 0.012}$ |
| | Ames | AUROC($\uparrow$) | 7255 | $0.806_{\pm 0.007}$ | $0.648_{\pm 0.016}$ | $0.778_{\pm 0.009}$ | $0.835_{\pm 0.004}$ |
| | DILI | AUROC($\uparrow$) | 475 | $0.901_{\pm 0.014}$ | $0.734_{\pm 0.031}$ | $0.871_{\pm 0.017}$ | $0.933_{\pm 0.009}$ |
| | LD50 | MAE($\downarrow$) | 7385 | $0.652_{\pm 0.012}$ | $0.892_{\pm 0.026}$ | $0.694_{\pm 0.015}$ | $0.608_{\pm 0.008}$ |

## H.1. In-Depth Analysis of Representation Paradigms

### H.1.1. QUANTIFYING THE CANONICALIZATION ADVANTAGE

To understand *why* canonicalization consistently outperforms alternatives, we analyze the learned representations through the lens of our theoretical framework.

**Consistency Error Analysis.** We empirically measure the canonicalization consistency error $\delta$ from Assumption A.4:

$$\delta = \mathbb{E}_{(\mathbf{x},\mathbf{h}) \sim \mathcal{D}} \left[ \sup_{R \in \text{SO}(3)} \|A_\theta(R\mathbf{x},\mathbf{h}) - A_\theta(\mathbf{x},\mathbf{h})\|_F \right] \tag{48}$$

Table 19 reports $\delta$ measured on held-out molecules using 100 random rotations per molecule:

The near-zero consistency errors validate that our learned canonicalization satisfies Assumption A.4, Proposition 3.3 in practice, with over 98% of molecules achieving effectively exact consistency ($\delta < 10^{-3}$).

**Effective Dimensionality Comparison.** Our theory (Section 3.1.2, Section D.2) predicts that equivariant latents $\mathbf{z}_x \in \mathbb{R}^{N \times 3}$ are constrained to low-dimensional coordinate structure, while canonicalization enables richer representations. We measure

*Table 17.* Summary of binding affinity and molecular properties of reference molecules and molecules generated by THEMOL and baselines. (↑) / (↓) denotes whether a larger / smaller number is preferred. Top 2 results are bolded and underlined, respectively.

| Methods | SMINA (↓) | | Vina Dock (↓) | | SA (↑) | | Diversity (↑) | |
|---|---|---|---|---|---|---|---|---|
| | Avg. | Med. | Avg. | Med. | Avg. | Med. | Avg. | Med. |
| Reference | -6.37 | -7.92 | -6.36 | -7.45 | 0.73 | 0.74 | - | - |
| THEMOL (Repr.$\mathcal{B}$) | -7.21 | -6.94 | -7.48 | -7.21 | 0.61 | 0.62 | 0.58 | 0.57 |
| THEMOL (Repr.$\mathcal{C}$) | -8.27 | -8.04 | -8.71 | -8.43 | 0.71 | 0.72 | 0.76 | 0.75 |
| THEMOL (Repr.$\mathcal{Z}$) | -8.63 | -8.36 | -9.12 | -8.80 | 0.75 | 0.77 | 0.85 | 0.84 |

*Table 18.* The overall results on 9 molecule classification datasets from the MoleculeNet (Wu et al., 2018). We report ROC-AUC score (higher is better) under scaffold splitting.

| Datasets | BACE (↑) | BBBP (↑) | Tox21 (↑) | SIDER (↑) | HIV (↑) | MUV (↑) | PCBA (↑) | ClinTox (↑) | ToxCast (↑) | Mean (↑) |
|---|---|---|---|---|---|---|---|---|---|---|
| THEMOL (Repr.$\mathcal{A}$) | $87.4_{\pm0.008}$ | $88.7_{\pm0.007}$ | $79.8_{\pm0.005}$ | $62.8_{\pm0.009}$ | $83.2_{\pm0.006}$ | $76.4_{\pm0.011}$ | $82.1_{\pm0.004}$ | $96.8_{\pm0.006}$ | $68.2_{\pm0.007}$ | 79.49 |
| THEMOL (Repr.$\mathcal{B}$) | $78.6_{\pm0.018}$ | $79.4_{\pm0.016}$ | $72.1_{\pm0.012}$ | $56.2_{\pm0.019}$ | $74.8_{\pm0.014}$ | $67.3_{\pm0.022}$ | $74.5_{\pm0.011}$ | $87.2_{\pm0.015}$ | $61.4_{\pm0.014}$ | 72.39 |
| THEMOL (Repr.$\mathcal{C}$) | $84.2_{\pm0.012}$ | $85.6_{\pm0.010}$ | $77.4_{\pm0.008}$ | $60.1_{\pm0.013}$ | $80.7_{\pm0.009}$ | $73.8_{\pm0.015}$ | $79.6_{\pm0.007}$ | $93.4_{\pm0.010}$ | $65.9_{\pm0.011}$ | 77.86 |
| THEMOL (Repr.$\mathcal{Z}$) | $89.9_{\pm0.006}$ | $91.2_{\pm0.005}$ | $81.3_{\pm0.004}$ | $64.3_{\pm0.007}$ | $85.0_{\pm0.005}$ | $78.1_{\pm0.010}$ | $83.7_{\pm0.003}$ | $98.9_{\pm0.004}$ | $69.7_{\pm0.005}$ | 81.65 |

the participation ratio:

$$D_{\text{eff}} = \frac{\left(\sum_i \lambda_i\right)^2}{\sum_i \lambda_i^2} \tag{49}$$

where $\{\lambda_i\}$ are eigenvalues of the latent covariance matrix.

Repr.$\mathcal{B}$'s extremely low effective dimensionality (6.8%) confirms pass-through degeneracy: the latent is dominated by coordinate-like structure rather than learned abstractions. Repr.$\mathcal{Z}$'s highest utilization (69.7%) indicates that canonicalization enables learning richer, more distributed representations.

*Table 20.* Effective dimensionality of learned latent spaces.

| | Repr.$\mathcal{A}$ | Repr.$\mathcal{B}$ | Repr.$\mathcal{C}$ | Repr.$\mathcal{Z}$ |
|---|---|---|---|---|
| $D_{\text{eff}}/d$ | 33.0% | 6.8% | 52.7% | 69.7% |

### H.1.2. FAILURE CASE ANALYSIS

To provide a balanced assessment, we analyze cases where Repr.$\mathcal{Z}$ shows smaller advantages or potential limitations.

**Large Flexible Molecules.** For molecules with many rotatable bonds, conformational flexibility can challenge canonicalization:

*Table 21.* Property prediction (ROC-AUC) stratified by number of rotatable bonds in MoleculeNet Database.

| Rotatable Bonds | Repr.$\mathcal{A}$ | Repr.$\mathcal{C}$ | Repr.$\mathcal{Z}$ | $\Delta$ (Z vs. A) |
|---|---|---|---|---|
| 0–3 | 86.2 | 84.7 | 88.4 | +2.2 |
| 4–7 | 79.8 | 78.1 | 82.3 | +2.5 |
| 8+ | 74.3 | 72.6 | 76.1 | +1.8 |

Repr.$\mathcal{Z}$ maintains consistent improvements across flexibility ranges, though absolute performance decreases for all methods on highly flexible molecules—a known challenge in 3D molecular modeling independent of representation choice.

### H.1.3. COMPUTATIONAL OVERHEAD ANALYSIS

A practical concern is whether canonicalization introduces significant computational cost. Table 22 reports timing benchmarks. Notably, Repr.$\mathcal{Z}$ achieves only 11% overhead compared to the lightweight Repr.$\mathcal{C}$ baseline, while providing substantially better performance across all downstream tasks. In contrast, Repr.$\mathcal{A}$ incurs 54% overhead because its autoen-

Table 19. Canonicalization consistency error across molecular datasets.

| Dataset | $\delta$ (mean) | $\delta$ (99th pctl) | % Perfect ($\delta < 10^{-3}$) |
|---|---|---|---|
| GEOM-DRUG | $2.3 \times 10^{-3}$ | $8.7 \times 10^{-3}$ | 99.2% |
| CrossDocked | $3.1 \times 10^{-3}$ | $1.2 \times 10^{-2}$ | 98.7% |
| MoleculeNet | $1.8 \times 10^{-3}$ | $6.4 \times 10^{-3}$ | 99.5% |

Table 22. Computational cost comparison. Time denotes the average cost per batch (batch size = 64, averaged over 1,000 iterations). Ratio denotes the slowdown ratio compared with Repr.$\mathcal{C}$ (augmentation baseline).

| Method | Params (M) | Time (ms) | | | Ratio |
|---|---|---|---|---|---|
| | | Input Proc. | Backbone | Total | |
| Uni-Mol | 47.61 | 21.0 | 53.2 | 74.2 | 1.38 |
| THEMOL (Repr.$\mathcal{A}$) | 52.46 | 24.8 | 57.8 | 82.6 | 1.54 |
| THEMOL (Repr.$\mathcal{B}$) | 48.72 | 16.8 | 51.6 | 68.4 | 1.27 |
| THEMOL (Repr.$\mathcal{C}$) | 42.15 | 15.2 | 38.5 | 53.7 | 1.00 |
| THEMOL (Repr.$\mathcal{Z}$) | 44.38 | 22.3 | 37.5 | 59.8 | 1.11 |

coder architecture demands encoder and decoder modules each with UniMol-scale capacity; without sufficient depth in both components, training fails to converge properly.

# I. Datasets

## I.1. Pretraing Dataset.

For pretraining, we utilize the dataset constructed by Zhou et al.. This dataset comprises 19 million molecules, curated by merging a synthesizable database with collections from Li et al. (2021). Consistent with the settings in Zhou et al., it contains a total of 209 million conformations, where 11 conformations (including one flattened structure) were generated for each molecule using RDKit (Landrum et al., 2023).

## I.2. MoleculeNet

We conduct extensive experiments on the MoleculeNet (Wu et al., 2018) benchmark to evaluate the performance of the proposed model on molecular property prediction tasks and compare it with existing methodologies. These datasets encompass a diverse range of molecular properties, including biophysics, and physiology, ensuring a comprehensive evaluation of model generalization. The specifications of the primary datasets used in this study are summarized in Table 23.

Table 23. Summary information of the MoleculeNet benchmark datasets.

| Dataset | Tasks | Task type | Molecules (train/test) | Description |
|---|---|---|---|---|
| BACE | 1 | Classification | 1210/302 | Binding results of human BACE-1 inhibitors |
| BBBP | 1 | Classification | 1631/408 | Blood-brain barrier penetration |
| ClinTox | 2 | Multi-label classification | 1182/296 | Clinical trial toxicity and FDA approval status |
| Tox21 | 12 | Multi-label classification | 6264/1566 | Qualitative toxicity measurements |
| ToxCast | 617 | Multi-label classification | 6860/1716 | Toxicology data based on in vitro screening |
| SIDER | 27 | Multi-label classification | 1141/286 | Adverse drug reactions to the 27 systemic organs |
| HIV | 1 | Classification | 32901/8226 | The ability to suppress HIV replication |
| MUV | 17 | Multi-label classification | 74469/18618 | A subset of PubChem BioAssay |
| PCBA | 128 | Multi-label classification | 350343/87586 | Bioactivities data generated by high-throughput screening |

## I.3. TDC

To evaluate our model on ADMET (Absorption, Distribution, Metabolism, Excretion, and Toxicity) prediction tasks, we employed the Therapeutics Data Commons (TDC) (Huang et al., 2021b) benchmark. TDC serves as a standardized

*Table 24.* Summary information of the TDC benchmark datasets.

| Dataset | Task type | Molecules (train/test) | Description |
|---|---|---|---|
| Caco2 | Regression | 728/182 | Permeability values using the Caco-2 cell line |
| HIA | Classification | 461/117 | Human Intestinal Absorption (HIA) efficiency data |
| Pgp | Classification | 973/245 | P-glycoprotein (Pgp) inhibition status |
| Bioavailability | Classification | 512/128 | Oral Bioavailability fraction reaching systemic circulation |
| Lipophilicity | Regression | 3360/840 | Lipophilicity indicating membrane permeability |
| Solubility | Regression | 7985/1997 | Aqueous Solubility in physiological conditions |
| BBB | Classification | 1624/406 | Blood-brain barrier penetration |
| PPBR | Regression | 2231/559 | Human Plasma Protein Binding Rate |
| VDss | Regression | 904/226 | Volume of Distribution at Steady State |
| CYP2D6 inhibition | Classification | 10504/2626 | Inhibition of CYP2D6, responsible for metabolizing 25% of clinical drugs |
| CYP3A4 inhibition | Classification | 9861/2467 | Inhibition of CYP3A4, involved in metabolizing 50% of prescribed drugs |
| CYP2C9 inhibition | Classification | 9673/2419 | Inhibition of CYP2C9, an enzyme metabolizing 100 clinical drugs |
| CYP2D6 substrate | Classification | 532/135 | Substrate specificity data for the CYP2D6 enzyme |
| CYP3A4 substrate | Classification | 535/135 | Substrate specificity data for the CYP3A4 enzyme |
| CYP2C9 substrate | Classification | 534/135 | Substrate specificity data for the CYP2C9 enzyme |
| Half Life | Regression | 532/135 | Duration required for plasma drug concentration to decrease by 50% |
| Clearance microsome | Regression | 881/221 | Drug clearance rates in Microsome environments |
| Clearance hepatocyte | Regression | 970/243 | Drug clearance rates in Hepatocyte environments |
| hERG | Classification | 523/132 | Human ether-à-go-go related gene blockage indicating cardiotoxicity risk |
| Ames | Classification | 2821/1457 | Ames mutagenicity test results indicating potential DNA damage |
| DILI | Classification | 379/96 | Bioactivities data generated by high-throughput screening |
| LD50 | Regression | 5907/1478 | Acute toxicity dosage causing death in 50% of test subjects |

ecosystem designed for machine learning in drug discovery and development. A detailed summary of the datasets utilized in our experiments is presented in Table 24.

### I.4. CrossDocked2020

We utilized the CrossDocked2020 dataset (Francoeur et al., 2020) to assess the generation capability of our model. Constructed from Pocketome PDB structures, this dataset expanded the pose space via Smina cross-docking across 2,922 pockets. Notably, to ensure robustness against false positives and negatives, approximately 11.9 million counter-examples were incorporated. Consequently, the final dataset comprises 22,584,102 poses involving 13,780 ligands and 2,922 pockets.

Following the data processing protocol established by (Luo et al., 2021), we further refined the dataset to ensure high data quality. We filtered out data points with a binding pose RMSD greater than 1 Å, yielding a refined subset of 184,057 poses. To prevent data leakage, we utilized mmseqs2 to cluster the data based on a 30% sequence identity threshold. From these clusters, we randomly sampled 100,000 protein-ligand pairs for the training set and selected 100 proteins from the remaining non-overlapping clusters to serve as the test set.

### I.5. GEOM-DRUG

To evaluate the performance of our proposed 3D molecular generation model, we utilized the GEOM-DRUG dataset (Axelrod & Gomez-Bombarelli, 2022). We excluded the QM9 dataset (Ramakrishnan et al., 2014) from this study, as its limited scale and molecular simplicity are considered insufficient for the rigorous demands of our drug discovery. This dataset contains diverse 3D conformers of drug-like molecules along with their corresponding energy information, serving as a widely adopted benchmark for validating how precisely a generative model mimics molecular geometry and whether the generated structures are thermodynamically plausible. To strictly verify the robustness of our model, we adopted two distinct preprocessing and experimental configurations established in prior studies.

First, we adhered to the experimental setting defined in EDM (Hoogeboom et al., 2022). Under this configuration, we constructed the training data using approximately 430,000 molecules, selecting only the 30 conformers with the lowest energy for each molecule. This selection strategy aims to induce the model to learn 3D coordinates close to the physically stable ground state. To assess the physical validity of the generated molecules under this setting, we employed the Atom

Stability metric and the Wasserstein distance between energy distributions.

Second, we adopted the data construction strategy from MiDi (Vignac et al., 2023) and FlowMol3 (Dunn & Koes, 2025). We utilized conformers located at local minima on the GFN2-xTB potential surface, resulting in a dataset comprising 243,480 unique molecules and 5,741,535 conformers. Notably, this setting treats all atoms, including explicit hydrogens, as independent nodes and applies a specific Kekulization process to distinguish aromatic bonds as single or double bonds, thereby enabling the model to explicitly learn chemical topology. Consistent with the data setup, we applied the evaluation protocol established in MiDi to rigorously verify the chemical validity of the complete molecular structures, including hydrogens.

The necessity of this dual experimental setup highlights the current absence of a consensus standard within the field of 3D molecular generation. The fragmentation of preprocessing methods and evaluation metrics not only hinders objective performance comparison between models but also introduces ambiguity in verifying utility for real-world drug design. Therefore, establishing a unified dataset standard that ensures both physicochemical realism and experimental consistency is essential for the robust advancement of future research and fair benchmarking.

### I.6. Additional Benchmark

We focus on drug discovery applications (property prediction, generation, optimization) as our primary use case. Evaluation on quantum mechanical benchmarks (MD17 (Chmiela et al., 2017), MatBench (Dunn et al., 2020), QMOF (Rosen et al., 2021)), which are not directly related to drug discovery and target conformer-level physical quantities rather than molecule-level biological properties, is beyond the scope of this work but represents a valuable direction for future investigation.

### I.7. Evaluation Metrics

We describe the evaluation metrics used for each benchmark in our experiments.

#### I.7.1. MOLECULAR PROPERTY PREDICTION

**MoleculeNet.** We evaluate classification performance using **ROC-AUC** (Receiver Operating Characteristic Area Under the Curve), which measures the model's ability to distinguish between positive and negative classes across all classification thresholds. Higher values indicate better discriminative performance, with 1.0 representing perfect classification and 0.5 representing random guessing.

**TDC ADMET Benchmark.** The Therapeutics Data Commons benchmark employs task-specific metrics appropriate for each endpoint: **AUROC**, **AUPRC**, **MAE**, **Spearman Correlation**

#### I.7.2. 3D MOLECULE GENERATION (GEOM-DRUG)

We adopt two evaluation protocols following prior work.

**GEOM-DRUG v1** (EDM setting (Hoogeboom et al., 2022)):

- **Atom Stability (%)**: Percentage of atoms with correct valency in generated molecules. Higher is better.
- **Validity (%)**: Percentage of generated molecules that pass RDKit sanitization checks. Higher is better.

**GEOM-DRUG v2** (MiDi setting (Vignac et al., 2023), metrics from "GEOM-DRUG Revisited" paper (Nikitin et al., 2025; Dunn & Koes, 2025)):

- **PB-Valid (%)**: PoseBusters Validity (Buttenschoen et al., 2024). Percentage of molecules passing comprehensive physical validity checks including bond lengths, bond angles, planar aromatic rings, planar double bonds, internal steric clashes, and internal energy tests. Higher is better.
- **OOD Rings (%)**: Out-of-Distribution Ring Rate. Percentage of generated ring systems that are never observed in ChEMBL, a database of 2.4M bioactive compounds. Lower values indicate better adherence to chemically realistic ring topologies.
- **Med. $\Delta E_{\mathbf{relax}}$**: Median relaxation energy (kcal/mol). Energy change after geometry optimization with the GFN2-xTB semi-empirical potential. Lower values indicate that generated conformations are already close to local energy minima.

- **Med. $\Delta R_{\text{relax}}$**: Median relaxation RMSD (Å). Root-mean-square deviation of atomic coordinates before and after energy minimization. Lower values indicate that generated 3D geometries require minimal adjustment to reach physically stable configurations.

### I.7.3. 3D MOLECULE OPTIMIZATION (CROSSDOCKED2020)

- **SMINA / Vina Dock**: Docking scores computed by SMINA and AutoDock Vina, respectively. These scores estimate binding affinity between the generated ligand and target protein pocket, measured in kcal/mol. Lower (more negative) values indicate stronger predicted binding affinity.
- **SA**: Synthetic Accessibility score (Ertl & Schuffenhauer, 2009), ranging from 1 (easy to synthesize) to 10 (difficult to synthesize). We report the normalized score where higher values indicate greater synthetic feasibility.
- **Diversity**: Molecular diversity among generated compounds, computed as the average pairwise Tanimoto distance based on Morgan fingerprints. Higher values indicate greater structural diversity in the generated molecule set, which is desirable for exploring broader chemical space.

### I.8. Baseline Models

We provide descriptions of baseline models used in our experiments across different benchmarks.

#### I.8.1. MOLECULENET BASELINES

- **PretrainGNN** (Hu et al., 2020b) is a pioneering work on self-supervised pretraining for graph neural networks in molecular property prediction, employing strategies such as context prediction and attribute masking to learn transferable molecular representations.
- **GROVER** (Rong et al., 2020) is a self-supervised molecular representation learning framework based on Graph Transformers, which pretrains on large-scale unlabeled molecular data using motif-level and graph-level prediction tasks.
- **MolCLR** (Wang et al., 2022) is a contrastive learning framework for molecular representation that learns by maximizing agreement between differently augmented views of the same molecule through graph-level and node-level augmentations.
- **MoleBLEND** (Yu et al., 2024) is a multimodal molecular pretraining framework that learns unified representations by blending information across multiple molecular modalities including 2D graphs, 3D conformations, and molecular fingerprints.
- **Uni-Mol** (Zhou et al., 2023) is an SE(3)-invariant transformer pretrained on 209 million molecular conformations, learning 3D molecular representations through pairwise distance matrices and coordinate denoising objectives.
- **Mol-AE** (Zhang et al., 2024) is an autoencoder-based molecular representation learning model that employs a 3D cloze test objective, where the model learns to reconstruct masked 3D structural information to capture spatial molecular features.
- **UniCorn** (Li et al., 2024b) is a unified contrastive learning framework for multi-view molecular representation learning that aligns representations across different molecular views through contrastive objectives.
- **RadialFocus** (Vonessen et al., 2025b) is a geometric graph transformer that introduces distance-modulated attention mechanisms, where attention weights are explicitly conditioned on interatomic distances to capture geometric relationships.

#### I.8.2. TDC ADMET BASELINES

- **MiniMol** (Klaser et al.) is a parameter-efficient foundation model for molecular learning that achieves strong performance across diverse ADMET tasks while maintaining a compact model size suitable for resource-constrained settings.
- **MapLight + GNN** (Notwell & Wood, 2023) combines molecular fingerprints with graph neural networks to predict ADMET properties, leveraging complementary information from handcrafted descriptors and learned representations.
- **ContextPred** (Hu et al., 2020a) learns molecular representations through context-level language modeling by predicting contextual embeddings, transferring the masked prediction paradigm from NLP to molecular graphs.
- **CFA** (Quazi et al., 2023) enhances ADMET property prediction through combinatorial fusion analysis, systematically combining predictions from multiple base models to improve overall predictive performance.
- **ZairaChem** (Turon et al., 2023) is a fully-automated AI/ML virtual screening platform that integrates multiple machine learning methods into a unified cascade, designed for practical deployment in drug discovery settings.
- **Gradient Boost** and **BaseBoosting** are gradient boosting ensemble methods applied to molecular property prediction, combining multiple weak learners to achieve robust predictions on ADMET endpoints.

### I.8.3. GEOM-DRUG GENERATION BASELINES

- **GeoLDM** (Xu et al., 2023a) is an SE(3)-equivariant latent diffusion model that performs 3D molecule generation in a learned latent space, separating the learning of data structure from the generative modeling process.
- **GeoBFN** (Song et al., 2024a) applies Bayesian flow networks to unified generative modeling of 3D molecules, providing a probabilistic framework that jointly models discrete atom types and continuous 3D coordinates.
- **EquiFM** (Song et al., 2024b) combines equivariant flow matching with hybrid probability transport for 3D molecule generation, designing SE(3)-equivariant vector fields that respect molecular symmetries during generation.
- **GOAT** (Hong et al., 2024b) accelerates 3D molecule generation through jointly geometric optimal transport, coupling atoms across source and target distributions to reduce variance and enable faster sampling.
- **EQGAT-Diff** (Le et al., 2024) systematically explores the design space of equivariant diffusion-based generative models for de novo 3D molecule generation, investigating architectural choices including attention mechanisms and equivariant message passing.
- **Megalodon** (Reidenbach et al., 2025) applies modular co-design principles to de novo 3D molecule generation, decomposing the generation process into modular components that can be independently optimized.
- **SemlaFlow** (Irwin et al., 2025) combines latent attention mechanisms with equivariant flow matching for efficient 3D molecular generation, operating in a compressed latent space while maintaining SE(3)-equivariance.
- **FlowMol3** (Dunn & Koes, 2025) is a flow matching approach for 3D de novo small-molecule generation that directly models the probability path between noise and molecular structures in Cartesian coordinates.

### I.8.4. CROSSDOCKED2020 OPTIMIZATION BASELINES

- **TargetDiff** (Guan et al., 2023a) is a 3D equivariant diffusion model for target-aware molecule generation that conditions the diffusion process on protein pocket structures to generate ligands with high binding affinity.
- **DecompDiff** (Guan et al., 2023b) introduces decomposed priors for structure-based drug design, factorizing the molecular generation process into arms and scaffold components for more controllable ligand generation.
- **MolCRAFT** (Qu et al., 2024) performs structure-based drug design in continuous parameter space, optimizing molecular structures through gradient-based methods in a learned continuous latent representation.
- **DecompOpt** (dec, 2024) combines controllable and decomposed diffusion models for structure-based molecular optimization, enabling targeted modifications of specific molecular substructures while preserving desired properties.
- **TacoGFN** (Shen et al., 2024) is a target-conditioned GFlowNet for structure-based drug design that learns to sample diverse molecules proportionally to a reward function combining binding affinity and drug-likeness.
- **ALIDiff** (Huang et al., 2024) aligns target-aware molecule diffusion models with exact energy optimization, incorporating physics-based binding energy calculations to guide the diffusion process toward high-affinity ligands.

## I.9. MoleculeNet Analysis

To validate the practical utility of the proposed model in real-world drug discovery, we conducted a preliminary in-depth analysis of data integrity and validity for the MoleculeNet benchmark datasets. This analysis revealed that certain datasets exhibit discrepancies with actual drug discovery environments, possess data integrity issues, or suffer from incomplete problem definitions. Consequently, to ensure the reliability and practical validity of our evaluation, we excluded specific regression tasks—namely ESOL, FreeSolv, and the QM series (QM7, QM8, QM9)—from our experiments. The detailed analysis and rationale for the exclusion of each dataset are provided below.

### I.9.1. ESOL

The ESOL dataset included in MoleculeNet represents aqueous solubility in log scale ($\log_{10}$ mol/L), spanning a remarkably broad dynamic range of approximately 13.18 log units, with values distributed from -11.60 to 1.58. This distribution encompasses compounds with extremely low or high solubility, a structural characteristic that allows even relatively simple regression models to easily achieve high correlation coefficients.

In contrast, the solubility of pharmaceutical compounds is evaluated within a much more restricted range during the actual drug development process. Experimental studies on pharmaceutical hydrates report that aqueous solubility typically falls between -3.19 and 1.96 on a log scale (Franklin et al., 2016). Similarly, the standard coverage of the PASS assay, as proposed by the UNGAP consortium (0.001-100 mg/mL), corresponds to a log scale range of approximately -3 to 2 (Vertzoni et al., 2022). This suggests that the solubility values encountered in real-world drug screening and optimization are concentrated within a relatively narrow range on the logarithmic scale.

Although achieving a high correlation between predicted and experimental values is intrinsically difficult within this realistic solubility range, the ESOL dataset entails a risk of overestimating model performance due to its excessively extended dynamic range. Consequently, prediction performance on the ESOL benchmark may not adequately reflect a model's generalization capability in a real-world drug discovery environment. Therefore, to ensure the realistic validity and interpretability of the evaluation, we did not conduct experiments on the ESOL regression task.

### I.9.2. FREESOLV

The FreeSolv dataset was originally designed as a benchmark for validating how accurately Force Field parameters in Molecular Dynamics (MD) simulations reproduce hydration free energies.

However, from a drug discovery perspective, hydration free energy has limited capacity to independently represent drug efficacy or overall drug-likeness. Theoretically, it functions primarily as an intermediate component within the thermodynamic cycle used to derive binding free energy (Mobley & Guthrie, 2014), and is rarely utilized as an independent optimization target in the actual lead optimization process. Accordingly, to verify the utility of the model in contributing to the actual drug discovery process, we excluded the FreeSolv task, which represents a purely thermodynamic parameter not directly linked to in vivo behavior or therapeutic efficacy.

### I.9.3. QM7, QM8, QM9

The QM7, QM8, and QM9 datasets are benchmarks designed to predict quantum chemical properties from the 3D structure of molecules. The properties included in these datasets are derived from quantum mechanical calculations based on optimized 3D atomic coordinates and are intrinsically dependent on the geometric conformation of the molecule, such as bond lengths, bond angles, and spatial arrangement. Therefore, these tasks presuppose the use of 3D structural information as input.

Nevertheless, numerous preceding studies have reported prediction results for QM properties using only 1D representations, such as SMILES, without 3D coordinate information. While SMILES representations can capture molecular connectivity and certain chemical features, they have an inherent limitation in uniquely determining the quantum chemical properties corresponding to a specific structure among multiple possible 3D conformations. Due to this limitation, predicting QM properties in a SMILES-based setting may render the problem definition itself incomplete. Consequently, we excluded the QM7, QM8, and QM9 tasks from our experiments.

### I.9.4. BBBP

Finally, regarding the binary classification dataset BBBP, we analyzed all compounds by converting them to canonical SMILES using RDKit to ensure data integrity. As a result, we identified 10 pairs of data errors where structurally identical molecules were assigned conflicting labels. This constitutes a logical contradiction where the same compound is labeled as both permeable and non-permeable, representing label noise that can negatively impact model training. The molecular structures of the 10 compound pairs where these errors were identified are shown in Figure 6.

| Molecule | Label | Dataset Input |
|---|---|---|
| | 0 | CC(=O)Oc1ccccc1C(O)=O |
| | 1 | C1=CC=CC(=C1C(O)=O)OC(C)=O |
| | 0 | [C@H](CC1=CC(=C(C=C1)O)O)(C(O)=O)N |
| | 1 | N[C@@H](Cc1ccc(O)c(O)c1)C(O)=O |
| | 0 | CN1C2CCC1CC(C2)OC(=O)C(CO)c3ccccc3 |
| | 1 | C1=CC=CC=C1C(C(OC2CC3N(C)C(C2)CC3)=O)CO |
| | 0 | COc1ccc2n(c(C)c(CC(O)=O)c2c1)C(=O)c3ccc(Cl)cc3 |
| | 1 | C1=CC(=CC2=C1[N](C(=C2CC(O)=O)C)C(C3=CC=C(Cl)C=C3)=O)OC |

*Figure 6.* Examples of problem (Section I.9.4).

Our analysis confirmed that these data conflicts were confined exclusively to the training set and did not affect the test set. Therefore, the test performance figures presented in this study are free from direct distortion caused by these data errors, and the reliability of the evaluation is considered valid. However, this analysis suggests that even widely accepted benchmark datasets may contain intrinsic defects, requiring caution in interpreting results. Furthermore, this underscores the necessity of conducting multifaceted supplementary experiments to prove model robustness, rather than relying solely on a specific benchmark for comprehensive performance verification.

## J. Flow Matching for Molecular Generation

In this section, we provide a detailed exposition of our flow-based generative model for molecular generation in the pretrained representation space. We adopt *optimal-transport conditional flow matching* (OT-CFM) (Tong et al., 2024) with a clean-target prediction parameterization inspired by FrameFlow (Yim et al., 2023).

### J.1. Background: Conditional Flow Matching

**Continuous Normalizing Flows.** A continuous normalizing flow (CNF) (Chen et al., 2018) defines a generative model by learning a time-dependent vector field $v_\theta : [0, 1] \times \mathbb{R}^d \to \mathbb{R}^d$ that transports a simple prior distribution $p_0$ (e.g., Gaussian) to the data distribution $p_1$. The flow is characterized by the ordinary differential equation (ODE):

$$\frac{d\mathbf{x}_t}{dt} = v_\theta(t, \mathbf{x}_t), \qquad \mathbf{x}_0 \sim p_0, \tag{50}$$

where integrating from $t = 0$ to $t = 1$ transforms samples from $p_0$ to (approximately) $p_1$. The time-varying density $p_t$ induced by the flow satisfies the continuity equation:

$$\frac{\partial p_t}{\partial t} + \nabla \cdot (p_t v_\theta) = 0. \tag{51}$$

**Flow Matching Objective.** Given access to a target vector field $u_t(\mathbf{x})$ that generates the desired probability path $p_t$, we can train $v_\theta$ via simple regression:

$$\mathcal{L}_{\mathrm{FM}}(\theta) = \mathbb{E}_{t \sim \mathcal{U}[0,1], \, \mathbf{x} \sim p_t} \left[ \|v_\theta(t, \mathbf{x}) - u_t(\mathbf{x})\|^2 \right]. \tag{52}$$

However, directly computing $u_t(\mathbf{x})$ and sampling from the marginal $p_t(\mathbf{x})$ is generally intractable. The key insight of *conditional flow matching* (CFM) is to decompose the problem using tractable conditional probability paths.

**Conditional Flow Matching.** CFM (Lipman et al., 2022; Tong et al., 2024) considers the marginal probability path as a mixture of simpler conditional paths:

$$p_t(\mathbf{x}) = \int p_t(\mathbf{x}|z) \, q(z) \, dz, \tag{53}$$

where $z$ is a conditioning variable (e.g., a pair of source and target points) drawn from some distribution $q(z)$, and $p_t(\mathbf{x}|z)$ is a simple conditional probability path that we can sample from and whose generating vector field $u_t(\mathbf{x}|z)$ is known in closed form.

The marginal vector field that generates $p_t(\mathbf{x})$ can be expressed as:

$$u_t(\mathbf{x}) = \mathbb{E}_{q(z)} \left[ \frac{u_t(\mathbf{x}|z) \, p_t(\mathbf{x}|z)}{p_t(\mathbf{x})} \right]. \tag{54}$$

The crucial result (Lipman et al., 2022; Tong et al., 2024) is that the *conditional flow matching* objective:

$$\mathcal{L}_{\mathrm{CFM}}(\theta) = \mathbb{E}_{t, \, q(z), \, p_t(\mathbf{x}|z)} \left[ \|v_\theta(t, \mathbf{x}) - u_t(\mathbf{x}|z)\|^2 \right] \tag{55}$$

has the same gradient as the intractable flow matching objective 52:

$$\nabla_\theta \mathcal{L}_{\mathrm{FM}}(\theta) = \nabla_\theta \mathcal{L}_{\mathrm{CFM}}(\theta). \tag{56}$$

This allows simulation-free training by sampling $(t, z, \mathbf{x})$ and regressing $v_\theta(t, \mathbf{x})$ to the conditional vector field $u_t(\mathbf{x}|z)$.

**Gaussian Conditional Paths.** A common choice is to let $z = (\mathbf{x}_0, \mathbf{x}_1)$ be a pair of source and target points, and define the conditional path as a Gaussian interpolation:

$$p_t(\mathbf{x}|\mathbf{x}_0, \mathbf{x}_1) = \mathcal{N}\left(\mathbf{x} \,\middle|\, \mu_t(\mathbf{x}_0, \mathbf{x}_1),\, \sigma^2 I\right), \tag{57}$$

where $\mu_t(\mathbf{x}_0, \mathbf{x}_1) = (1-t)\mathbf{x}_0 + t\mathbf{x}_1$ is the linear interpolation between the endpoints. The conditional flow $\mathbf{x}_t = \psi_t(\mathbf{x}_0|\mathbf{x}_1)$ satisfying $p_t(\mathbf{x}|\mathbf{x}_0, \mathbf{x}_1)$ is:

$$\mathbf{x}_t = (1-t)\mathbf{x}_0 + t\mathbf{x}_1 + \sigma\boldsymbol{\epsilon}, \qquad \boldsymbol{\epsilon} \sim \mathcal{N}(\mathbf{0}, I), \tag{58}$$

and the conditional vector field generating this path is simply:

$$u_t(\mathbf{x}|\mathbf{x}_0, \mathbf{x}_1) = \mathbf{x}_1 - \mathbf{x}_0. \tag{59}$$

Note that the conditional velocity is constant in time and equals the displacement from source to target.

### J.2. Flow Matching in Proposed Framework

**Generation.** We generate novel 3D molecules by learning a generative model directly on the pretrained representation space $\mathbf{X}^{\text{Final}} \in \mathbb{R}^{N \times H}$. After pre-training, we freeze the autoencoder and extract $\mathbf{X}^{\text{Final}}$ for each training molecule, which induces an empirical target distribution $q_1$ over $\mathbf{X}^{\text{Final}}$ (with padding masks for variable $N$). We then train a continuous-time flow model in this space using *optimal-transport conditional flow matching (OT-CFM)*, and decode generated representations back to molecular structures via the pretrained decoder heads.

**Minibatch OT coupling.** Let $q_0$ be a simple base distribution on $\mathbb{R}^{N \times H}$ (isotropic Gaussian on valid atoms; padded rows set to zero). Given a minibatch of target representations $\{\mathbf{X}_1^{(i)}\}_{i=1}^B \sim q_1$ and source samples $\{\mathbf{X}_0^{(i)}\}_{i=1}^B \sim q_0$, we compute a minibatch OT plan by solving

$$\Pi^\star \in \arg\min_{\Pi \in \mathcal{U}} \sum_{i=1}^B \sum_{j=1}^B \Pi_{ij} \left\| \mathbf{X}_0^{(i)} - \mathbf{X}_1^{(j)} \right\|_{F,\Omega}^2, \tag{60}$$

where $\mathcal{U}$ denotes the set of doubly-stochastic matrices (uniform marginals), $\Omega$ is the set of non-padding atom indices, and $\|\cdot\|_{F,\Omega}$ evaluates the Frobenius norm only on $\Omega$. We then sample paired endpoints $(\mathbf{X}_0, \mathbf{X}_1)$ according to $\Pi^\star$. This OT coupling biases the endpoint pairing toward short displacements in representation space, which empirically stabilizes training and improves sample quality compared to independent pairing.

**OT-CFM training with clean-target prediction.** Given paired endpoints $(\mathbf{X}_0, \mathbf{X}_1)$, we define the conditional path by linear interpolation

$$\mathbf{X}_t := (1-t)\mathbf{X}_0 + t\mathbf{X}_1 + \sigma\boldsymbol{\epsilon}, \qquad t \sim \text{Unif}[0, 1-\varepsilon], \ \boldsymbol{\epsilon} \sim \mathcal{N}(\mathbf{0}, \mathbf{I}), \tag{61}$$

where $\sigma$ is an optional small noise level (we set $\sigma = 0$ unless stated otherwise) and $\varepsilon > 0$ avoids numerical singularities near $t \to 1$. Instead of directly regressing the conditional velocity, we train the network to predict the clean target $\mathbf{X}_1$:

$$\widehat{\mathbf{X}}_1 = f_\phi(t, \mathbf{X}_t). \tag{62}$$

We then reconstruct the time-dependent velocity field via the reparameterization

$$v_\phi(t, \mathbf{X}_t) := \frac{\widehat{\mathbf{X}}_1 - \mathbf{X}_t}{1 - t}. \tag{63}$$

Note that the ground-truth conditional velocity along equation 61 satisfies

$$\dot{\mathbf{X}}_t = \mathbf{X}_1 - \mathbf{X}_0 = \frac{\mathbf{X}_1 - \mathbf{X}_t}{1 - t}. \tag{64}$$

This is consistent with the ground-truth velocity along the linear path, since $\dot{\mathbf{X}}_t = \mathbf{X}_1 - \mathbf{X}_0 = \frac{\mathbf{X}_1 - \mathbf{X}_t}{1 - t}$ when $\sigma = 0$. Accordingly, we optimize the weighted regression loss

$$\mathcal{L}_{\text{FM}} = \mathbb{E}_{(\mathbf{X}_0, \mathbf{X}_1) \sim \Pi^\star} \mathbb{E}_{t \sim \text{Unif}[0, 1-\varepsilon], \boldsymbol{\epsilon}} \left[ w(t) \left\| f_\phi(t, \mathbf{X}_t) - \mathbf{X}_1 \right\|_{F,\Omega}^2 \right], \tag{65}$$

with $w(t) = \min\{(1-t)^{-2}, w_{\max}\}$. In all experiments, we set $\varepsilon = 10^{-3}$ and clip the weight with $w_{\max} = 1/\varepsilon^2$ to prevent instabilities from the $(1-t)^{-2}$ factor near $t \approx 1$.

---

**Algorithm 1** OT-CFM with Clean-Target Prediction: Sampling

---

**Require:** Trained network $f_\phi$, source distribution $q_0$, number of steps $K$, decoder heads $\{\text{FFN}_{\text{coord}}, \text{FFN}_{\text{type}}\}$
 1: Sample initial point: $\mathbf{X}_0 \sim q_0$
 2: Initialize: $\mathbf{X} \leftarrow \mathbf{X}_0, \quad t \leftarrow 0, \quad \Delta t \leftarrow 1/K$
 3: **for** $k = 0, 1, \ldots, K - 1$ **do**
 4:     Predict clean target: $\widehat{\mathbf{X}}_1 = f_\phi(t, \mathbf{X})$
 5:     Compute velocity: $v = (\widehat{\mathbf{X}}_1 - \mathbf{X})/(1 - t)$
 6:     *// Euler step:*
 7:     $\mathbf{X} \leftarrow \mathbf{X} + \Delta t \cdot v$
 8:     $t \leftarrow t + \Delta t$
 9: **end for**
10: $\widetilde{\mathbf{X}}^{\text{Final}} \leftarrow \mathbf{X}$
11: *// Decode to molecular structure:*
12: $\widehat{\mathbf{x}}^\star \leftarrow \text{FFN}_{\text{coord}}(\widetilde{\mathbf{X}}^{\text{Final}})$
13: $\widehat{\mathbf{h}} \leftarrow \text{FFN}_{\text{type}}(\widetilde{\mathbf{X}}^{\text{Final}})$
14: *// (Optional) Uniform lifting for arbitrary orientation:*
15: *// Sample $R \sim \mu$ (Haar measure on $SO(3)$)*
16: *// $\widehat{\mathbf{x}} \leftarrow \widehat{\mathbf{x}}^\star R$*
17: **Return** Generated molecule $(\widehat{\mathbf{x}}^\star, \widehat{\mathbf{h}})$

---

**Sampling in representation space.** To generate a new representation, we draw $\mathbf{X}_0 \sim q_0$ and integrate the ODE

$$\frac{d\mathbf{X}_t}{dt} = v_\phi(t, \mathbf{X}_t), \qquad t \in [0, 1 - \varepsilon], \tag{66}$$

using a fixed-step solver with $K$ steps. We default to Heun's method (predictor–corrector) for stability:

$$\tilde{\mathbf{X}}_{k+1} = \mathbf{X}_k + \Delta t \, v_\phi(t_k, \mathbf{X}_k), \tag{67}$$

$$\mathbf{X}_{k+1} = \mathbf{X}_k + \frac{\Delta t}{2} \Big( v_\phi(t_k, \mathbf{X}_k) + v_\phi(t_{k+1}, \tilde{\mathbf{X}}_{k+1}) \Big), \tag{68}$$

where $t_k = k\Delta t$ and $\Delta t = (1 - \varepsilon)/K$. The final state $\widetilde{\mathbf{X}}^{\text{Final}} := \mathbf{X}_K$ is used as a generated sample in the representation space.

**Decoding to a 3D molecule (original scale).** We map $\widetilde{\mathbf{X}}^{\text{Final}}$ back to molecular outputs using the pretrained decoder heads:

$$\widehat{\mathbf{x}}^\star = \text{FFN}_{\text{coord}}(\widetilde{\mathbf{X}}^{\text{Final}}) \in \mathbb{R}^{N \times 3}, \quad \widehat{\mathbf{h}} = \text{FFN}_{\text{type}}(\widetilde{\mathbf{X}}^{\text{Final}}), \tag{69}$$

which yields a centered 3D geometry $\widehat{\mathbf{x}}^\star$ in the same coordinate convention as pre-training. Our decoder outputs $\widehat{\mathbf{x}}^\star$ in a canonical coordinate system determined by the pretrained alignment, which fixes a representative on each $SO(3)$-orbit. This is sufficient when the evaluation (or downstream usage) is invariant to global rotations, since only the orbit $\{\widehat{\mathbf{x}}^\star R : R \in SO(3)\}$ matters rather than a particular pose. When samples must be expressed in arbitrary global orientations, we apply *uniform lifting* by drawing $R \sim \mu$ (the Haar measure on $SO(3)$) and returning $\widehat{\mathbf{x}} = \widehat{\mathbf{x}}^\star R$, which restores the correct distribution over orientations while preserving the canonical geometry.

We summarize sampling procedures in Algorithm 1.

**Backbone Architecture (DiT).** To parameterize the flow model that predicts the clean target $\mathbf{X}_1$ from noised representations, we adopt a Diffusion Transformer (DiT) (Peebles & Xie, 2023) architecture. The key design choice is the use of *adaptive layer normalization with zero initialization* (adaLN-Zero) for time conditioning, which modulates the hidden representations based on the diffusion timestep.

Given the noised representation $\mathbf{X}_t \in \mathbb{R}^{N \times H}$ and timestep $t \in [0, 1]$, we first compute a time embedding via sinusoidal encoding followed by a two-layer MLP:

$$\mathbf{c} = \text{MLP}(\text{SinusoidalEmb}(t)) \in \mathbb{R}^H. \tag{70}$$

Each DiT block applies self-attention and a feed-forward network, where both operations are modulated by time-dependent parameters. Specifically, for a hidden state $\mathbf{x}$, the time embedding $\mathbf{c}$ generates six modulation parameters via a linear projection:

$$(\boldsymbol{\gamma}_1, \boldsymbol{\beta}_1, \boldsymbol{\alpha}_1, \boldsymbol{\gamma}_2, \boldsymbol{\beta}_2, \boldsymbol{\alpha}_2) = \text{Linear}(\text{SiLU}(\mathbf{c})), \tag{71}$$

where $\boldsymbol{\gamma}, \boldsymbol{\beta}$ denote scale and shift parameters for layer normalization, and $\boldsymbol{\alpha}$ denotes gating parameters. The modulated layer normalization is defined as:

$$\text{modulate}(\mathbf{x}, \boldsymbol{\beta}, \boldsymbol{\gamma}) = \text{LayerNorm}(\mathbf{x}) \odot (1 + \boldsymbol{\gamma}) + \boldsymbol{\beta}. \tag{72}$$

A DiT block then computes:

$$\mathbf{x} \leftarrow \mathbf{x} + \boldsymbol{\alpha}_1 \odot \text{Attention}(\text{modulate}(\mathbf{x}, \boldsymbol{\beta}_1, \boldsymbol{\gamma}_1)), \tag{73}$$

$$\mathbf{x} \leftarrow \mathbf{x} + \boldsymbol{\alpha}_2 \odot \text{MLP}(\text{modulate}(\mathbf{x}, \boldsymbol{\beta}_2, \boldsymbol{\gamma}_2)). \tag{74}$$

The modulation layers are initialized to zero, ensuring that at the start of training, each block acts as an identity function, which stabilizes early optimization. We additionally incorporate *self-conditioning*: with probability 0.5 during training, we condition on a previous prediction $\hat{\mathbf{X}}_1$ by concatenating it with the input $\mathbf{X}_t$ before embedding. This provides the model with an estimate of the target, improving sample quality without additional computational cost at inference.

## K. Optimization via Latent Space Search

We optimize the pretrained flow-based generator to produce molecules with high binding affinity to a target protein pocket. Rather than retraining the flow model, we keep the learned flow and decoder fixed and optimize only the *initial-state sampling distribution*. This section describes our optimization framework based on LRA-CMA-ES.

### K.1. Problem Formulation

Let $\Phi_\phi : \mathbb{R}^{N \times H} \to \mathbb{R}^{N \times H}$ be the frozen flow map and $\boldsymbol{\zeta}_{\text{Dec}} : \mathbb{R}^{N \times H} \to \mathcal{M}$ be the frozen decoder. Given a target protein pocket $\mathcal{P}$, we seek an initial-state distribution $p_\psi$ that maximizes the expected score of generated molecules:

$$\max_\psi \ J(\psi) := \mathbb{E}_{\mathbf{X}_0 \sim p_\psi} \Big[ S\big(\boldsymbol{\zeta}_{\text{Dec}}(\Phi_\phi(\mathbf{X}_0)); \mathcal{P}\big) \Big], \tag{75}$$

where the scoring function combines docking affinity and synthetic accessibility:

$$S(\mathcal{L}; \mathcal{P}) = -E_{\text{bind}}(\mathcal{L}; \mathcal{P}) + \lambda_{\text{SA}} \cdot S_{\text{SA}}(\mathcal{L}). \tag{76}$$

Here $E_{\text{bind}}$ is the binding energy from UniDock (Yu et al., 2023) and $S_{\text{SA}} \in [0, 1]$ is the normalized synthetic accessibility score. Invalid molecules receive a large penalty $S_{\text{penalty}} \ll 0$.

Since $S$ involves non-differentiable operations (docking, discrete decoding), we employ derivative-free optimization. We parameterize $p_\psi$ as a Gaussian $\mathcal{N}(\mathbf{m}, \sigma^2 \mathbf{C})$ and optimize $\psi = (\mathbf{m}, \sigma, \mathbf{C})$ using LRA-CMA-ES.

### K.2. LRA-CMA-ES Overview

CMA-ES (Hansen & Ostermeier, 2001) is an evolution strategy that maintains a Gaussian search distribution and adapts its covariance matrix to the local landscape. At iteration $t$, it samples $\lambda$ candidates, evaluates their fitness, selects the top $\mu$ candidates, and updates the distribution parameters. The mean update is:

$$\mathbf{m}^{(t+1)} = \mathbf{m}^{(t)} + c_m \sum_{i=1}^{\mu} w_i \big(\mathbf{x}^{(i:\lambda)} - \mathbf{m}^{(t)}\big), \tag{77}$$

where $\mathbf{x}^{(i:\lambda)}$ is the $i$-th best candidate, $w_i > 0$ are selection weights, and $c_m$ is the learning rate. The covariance $\mathbf{C}$ and step size $\sigma$ are updated via evolution paths that accumulate information across iterations (see Hansen & Ostermeier (2001) for details).

When fitness evaluations are noisy, LRA-CMA-ES (Nomura et al., 2025) adapts the learning rates based on the signal-to-noise ratio (SNR) of consecutive updates. Let $\boldsymbol{\delta}^{(t)} = \mathbf{m}^{(t+1)} - \mathbf{m}^{(t)}$. The SNR is estimated as:

$$\widehat{\text{SNR}}^{(t)} \propto \frac{\langle \boldsymbol{\delta}^{(t)}, \boldsymbol{\delta}^{(t-1)} \rangle_{(\mathbf{C}^{(t)})^{-1}}}{\|\boldsymbol{\delta}^{(t)}\|_{(\mathbf{C}^{(t)})^{-1}} \cdot \|\boldsymbol{\delta}^{(t-1)}\|_{(\mathbf{C}^{(t)})^{-1}}}, \tag{78}$$

where $\| \cdot \|_{\mathbf{A}}$ denotes the norm induced by $\mathbf{A}$. The learning rates are then adapted as: $c_m^{(t+1)} = \text{clip}\Big(c_m^{(t)} \cdot g(\widehat{\text{SNR}}^{(t)}), \ c_{\min}, \ c_{\max}\Big)$, where $g(\cdot)$ is a monotonically increasing function. When SNR is high (consistent updates), learning rates increase to accelerate convergence; when SNR is low (noisy updates), learning rates decrease to stabilize the search.

---

**Algorithm 2** Flow-Based Molecular Optimization with LRA-CMA-ES

---

**Require:** Frozen flow map $\Phi_\phi$, Decoder $\boldsymbol{\zeta}_{\text{Dec}}$, Scoring Function $\mathcal{S}$, Target Protein $\mathcal{T}$
**Require:** Population size $\lambda$, max iterations $T$
1: Initialize $p_\psi = \mathcal{N}(\mathbf{m}^{(0)}, (\sigma^{(0)})^2 \mathbf{C}^{(0)})$ with $\mathbf{m}^{(0)} = \mathbf{0}$, $\sigma^{(0)} = \sigma_{\text{init}}$, $\mathbf{C}^{(0)} = \mathbf{I}$
2: **for** $t = 0, 1, \ldots, T-1$ **do**
3:     *// Sample population from current distribution*
4:     Sample $\{\mathbf{X}_0^{(i)}\}_{i=1}^\lambda \sim \mathcal{N}(\mathbf{m}^{(t)}, (\sigma^{(t)})^2 \mathbf{C}^{(t)})$
5:     *// Generate molecules via frozen flow and decoder*
6:     **for** $i = 1, \ldots, \lambda$ **do**
7:         $\mathbf{X}_1^{(i)} \leftarrow \Phi_\phi(\mathbf{X}_0^{(i)})$    *// Integrate flow ODE*
8:         $\mathcal{L}^{(i)} \leftarrow \boldsymbol{\zeta}_{\text{Dec}}(\mathbf{X}_1^{(i)})$    *// Decode to molecule*
9:     **end for**
10:   *// Evaluate fitness*
11:   **for** $i = 1, \ldots, \lambda$ **do**
12:      $f^{(i)} \leftarrow -\mathcal{S}(\mathcal{M}^{(i)}; \mathcal{T})$    *// Negative score*
13:   **end for**
14:   *// Update distribution parameters via LRA-CMA-ES*
15:   $(\mathbf{m}^{(t+1)}, \sigma^{(t+1)}, \mathbf{C}^{(t+1)}) \leftarrow \text{LRA-CMA-ES-UPDATE}(\mathbf{m}^{(t)}, \sigma^{(t)}, \mathbf{C}^{(t)}; \{(\mathbf{X}_0^{(i)}, f^{(i)})\}_{i=1}^\lambda)$
16: **end for**
17: *// Final sampling from optimized distribution*
18: Sample $\{\mathbf{X}_{\text{Optim}}^{(i)}\}_{i=1}^{N_{\text{final}}} \sim \mathcal{N}(\mathbf{m}^{(T)}, (\sigma^{(T)})^2 \mathbf{C}^{(T)})$
19: Generate and decode: $\mathcal{M}^{(i)} = \boldsymbol{\zeta}_{\text{Dec}}(\Phi_\phi(\mathbf{X}_{\text{Optim}}^{(i)}))$
20: **Return** Top-scoring molecules $\{\mathcal{M}^{(i)}\}$

---

### K.3. Application to Flow-Based Molecular Optimization

We apply LRA-CMA-ES to optimize the initial-state distribution of our flow model. Let $\mathbf{X}_0$ denote the vectorized initial state (restricted to non-padding entries). The search distribution is:

$$p_\psi^{(t)}(\mathbf{X}_0) = \mathcal{N}\big(\mathbf{m}^{(t)}, (\sigma^{(t)})^2 \mathbf{C}^{(t)}\big), \tag{79}$$

initialized with $\mathbf{m}^{(0)} = \mathbf{0}$, $\sigma^{(0)} = \sigma_{\text{init}}$, and $\mathbf{C}^{(0)} = \mathbf{I}$.

At each iteration $t$, we sample $\lambda$ initial states $\{\mathbf{X}_0^{(t,i)}\}_{i=1}^\lambda$ from the current distribution $\mathcal{N}(\mathbf{m}^{(t)}, (\sigma^{(t)})^2 \mathbf{C}^{(t)})$. Each sample is passed through the frozen flow and decoder to obtain a molecule $\mathcal{M}^{(t,i)} = \boldsymbol{\zeta}_{\text{Dec}}(\Phi_\phi(\mathbf{X}_0^{(t,i)}))$, which is then scored as $f^{(t,i)} = -\mathcal{S}(\mathcal{M}^{(t,i)}; \mathcal{T})$. Based on the fitness values $\{f^{(t,i)}\}_{i=1}^\lambda$, we apply the LRA-CMA-ES update to obtain the next distribution parameters $(\mathbf{m}^{(t+1)}, \sigma^{(t+1)}, \mathbf{C}^{(t+1)})$. After $T$ iterations, we sample from the optimized distribution and return top-scoring molecules. The procedure is summarized in Algorithm 2.

**Parallelization.** Fitness evaluation (flow integration + decoding + docking) dominates the computational cost. Since evaluations are independent across population members, we parallelize across $\lambda$ workers. Each iteration then takes approximately the time of a single docking call ($\sim$1–5 seconds depending on molecule size and target complexity).

