# OpenReview forum: "Learning Canonical Representations for Unified 3D Molecular Modeling"
_ICML.cc/2026/Conference — Submitted to ICML 2026_

### Official Review · Reviewer_UsTs · 2026-02-16

**Soundness:** 2
**Presentation:** 3
**Significance:** 2
**Originality:** 2
**Overall Recommendation:** 3
**Confidence:** 4

**Summary:**

The paper proposes to use canonicalization of the molecule orientation to enhance molecule representation learning. This is used in the TheMol architecture to perform multiple molecule-related tasks within the same framework. This framework is evaluated on a number of standard benchmarks, as well as using specific metrics on the learned embedding space.

**Compliance With Llm Reviewing Policy:**

Affirmed.

**Final Justification:**

In my opinion, the motivation of the paper becomes much clearer with the example given in [this Reply Rebuttal Comment](https://openreview.net/forum?id=CrWIwbEb1d&noteId=Emgdb0XCnL), as it is a concrete case where I could see the benefit of the canonicalization approach over equivariance. I believe that there is value to the paper, but it currently has some writing issues in terms of the technical accuracy of certain statements as pointed out by kaHx and me, as well as the fact that it was not motivated well enough with the contents of the paper alone. I believe that this would require a number of changes to the framing in the paper, and I would prefer to see an updated version first before I can give a recommendation for acceptance.

One suggestion is to add a toy experiment that confirms the hypothesis of what happens in the [scenario](https://openreview.net/forum?id=CrWIwbEb1d&noteId=Emgdb0XCnL), which would be a strong motivation in favour of the canonicalization approach.

My judgment right now is that the paper has clear value already but needs more polish on the writing side. Therefore, I raise my score from reject to a weak reject for now, but with a strong encouragement to the authors that they should resubmit the paper once those improvements to the writing have been made, particularly to the core motivation.

**Key Questions For Authors:**

1. Can you further explain why you believe that equivariant representations (like I mentioned in my weakness point 1) are inferior to canonicalization? This would improve the argument in the paper for why canonicalization should be used.
2. In reference to my weakness point 3, how much do you think does the canonicalization aspect play a role in the results, in comparison to the numerous modeling choices in the framework itself? Since there is currently a lack of ablations for this aspect, answering this question sufficiently would greatly improve the scientific value of the paper.

**Limitations:**

Molecules have many applications, both positive and negative, so I encourage the authors to think about these and include an impact statement. It would also be good to see a more substantive discussion of potential limitations such as sample efficiency and inference speed.

**Strengths And Weaknesses:**

# Strengths
- Overall, the writing is clear and understandable.
- The unified framework supports a number of relevant real-world molecule tasks.
- The way of evaluating the molecular representations outside of the standard numerical benchmarks is interesting.
- The experimental results are quite good, showing the practicality of the proposed TheMol framework. The fact that the unified architecture performs decently across different tasks is impressive. I appreciate the novel qualitative evaluation of section 4.1.

# Weaknesses
1. I remain unconvinced on the arguments about the potential shortcomings of equivariant models in section 3.1.2. The authors mention the problems of using equivariance in an autoencoding framework, but the use of equivariance is broader than that, most notably in machine learning force fields (MLFF). For example, the line of work based on Equiformers (EquiformerV2, eSEN, UMA) can be trained to predict invariant energy and equivariant forces simultaneously, which can be used for property prediction and generating coordinates. It is easy to obtain invariant features from equivariant features, so the authors' argument of "learning to suppress rotational nuisance" does not work. With the Equiformer-based models at least, the invariant and equivariant features interact throughout the whole architecture, so relying on the invariant features at the end for an invariant prediction task is okay, without discarding much geometric information that the authors are concerned about. Overall, there is also no experimental validation that these claims of the authors are actual problems.
2. The novelty of the canonicalization aspect is somewhat limited because the main idea is basically an application of learned canonicalization like [1] to the molecular domain.
3. In my view, there are two separate contributions that are combined: the canonicalization and the unified framework. In the experiments, it is difficult to understand their individual contributions as there is a lack of ablations of this aspect. A simple ablation would have been to apply the same unified framework without the canonicalization, and one with the data augmentation strategy.
4. I find the discussion of related work somewhat lacking, I would have particularly liked to see a section on discussing related approaches for unified molecular frameworks and how they are different from TheMol.

[1]: Equivariance with Learned Canonicalization Functions. Kaba et al, ICML 2023. https://arxiv.org/pdf/2211.06489

---

> ### Author Rebuttal · Authors · 2026-03-27
>
> We would have liked to address each point in full detail; we apologize that the character limit requires substantial condensation.
>
> **W1, Q1.** Our question is not whether one model can handle both invariant and equivariant prediction, but how to represent 3D molecules so that **one autoencoder latent** can jointly support **prediction, generation, and optimization**. This is the setting studied in our paper. Prompted by the reviewer's comments, we extended our analysis of Repr.B to include cases where spherical equivariant modules (e.g., EquiformerV2) have been applied within autoencoders, discussed below.
>
> We fully acknowledge that Equiformer-family models excel at joint invariant/equivariant prediction, and that deriving invariant features from equivariant ones is straightforward. **What we wish to distinguish in this discussion is** whether a single model can perform both invariant and equivariant prediction well, **versus** whether a single model can perform prediction, generation, and optimization well (with a focus on drug discovery benchmark rather than MLFF). Our research was conducted to answer **the latter question**. Equivariant representations are well-suited for prediction, but when the same latent must also support generation and optimization, structural challenges arise.
>
> Specifically, in our framework optimization searches the latent space with frozen flow and decoder. If the latent is full equivariant irreps, it becomes fundamentally difficult to disentangle **whether a search direction reflects a chemically meaningful molecular change or merely orientation/gauge bookkeeping** — this is the core concern, compounded by the high dimensionality (e.g., EquiformerV2's 6,272 dims per node vs. TheMol's 64). Conversely, relying only on invariant features makes reconstructing distinct 3D poses extremely difficult.
>
> For generation, equivariant latents satisfy z(Rx)=ρ(R)·z(x), so rotated versions of the same molecule occupy different latent points, reducing latent generative modeling efficiency. AvgFlow (ICML 2025) reports rotation-induced trajectory crossings under conditional OT flow, proposing SO(3)-Averaged Flow as an alternative. AlphaFold3 and UAE-3D (NeurIPS 2025) similarly adopted non-equivariant diffusion with augmentation. In the **AE-based latent generation setting** that we premise, concerns regarding equivariant encoders are also empirically supported. ADiT (ICML 2025, corresponding to our Repr.C setting) compared an equivariant AE with EquiformerV2 encoder against a non-equivariant AE. The non-equivariant AE outperformed in both reconstruction and generation, with equivariant latent space proving less suitable for latent diffusion (p.20, Tables 6-7).
>
> TheMol resolves these tensions via canonicalization. We kindly refer the reviewer to Section 3.2 for a detailed description of the mechanism. We acknowledge that "learning to suppress rotational nuisance" was overstated given Equiformer's capabilities. We will reframe Section 3.1.2 to "latent-structure suitability." We are also integrating Equiformer-based module into our pipeline for direct comparison and will report results before the rebuttal deadline.
>
> We note that Appendix I.6 provides additional benchmarks; however, our focus was drug-discovery models, and force-field-oriented models were not examined with equal depth. We will extend this in the revision.
>
> **W3, Q2.** The requested ablations are already in **Appendix H** (including inference speed, failure-case analysis, canonicalization consistency error). Appendix H compares four paradigms in the same framework: Repr.A (invariant), Repr.B (equivariant), Repr.C (augmentation), and Repr.Z (ours). The results consistently demonstrate canonicalization's significant contribution independent of other modeling choices across all tasks. We would very much like to detail each result here, but the character limit prevents us — we kindly ask the reviewer to refer to Tables 15-18, 20, and 22 in Appendix H. We will reference these more prominently in the main text.
>
> **W4.** Appendix E.1 contains a comparative discussion of existing molecular foundation models and TheMol's distinctions. We agree this was not prominent enough and will move it to the main text or add an appendix table of contents.
>
> **W2.** We agree that canonicalization itself should not be framed as the sole novelty. Our contribution is establishing it as a **unified foundation-representation strategy** where one shared representation supports prediction, generation, and optimization, together with **generation-specific theory** and **representation-level evaluation criteria** beyond benchmark reporting. We also emphasize that competitive performance with one model across three task categories and five benchmarks is itself a strong foundation-model result. To our knowledge, TheMol is the first molecular representation model to remain competitive across all three task categories from one shared representation.

---

> > ### Author Rebuttal · Reviewer_UsTs · 2026-03-31
> >
> > I have read all reviews and rebuttals. Thank you for pointing me towards Appendix H and E.1, a table of contents as suggested would be appreciated for the 30 page appendix.
> >
> > > whether a search direction reflects a chemically meaningful molecular change or merely orientation/gauge bookkeeping
> >
> > I don't fully agree with this argument, as I don't see how the canonicalization approach avoids it. Instead of doing bookkeeping in an equivariant way with an equivariant model, shouldn't the same bookkeeping need to be done in the canonicalized frame?

---

> > > ### Author Response · Authors · 2026-04-01
> > >
> > > We agree that canonicalization does not eliminate orientation/gauge bookkeeping altogether; any 3D coordinate-based method must handle orientation somehow. The key distinction is **where this bookkeeping resides**. In an equivariant latent, orbit degrees of freedom remain inside the latent itself, so optimization and generation are performed on a space that mixes **intrinsic molecular variation** with **orientation/gauge variation**. In our approach, the gauge is handled once by the selector $\hat{R}_\theta$, which aligns the molecule to a canonical frame, after which the aligned 3D structure is mapped into latent space by the DiT encoder. The point is therefore not that bookkeeping disappears, but that it is **removed from the reusable latent on which downstream prediction, optimization, and generation operate**.
> > >
> > > We also do **not** claim that the learned canonical frame is the uniquely optimal frame among all possible gauges. The benefit comes from choosing a **consistent representative per rotation orbit**, so the downstream latent no longer needs to model the full orbit structure. The gain is therefore not in finding a universally best frame, but in removing redundant rotational degrees of freedom from the **working latent space**. Because the selector and encoder-decoder are learned jointly, this representative is not a fixed external convention but one that co-adapts to the backbone and is better aligned with downstream optimization and generation.
> > >
> > > This is the key difference from an equivariant latent. In the equivariant case, the latent itself must still carry orbit structure; in our method, the reusable latent is built **after** alignment in the canonical frame. Pose information is retained separately when equivariant outputs are needed, but the latent being optimized is more directly aligned with intrinsic molecular variation.
> > >
> > > | Method [*Optimization task*]|SMINA(↓) |VINA (↓) |SA (↑)|
> > > | :--- | :---: | ---: | ---: |
> > > |TheMol(EquiformerV2 latent baseline)| -7.04|-7.22|0.63|
> > > |TheMol(Repr.Z)(**Ours**)|-8.63|-9.12|0.75|
> > >
> > > | Method [*Generation task*]|Atom Sta(↑) |PB-Valid (↑)|Med.$\Delta$E(↓)|Med.$\Delta$R(↓)|
> > > | :--- | :---: | ---: | ---: | ---: |
> > > |TheMol(EquiformerV2 latent baseline)|81.7|87.3|5.78|0.43
> > > |TheMol(Repr.Z)(**Ours**)|86.8|93.7|4.97|0.17
> > >
> > > As committed in our rebuttal, we additionally completed the comparison with an EquiformerV2-based latent baseline. For generation, EquiformerV2 was used both for latent construction and as the latent-generation backbone. Even under this stronger Repr.B-style setting, TheMol (Repr.Z) remains consistently better on all reported optimization and generation metrics. This provides direct experimental support for our central claim: the key issue is not whether equivariant modules are expressive, but whether the resulting latent is structurally suitable for unified optimization and generation. (Generation alone is less problematic in this respect. However, we emphasize again that our goal is to identify a representation strategy that remains consistently strong across prediction, optimization, and generation, rather than one that is favorable for only a subset of tasks.)
> > >
> > > Relatedly, direct black-box optimization (e.g., CMA-ES or Bayesian optimization) on a fully equivariant latent is possible in principle, **but it introduces structural inefficiencies.** First, rotated versions of the same molecule occupy different latent points, so the optimizer must explore a redundant search space containing **multiple latent codes for what is chemically the same molecule.** Second, unconstrained perturbations can disrupt irrep-level coherence, producing candidates that are neither chemically meaningful variations nor valid pose changes. Conversely, restricting optimization only to invariant components avoids some of these issues, **but then distinct 3D poses cannot be reliably reconstructed.** This is why we view **latent-structure suitability**, rather than equivariance alone, as the central issue for a reusable latent that must support both optimization and generation.

---

### Official Review · Reviewer_uhRk · 2026-03-12

**Soundness:** 3
**Presentation:** 3
**Significance:** 3
**Originality:** 4
**Overall Recommendation:** 4
**Confidence:** 4

**Summary:**

This paper proposes a 3D molecular representation learning framework named THEMOL. Its core idea is to enable a single non-equivariant Transformer architecture to simultaneously support invariance tasks and equivariance tasks by canonicalizing molecules into a learned standard pose. The authors theoretically analyze the limitations of traditional equivariant architectures, invariant architectures, and data augmentation schemes, demonstrating that the canonicalization approach significantly reduces computational overhead while preserving expressiveness.

**Compliance With Llm Reviewing Policy:**

Affirmed.

**Final Justification:**

Most of my concerns have been addressed. I maintain my positive recommendation with a score of 4.

**Key Questions For Authors:**

See weaknesses.

**Limitations:**

Yes.

**Strengths And Weaknesses:**

**Strengths:**

1. The manuscript provides rigorous theoretical proofs, notably explaining how the canonicalization scheme successfully avoids pass-through degeneracy, a common failure mode in traditional equivariant autoencoders.
2. The paper moves beyond standard downstream benchmarks by proposing insightful representation quality metrics (e.g., Effective Dimension, Alignment Score, and Coherence). These metrics offer a more fundamental understanding of the latent space's geometric structure and semantic consistency.

**Weakness:**

1. While the theoretical sections discuss chirality and cite ChIRo [1], the experiments fail to evaluate this. Chirality is a critical aspect of 3D modeling. The authors should include tasks such as R/S classification, optical rotation prediction, or chirality-sensitive docking score predictions to substantiate their claims.

2. SE(3)-equivariant models like Equiformer are primarily valued for force field and potential energy surface calculations. These properties are extremely sensitive to fine-grained atomic perturbations. The proposed method lacks proof or analysis regarding its performance and stability in these physically demanding scenarios.

3. Yet it remains unclear if the model can sensitively perceive conformational changes within the same molecule [2]. Since physical properties fluctuate significantly across different conformers, the authors should demonstrate whether their framework achieves true physical knowledge modeling rather than just identifying molecular identity.

---
[1] Adams, K., Pattanaik, L., & Coley, C. W. Learning 3D Representations of Molecular Chirality with Invariance to Bond Rotations. In *International Conference on Learning Representations*.

[2] Wang, X., Zhang, Y., Zhang, Y., Cai, Y., & Wan, S. 3DCS: Datasets and Benchmark for Evaluating Conformational Sensitivity in Molecular Representations. In *The Fourteenth International Conference on Learning Representations*.

---

> ### Author Rebuttal · Authors · 2026-03-29
>
> Thank you for this important comment. We agree that the current manuscript would be stronger with more direct chirality- and conformer-sensitive evaluations.
>
> First, our present chirality claim is narrower than a full chirality benchmark. The paper’s point is that distance-only invariant representations can collapse reflection-odd information, whereas our canonicalization aligns rotations but does not apply reflections; hence it does not impose this O(3)-invariant collapse. This is the reason for the discussion in Sec. 3.1.4 / App. D.1 and the opposite-chirality visualization in Fig. 5. We agree, however, that this is not yet a task-level chirality evaluation.
>
> To address this directly, we will add a **held-out R/S classification experiment**. Specifically, we will construct a test set of molecules with **exactly one RDKit-defined tetrahedral stereocenter**, assign the **CIP label (R/S)**, generate multiple 3D conformers per molecule, extract **single-conformer embeddings without conformer averaging**, and train the same lightweight linear probe on top of frozen representations for all methods. This isolates whether the learned representation preserves stereochemical information rather than merely molecular identity. We will also report an **enantiomer-retrieval test**, where the model must distinguish same-connectivity opposite-chirality pairs from same-connectivity same-chirality conformers.
>
> Second, regarding Equiformer-style force-field / PES settings: we agree these are important and highly sensitive to fine-grained geometry. However, our paper does **not** claim to replace specialist QM/force models. The empirical scope of this work is unified molecular foundation modeling for **drug-discovery tasks**: property prediction, 3D generation, and structure-based optimization. Within this scope, we already evaluate physically grounded geometry quality in generation via **PB-Valid, $\Delta E_{\mathrm{relax}}$, and $\Delta R_{\mathrm{relax}}$**, but not forces or PES. We will revise the text to make this scope explicit and avoid overclaiming.
>
> Third, we agree that our current conformer analysis mainly reflects identity preservation across conformers (as evaluated by ROC-AUC in Sec. 3.3), rather than whether the representation resolves physically meaningful intra-molecular conformational variation. To address this directly, we will add a conformer-sensitivity analysis on held-out GEOM molecules with multiple conformers and energy annotations, measuring whether within-molecule embedding distances correlate with pairwise heavy-atom RMSD, torsional differences, and absolute conformer energy gaps. This evaluates conformer-level representation sensitivity, while remaining distinct from force/PES prediction, which is outside the present scope.
>
> We believe these additions will directly address the reviewer’s concern by separating three distinct questions: chirality preservation, conformer-level physical sensitivity, and force/PES scope. We thank the reviewer for pointing out that our current wording did not make these distinctions sufficiently explicit.

---

> > ### Author Rebuttal · Reviewer_uhRk · 2026-04-03
> >
> > I thank the authors for the detailed rebuttal and for clarifying the model's scope. The proposed plan to include R/S classification and conformer sensitivity analysis is technically sound and directly addresses my concerns regarding the physical and stereochemical awareness of the representation.
> >
> > However, since the actual numerical results for these critical experiments have not been provided in the current response, it is difficult to fully assess the effectiveness of the proposed additions at this stage. Therefore, I maintain my positive recommendation with a score of 4.

---

### Official Review · Reviewer_kaHx · 2026-03-13

**Soundness:** 2
**Presentation:** 3
**Significance:** 2
**Originality:** 2
**Overall Recommendation:** 2
**Confidence:** 3

**Summary:**

In this work, the authors propose a framework for canonicalizing molecules into a standard pose, such that a single model can support invariant and equivariant tasks.
The empirical results include Molecule classification datasets from the MoleculeNet, 3D molecule generation results on GEOM-DRUG and TDC Benchmark for property prediction.

**Compliance With Llm Reviewing Policy:**

Affirmed.

**Final Justification:**

My concerns about Prop D3 (my Q3) still remain. The authors do add some clarity on this on rebutal but it is not fully resolved. Also the argument for the latents seems to me, is subjective to equivariant architecture used and how. Here some theoretical arguments would have made a stronger claim.
The results presented cannot be claimed to generalized across these architectures which adds to the limited contributions in this work.
Given my concerns my these I maintain my score.

**Key Questions For Authors:**

1. Are there cases in which the gauge consistency loss go to zero? (in somewhat symmetric molecules maybe) What are the limitations in this case?
2. Assumption 3.1 is difficult to understand, and could help with a rewrite.
3. Proposition D3 mentions * In this case z_x serves as a coordinate pass-through rather than a compact representation.* this is unclear and in my opinion not true. If you extract a pose (i.e. in the case of SO(2), extract rotation angle through the equivariant architecture [1], the statement is untrue)
4. How is the Prop 3.3, 3.4 different from that in [2]?
5. Appendix line 1672 *Repr.B’s extremely low effective dimensionality (6.8%) confirms pass-through degeneracy: the latent is dominated by coordinate-like structure rather than learned abstractions.* considering an equivariant model represent only one element of the group orbit, it probably isnt pass through degeneracy. Could you explain why you mention it is so?



[1] Kendal Shape VAEs, Vadgama et al.

[2] Equivariance with Learned Canonicalization Functions, Kaba et al

**Limitations:**

No, the limitations are not discussed.

**Strengths And Weaknesses:**

## Strengths
1. The experimental section is detailed and considers more datasets across different tasks.
2. Presentation is good.



## Weaknesses
1. Missing citations on canonicalization [1], unified 3D molecule generation [2,3,4],etc.
2. The paper has incorrect statements or statements that are not well justified which raises concerns of claims in the paper.
3. The methodological contributions of this work is hard to gauge as primary works in the area are not cited, and with that in mind, the canonicalization part seems un-original in its current version.
4. It is unclear what alignment score truly represents.
5. The theoretical formalism could be a bit readable.


Missing citations
1. Equivariance with Learned Canonicalization Functions, Kaba et al
2. Mixed Continuous and Categorical Flow Matching for 3D De Novo Molecule Generation, Dunn et al
3. Controlled Generation with Equivariant Variational Flow Matching, Eijelboom et al
4. Fast, Expressive  Equivariant Networks through Weight-Sharing in Position-Orientation Space, Bekkers et al

---

> ### Author Rebuttal · Authors · 2026-03-27
>
> **Q3.** We agree with the reviewer that the SO(2) example is valid: an equivariant architecture can, in principle, extract meaningful pose information rather than trivially copying coordinates.
> The pass-through degeneracy in Repr.B was analyzed for coordinate-state autoencoders like GeoLDM, not as a general claim. Proposition D.3 describes this worst case and we will scope it accordingly in the revision.
>
> **However, our core argument does not depend on pass-through.** Our point is not that equivariance is unusable, but that in conventional 3D molecule **autoencoder or reprsentation settings,** where a single latent is expected to support prediction, optimization, and generation jointly, **latent-structure suitability remains the central issue.** Even when pass-through does not occur, full equivariant latents mix chemically meaningful variation with orientation bookkeeping, which complicates latent optimization.
> In generation, equivariant latents satisfy z(Rx)=ρ(R)·z(x), so the same molecule under different rotations occupies different points in latent space, forcing the flow model to learn across SO(3) orbits (Conversely, using only the invariant components from such embeddings makes coordinate reconstruction extremely difficult). AvgFlow (ICML 2025) shows that this rotation-induced multiplicity causes trajectory crossings and increased variance. AlphaFold3,UAE-3D(NeurIPS2025) similarly adopted non-equivariant diffusion.
>
> In autoencoder-based settings, **these concerns are also empirically supported.** ADiT (ICML 2025) compared an **Equivariant AE** with an EquiformerV2 encoder against a **non-Equivariance AE,** finding **the latter outperforms** in both reconstruction and generation, with equivariant latent space proving less suitable for latent diffusion (p.20, Tables 6–7).
>
> **We will therefore revise Section 3.1.2 to clarify that the issue is not “pass-through degeneracy” in general,** but latent-structure suitability for unified molecular representation learning.
>
> Relatedly, we also considered a stronger interpretation of the reviewer’s point: **namely, that an equivariant front-end might itself identify a canonical pose.** We agree this is a meaningful alternative to consider. In fact, we explored this direction early in the project. In our preliminary internal experiments, **however, unsupervised canonical-pose identification was substantially less stable than the present formulation,** especially in consistency across rotated inputs and in downstream suitability for reconstruction and generation(In 3D Molecule domain). Because these experiments were exploratory and not yet benchmarked to our standard, they were not included in the submission. We are now consolidating these existing results and will report them before the discussion deadline
> * * *
> **Q5.** We also agree that the low effective dimensionality in Appendix H need not be attributed solely to pass-through: since equivariant latents encode one orbit element, part of the capacity may be consumed by orbit structure itself. We will revise Appendix H to reflect both explanations. Nevertheless, **our central observation remains that canonicalization removes orientation at the input stage,** allowing the latent to focus on intrinsic molecular variation.
> * * *
> **Q1.** Yes. The loss can be zero if the selector is perfectly consistent on the data. The limitation appears for symmetric or near-symmetric molecules: several canonical poses may be equally valid, so the selected canonical pose can become ambiguous or discontinuous even when the loss is very small.
> **However, in practice, such exact symmetries are rare for drug-like conformations, and our method only requires approximate consistency on the data support**(chemical structures such as *benzene* are merely substructures and generally cannot serve as drugs on their own), Proposition A.5 provides the corresponding stability bound. We will clarify this limitation.
> * * *
> **Q2.** We agree and will rewrite Assumption 3.1 with clearer notation and intuition.
> * * *
> **Q4/W3.** We agree that the related-work discussion was insufficient. The canonicalization mechanism itself is not claimed as new. Our contribution is its use in **unsupervised molecular foundation representation learning** that jointly targets prediction, generation, and optimization **with competitive results**, together with generation-specific results (Propositions 3.4/C.5) and **the representation-quality evaluation criteria in Sec. 3.3.** We will clarify the relation to [2]: Proposition 3.3 is conceptually aligned with learned canonicalization, **whereas Proposition 3.4 and Proposition C.5 address generative modeling, which is outside the scope of [2].**
> * * *
> **W4.** **Alignment score measures principal-subspace similarity between pretraining and fine-tuning embeddings;** a high score indicates that downstream performance is well aligned with the pretrained representation rather than being created only by task-specific fine-tuning.

---

> > ### Author Rebuttal · Reviewer_kaHx · 2026-04-03
> >
> > I do not agree with this 'with equivariant latent space proving less suitable for latent diffusion', as it is caveated. It depends on how equivariance is achieved.
> > I also find the writing of the paper is not as per the standards of ICML as there were clearly statements that seemed not well thought out or incorrect (W2, Q2) I also notice that reviewer UsTs has pointed one of those out.

---

> > > ### Author Response · Authors · 2026-04-08
> > >
> > > **Before responding further, we respectfully ask that our rebuttal be read in full, as it directly addresses several of the concerns raised in the review.**
> > >
> > > We are not claiming that equivariance is bad in general. Our point is narrower: when the same representation must also support **generation and optimization**, there are additional practical issues to consider.
> > >
> > > Take one molecule $A$, and two rotated versions $A^{'}$ and $A^{''}$. They are chemically identical, but they appear as different coordinate states. At highly noisy diffusion steps, the clean structure is hard to identify. In that regime, the model cannot reliably tell which rotated version of the same underlying molecule it is seeing, so the ambiguity is not only about the molecule’s shape but also about its orientation (even though there is no need to learn or identify the orientation). As a result, training at early noisy steps can become more confusing and unstable in practice(At highly noisy states, the model cannot tell whether it should reconstruct $A$, $A^{'}$ , or $A^{''}$ ). We are not saying that this is mathematically invalid; rather, our point is about **practical learning difficulty**. A standard equivariant model can still be correct in principle, but from the standpoint of optimization and sample efficiency, it may have to spend capacity handling orientation ambiguity in addition to the molecular structure itself.
> > >
> > > This is also the direction of several recent works.[1,2,3] *Quotient-Space Diffusion Models* argues that, under SE(3) symmetry, the true target is naturally defined after factoring out rotation, and that standard equivariant diffusion still needs to learn the rotation-related part, while quotient-space modeling reduces that burden. It validates this idea on molecular structure generation. *AvgFlow* also shows that an SO(3)-averaged objective can converge faster and achieve better conformer generation quality than alignment-based alternatives.
> > >
> > > #### [1] Quotient-Space Diffusion Models [ICLR 2026]
> > > #### [2] AvgFlow [ICML2025]
> > > #### [3] UAE-3D [NeurIPS2025]
> > >
> > > Our motivation is closely related. We are not saying that symmetry handling disappears. Rather, we are saying that it is beneficial if the reusable latent does **not** need to keep all rotated versions of the same molecule as separate latent states. This is why we view canonicalization as helpful for a representation that must support **prediction, generation, and optimization together**.

---

### Official Review · Reviewer_CDey · 2026-03-17

**Soundness:** 3
**Presentation:** 3
**Significance:** 2
**Originality:** 2
**Overall Recommendation:** 3
**Confidence:** 4

**Summary:**

The authors propose a SE(3)-equivariant model THEMOL by centering geometry and learning a SO(3)-canonicalizer with regularization using a standard transformer. The effectiveness is justified over several benchmarkings, including MoleculeNet and TDC Benchmark.

**Compliance With Llm Reviewing Policy:**

Affirmed.

**Final Justification:**

The rebuttal did not address my concern. The paper spends substantial space discussing limitations of equivariant models, but in the end, the claim is "narrower" under a certain condition without actual proofs or empirical results. This still leaves a question unresolved: why not just use an equivariant model directly? As written, the motivation still appears weak, and I view the contribution as marginal.

**Key Questions For Authors:**

See weaknesses.

**Limitations:**

The authors should provide a section that explicitly states limitations, e.g., discontinuity of canonicalization.

**Strengths And Weaknesses:**

Strength:

- The paper is clearly written and demonstrated.

Weaknesses:
- I do not understand the argument that equivariant models suffer from pass-through degeneracy, which seems overstated. One can increase the dimension via additional channels or multiplicities over SO(3) irreps in spherical equivariant networks, so the representation is not restricted to the coordinate dimension. Invariants are also naturally included as the $\ell = 0$ components, so invariant and equivariant features can coexist in one network. As written, the motivation appears weak.
- Canonicalization itself suffers from discontinuity [1]. This issue is intrinsic and cannot be resolved.
- The authors fail to discuss existing canonicalization and model-agnostic equivariance work. There is a substantial literature on canonicalization [1] [2] [3], frame averaging [4], and related approaches [5] [6]. The idea is not new, and the theoretical contribution appears limited.
- The results are not consistently SOTA across several benchmarks. Why and what can be done to improve?

[1] Equivariant Frames and the Impossibility of Continuous Canonicalization. Dym., et al.

[2] Equivariance with Learned Canonicalization Functions. Kaba., et al.

[3] Generalization Bounds for Canonicalization: A Comparative Study with Group Averaging. Tahmasebi., et al.

[4] Frame Averaging for Invariant and Equivariant Network Design. Puny., et al.

[5] Equivariance via Minimal Frame Averaging for More Symmetries and Efficiency. Lin., et al.

[6] FAENet: Frame Averaging Equivariant GNN for Materials Modeling. Duval., et al.

---

> ### Author Rebuttal · Authors · 2026-03-28
>
> **We would have liked to address each point in full detail; we apologize that the character limit requires substantial condensation.**
>
> **W1.**
> We acknowledge the reviewer's points are technically correct: expanded channels over higher-degree irreps go beyond coordinate dimensions, and L=0 components naturally provide invariant features. The pass-through degeneracy in Repr.B was analyzed for coordinate-state autoencoders like GeoLDM, not as a general claim. Proposition D.3 describes this worst case and does not apply to high-dimensional equivariant modeling (e.g., EquiformerV2). We will scope it accordingly in the revision.
>
> Before sharing our perspective, we wish to make one thing clear. **Our work does not start** from the question of whether a single model can handle both invariant and equivariant representations. **Our work starts** from the question of what representation strategy is most effective **for a single model** (specifically an autoencoder-based representation model) **to perform prediction, optimization, and generation well.** Equivariant representations excel at prediction, but when the same latent must also support generation and optimization, structural challenges arise.
>
> In optimization, if the latent is full equivariant irreps, disentangling chemically meaningful directions from orientation bookkeeping becomes difficult — compounded by high dimensionality (equiformerV2: 6,272 dims per node). Conversely, relying only on invariant features makes reconstructing distinct 3D poses extremely difficult. In generation, equivariant latents satisfy z(Rx)=ρ(R)·z(x), so the same molecule under different rotations occupies different points in latent space, forcing the flow model to learn across SO(3) orbits.
>
> More broadly, recent work suggests that rotational ambiguity can make 3D generation harder to optimize. AvgFlow(ICML2025) addresses this with an SO(3)-averaged flow objective, and UAE-3D(NeurlPS2025), AlphaFold3 likewise adopts a non-equivariant diffusion module.
>
> **In AE setting, ADiT** (ICML 2025, corresponding to our Repr.C) **compared EquiformerV2-based AE against Non-equivariance AE,** finding the latter outperforms in both reconstruction and generation (Tables 6-7). TheMol resolves these tensions via canonicalization.
>
> We kindly refer the reviewer to Section 3.1.4 for a detailed description of the mechanism. **We will reframe Section 3.1.2 from "pass-through degeneracy" to "latent-structure suitability," and are integrating high-dimensional equivariant modules into our pipeline for direct comparison.**
>
> * * *
> **W2.**
> Dym et al. (ICML 2024) prove continuous canonicalization is generally impossible for SO(3). We acknowledge our failure to discuss this important result.
>
> However, we would like to discuss the claim that "this issue cannot be resolved" further. **(a)** The impossibility arises at points with non-trivial stabilizers; **exact symmetric configurations are rare and non-generic for drug-like conformations** (Appendix A.2). **(b)** Our R̂_θ learns approximate consistency via L_gc rather than exact continuity, aligned with Dym et al.'s own weighted frames alternative; Proposition A.5 provides a formal bound. **(c)** Table 19 confirms 99%+ of molecules achieve δ < 10⁻³. Dym et al.'s impossibility applies to exact continuous canonicalization; our learned approximate canonicalization practically circumvents this.
>
> **W3.**
> We agree the related work discussion was insufficient and will add a subsection covering canonicalization [1][2][3], frame averaging [4][5][6], and their relation to TheMol.
>
> Regarding novelty, we do not claim the canonicalization mechanism itself is new. Our contributions are: (a) Canonicalization for foundation representation learning — prior work [2] focuses on supervised prediction; **we operate in unsupervised pre-training unifying prediction, generation, and optimization, centering on 3D molecular representation for drug discovery.** Our choice of frame size=1 is motivated by generation efficiency: frame size k multiplies per-step cost by k. (b) Generation-specific theory — Proposition 3.4 (canonical-space score matching optimality) and **Proposition C.5 (orbit-level fidelity bound) address generative modeling, which none of [1]-[6] treat.** **(c) Representation quality evaluation — the six metrics in Section 3.3 offer rigorous evaluation criteria beyond benchmark performance.**
>
> **W4.**
> TheMol is a unified foundation model, not a task-specific specialist. A single pretrained representation serving prediction, generation, and optimization will not surpass every specialist on every benchmark. However, achieving this level of performance with one model across three task categories and five benchmarks is itself a strong foundation-model result. To our knowledge, TheMol is the first to remain competitive across all three from one representation. Further gains may come from more pretraining, longer fine-tuning, and task-specific adapter

---

> > ### Author Rebuttal · Reviewer_CDey · 2026-04-03
> >
> > I have the same issue as reviewer kaHx points out. I disagree that equivariant architectures are less suitable for latent diffusion than the canonicalization modeling.

---

> > > ### Author Response · Authors · 2026-04-08
> > >
> > > **Before responding further, we respectfully ask that our rebuttal be read in full, as it directly addresses several of the concerns raised in the review.**
> > >
> > > We are not claiming that equivariance is bad in general. Our point is narrower: when the same representation must also support **generation and optimization**, there are additional practical issues to consider.
> > >
> > > Take one molecule $A$, and two rotated versions $A^{'}$ and $A^{''}$. They are chemically identical, but they appear as different coordinate states. At highly noisy diffusion steps, the clean structure is hard to identify. In that regime, the model cannot reliably tell which rotated version of the same underlying molecule it is seeing, so the ambiguity is not only about the molecule’s shape but also about its orientation (even though there is no need to learn or identify the orientation). As a result, training at early noisy steps can become more confusing and unstable in practice(At highly noisy states, the model cannot tell whether it should reconstruct $A$, $A^{'}$ , or $A^{''}$ ). We are not saying that this is mathematically invalid; rather, our point is about **practical learning difficulty**. A standard equivariant model can still be correct in principle, but from the standpoint of optimization and sample efficiency, it may have to spend capacity handling orientation ambiguity in addition to the molecular structure itself.
> > >
> > > This is also the direction of several recent works.[1,2,3] *Quotient-Space Diffusion Models* argues that, under SE(3) symmetry, the true target is naturally defined after factoring out rotation, and that standard equivariant diffusion still needs to learn the rotation-related part, while quotient-space modeling reduces that burden. It validates this idea on molecular structure generation. *AvgFlow* also shows that an SO(3)-averaged objective can converge faster and achieve better conformer generation quality than alignment-based alternatives.
> > >
> > > #### [1] Quotient-Space Diffusion Models [ICLR 2026]
> > > #### [2] AvgFlow [ICML2025]
> > > #### [3] UAE-3D [NeurIPS2025]
> > >
> > > Our motivation is closely related. We are not saying that symmetry handling disappears. Rather, we are saying that it is beneficial if the reusable latent does **not** need to keep all rotated versions of the same molecule as separate latent states. This is why we view canonicalization as helpful for a representation that must support **prediction, generation, and optimization together**.

---

### Decision · Program_Chairs · 2026-04-30

**Decision:**

Reject

**Comment:**

The authors propose learning canonicalization methods for 3D rotational transformations with applications in molecular modeling.

All reviewers appreciated the results and contributions and found the paper interesting. However, they mention some issues, mostly regarding the presentation of the results.


>  Reviewer CDey: This still leaves a question unresolved: why not just use an equivariant model directly? As written, the motivation still appears weak, and I view the contribution as marginal.

I believe the contribution of the paper is somewhat marginal, but this is merely a problem that the authors can address by improving the presentation, as the results are already really interesting. Reviewer CDey asked for more discussion on equivariance and canonicalization issues, as well as citing related work in the next version. Reviewer kaHx raises concerns about Prop D3 in their final justification, while Reviewer uhRk asks about the experimental results.

>  Reviewer UsTs: it currently has some writing issues in terms of the technical accuracy of certain statements as pointed out by kaHx and me

Note: The following reference in the paper does not exist; please consider modifying it in the next version.

Adams, K., Pattanaik, L., and Coley, C. W. Chiro:
Chirality-aware 3d molecule encoder. arXiv preprint
arXiv:2111.08638, 2021.